# Design of facilitated dissociation enables timing of cytokine signalling

Adam J. Broerman[1,2,3 ✉], Christoph Pollmann[4,5], Yang Zhao[6,7], Mauriz A. Lichtenstein[1,8], Mark D. Jackson[9], Maxx H. Tessmer[9], Won Hee Ryu[10], Masato Ogishi[6,7], Mohamad H. Abedi[1,2], Danny D. Sahtoe[1,2], Aza Allen[1,2], Alex Kang[1,2], Joshmyn De La Cruz[1,2], Evans Brackenbrough[1,2], Banumathi Sankaran[11], Asim K. Bera[1,2], Daniel M. Zuckerman[10], Stefan Stoll[9], K. Christopher Garcia[6,7,12], Florian Praetorius[1,2,14 ✉], Jacob Piehler[4,5] & David Baker[1,2,13 ✉]

Protein design has focused on the design of ground states, ensuring that they are sufficiently low energy to be highly populated[1]. Designing the kinetics and dynamics of a system requires, in addition, the design of excited states that are traversed in transitions from one low-lying state to another[2,3]. This is a challenging task because such states must be sufficiently strained to be poorly populated, but not so strained that they are not populated at all, and because protein design methods have focused on generating near-ideal structures[4–7]. Here we describe a general approach for designing systems that use an induced-fit power stroke[8] to generate a structurally frustrated[9] and strained excited state, allosterically driving protein complex dissociation. X-ray crystallography, double electron–electron resonance spectroscopy and kinetic binding measurements show that incorporating excited states enables the design of effector-induced increases in dissociation rates as high as 5,700-fold. We highlight the power of this approach by designing rapid biosensors, kinetically controlled circuits and cytokine mimics that can be dissociated from their receptors within seconds, enabling dissection of the temporal dynamics of interleukin-2 signalling.

Protein–protein interactions orchestrate much of biological function. High-affinity interactions enable protein circuits to respond to low concentrations of stimuli and to act potently on targets; fast exchange enables them to respond quickly to changes in stimuli. These two properties cannot usually be achieved simultaneously in binary interactions because they depend on the interaction off-rate in opposite ways: high affinity usually requires slow dissociation (low off-rate), whereas rapid exchange requires fast dissociation (high off-rate) (Supplementary Fig. 1). Several natural systems exhibit 'facilitated dissociation'[10–20], in which an effector (E) can bind to a target–host (TH) complex to form an excited ternary complex (THE)[20–26] from which the target dissociates quickly (Fig. 1a–c). In such a system, the target can bind tightly to the host, yet can also be rapidly released by adding the effector[27]. In engineered DNA systems, the kinetic control afforded by an analogous phenomenon (toehold-mediated strand displacement) has enabled the construction of many complex functions[28,29], but DNA systems have limited utility for directly interfacing with biology. Protein binding and unbinding can be readily coupled to biological processes, but there has been no general approach to design kinetic control over protein interactions.

We set out to design protein systems that undergo facilitated dissociation. Given an interacting protein binder–target pair, we reasoned that we could construct host proteins with controllable dissociation kinetics by fusing an effector-responsive conformational switch to the binder such that when the effector is not bound, the target can bind normally, but in the alternate effector-bound conformation, the switch clashes with the target, leading to strain in the target–host–effector ternary complex that resolves when the target dissociates[30] (Fig. 1d,e; nomenclature detailed in Supplementary Note 1). This would allosterically couple the effector and target, and facilitated dissociation could proceed through the strained ternary complex intermediate faster than spontaneous dissociation of the target in mutually exclusive competition (Fig. 1c). Crucially, the energy of this ternary intermediate must be neither too high (otherwise the facilitated dissociation pathway would not be faster) nor too low (otherwise the target would not dissociate) (Fig. 1b). To set the ternary intermediate energy within this optimal range (Supplementary Fig. 2), we reasoned that we could control the level of strain in the ternary complex by varying the geometry of the switch–binder fusion. To access this strained state effectively, we reasoned that binding the effector should rapidly drive the conformational change against the resisting force associated with generating strain. Such a 'driven' motion would be akin to the power strokes of motor proteins: a large conformational change that is both thermodynamically and kinetically favoured (a low-barrier descent down a steep

[1]Institute for Protein Design, University of Washington, Seattle, WA, USA. [2]Department of Biochemistry, University of Washington, Seattle, WA, USA. [3]Department of Chemical Engineering, University of Washington, Seattle, WA, USA. [4]Department of Biology/Chemistry, Osnabrück University, Osnabrück, Germany. [5]Center for Cellular Nanoanalytics, Osnabrück University, Osnabrück, Germany. [6]Department of Molecular and Cellular Physiology, Stanford University School of Medicine, Stanford, CA, USA. [7]Department of Structural Biology, Stanford University School of Medicine, Stanford, CA, USA. [8]Institute for Chemistry and Biochemistry, Freie Universität Berlin, Berlin, Germany. [9]Department of Chemistry, University of Washington, Seattle, WA, USA. [10]Department of Biomedical Engineering, Oregon Health and Science University, Portland, OR, USA. [11]Molecular Biophysics and Integrated Bioimaging, Lawrence Berkeley National Laboratory, Berkeley, CA, USA. [12]Howard Hughes Medical Institute, Stanford University, Stanford, CA, USA. [13]Howard Hughes Medical Institute, University of Washington, Seattle, WA, USA. [14]Present address: Institute of Science and Technology Austria, Klosterneuburg, Austria. ✉e-mail: broerman@uw.edu; florian.praetorius@ist.ac.at; dabaker@uw.edu

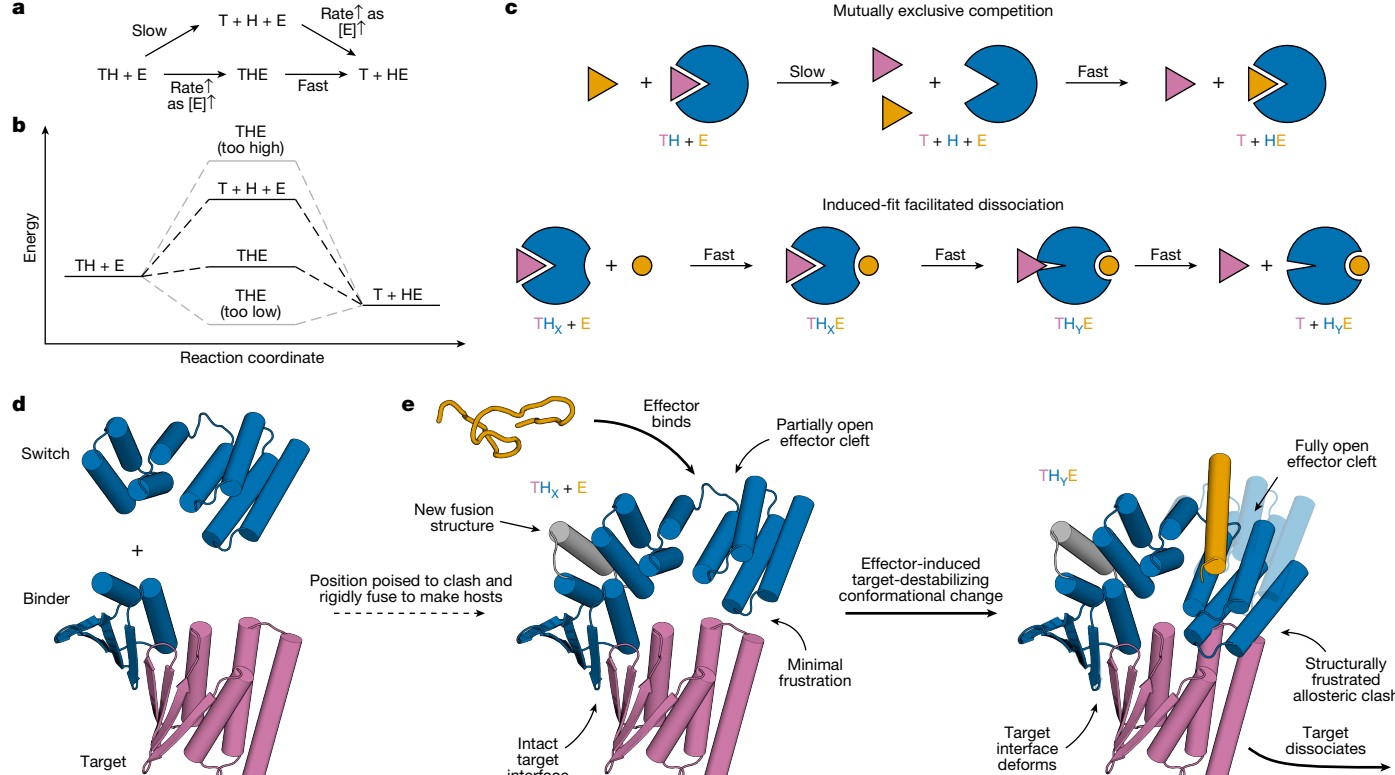

**Fig. 1 | Strategy for designing proteins that reconfigure through facilitated dissociation. a–c**, A high-affinity interaction can rapidly exchange through facilitated dissociation (bottom pathways), but not through mutually exclusive competition (top pathways). **a**, Reaction diagram. **b**, Energy diagram. **c**, Schematic of induced-fit facilitated dissociation (bottom) compared with slow mutually exclusive competition (top). The host protein (H, subscripted by conformational state X or Y) is shown in blue, the target (T) in pink and the effector (E) in orange.

**d**, Structural models of starting components (effector-responsive switch and arbitrary binder–target pair) combined to construct facilitated dissociation systems. **e**, Structural models of example proteins designed to undergo a facilitated dissociation process starting from a tightly interacting state X (left) through a structurally frustrated ternary intermediate in state Y (right, solid). State X (transparent) is included to show the conformational change. Thinner arrows indicate structural features and thicker arrows indicate state changes.

energy gradient)[8,31]. Compared with the mechanism of direct steric overlap between target and effector, as in many existing facilitated dissociation systems[17–20,26,32], this allosteric mechanism should be quite modular: such a force-generating switch could be fused to almost any binder to enable facilitated dissociation of its target.

We began by using a previously designed effector-responsive conformational switch (hinge protein cs221; ref. 33) to test the concept of allosteric coupling by switching steric clashes between the host and the target. This switch can undergo a rigid-body hinge motion to transition from a closed state (X) to an open state (Y), and in the open state Y can bind an effector peptide quite tightly ($k_{off} = 5 \times 10^{-6}$ s$^{-1}$). As a model binder–target interaction, we chose a designed heterodimer pair (LHD101 (ref. 34) modified as described in Supplementary Fig. 3) with dissociation slow enough ($k_{off} = 9 \times 10^{-5}$ s$^{-1}$) to allow us to easily measure substantial effector-induced acceleration of target dissociation, but not slower than the effector dissociation so that the target would be more likely than the effector to dissociate from the ternary complex. To allosterically couple binding of the effector to dissociation of the target, we designed structured fusions of the hinge switch and binder such that when the switch is in state X, the target can bind, but when in the effector-bound state Y, it will clash strongly (Fig. 1d,e, Supplementary Fig. 4a,b and Methods). We obtained synthetic genes encoding 12 designs, expressed and purified the proteins from *Escherichia coli* and found that the best (allosteric switch 0; AS0) showed slow and reduced effector association in the presence of the target (Supplementary Fig. 5d), indicating the desired allosteric coupling but also the need for designs with a faster, driven pathway for effector binding.

In these first designs, because the effector-binding cleft is closed in state X, the hinge switch must first transition to the open state Y before the effector can bind (a conformational selection mechanism). Because state Y clashes with the target, this conformational change is slow when the target is bound, limiting the rate of effector association and the overall rate of facilitated dissociation (Supplementary Fig. 4c). This slow step could be bypassed if the effector could instead first bind to state X and accelerate the transition to state Y. An induced-fit mechanism could provide the driving force: an intrinsically disordered effector could weakly engage with state X, and fold and make more extensive interactions in the strained ternary complex in state Y[35], driving the transition in a power-stroke-like motion[36,37].

Such a mechanism would require new switches that, in state X and throughout the conformational transition, retain an open effector-binding cleft where a flexible effector could associate (Fig. 1e and Supplementary Fig. 4d). Starting from AS0, we constructed a new state X by shifting the two domains from their state Y positions relative to each other (by one heptad along the helix of domain 1 that contacts domain 2), building a new loop between the domains, and optimizing single sequences that could adopt both this new state X and the original effector-bound state Y (Supplementary Fig. 4e and Methods). This approach maintains the open cleft in state X by introducing minimal rotation from state Y and simplifies the multi-state sequence-design challenge by minimizing the local structural differences between the two states. Transition between conformational states occurs by a register shift throughout which the cleft could remain open and bind the effector (Supplementary Fig. 4f). Because these new hosts retain the state Y backbone validated to clash with the target in AS0,

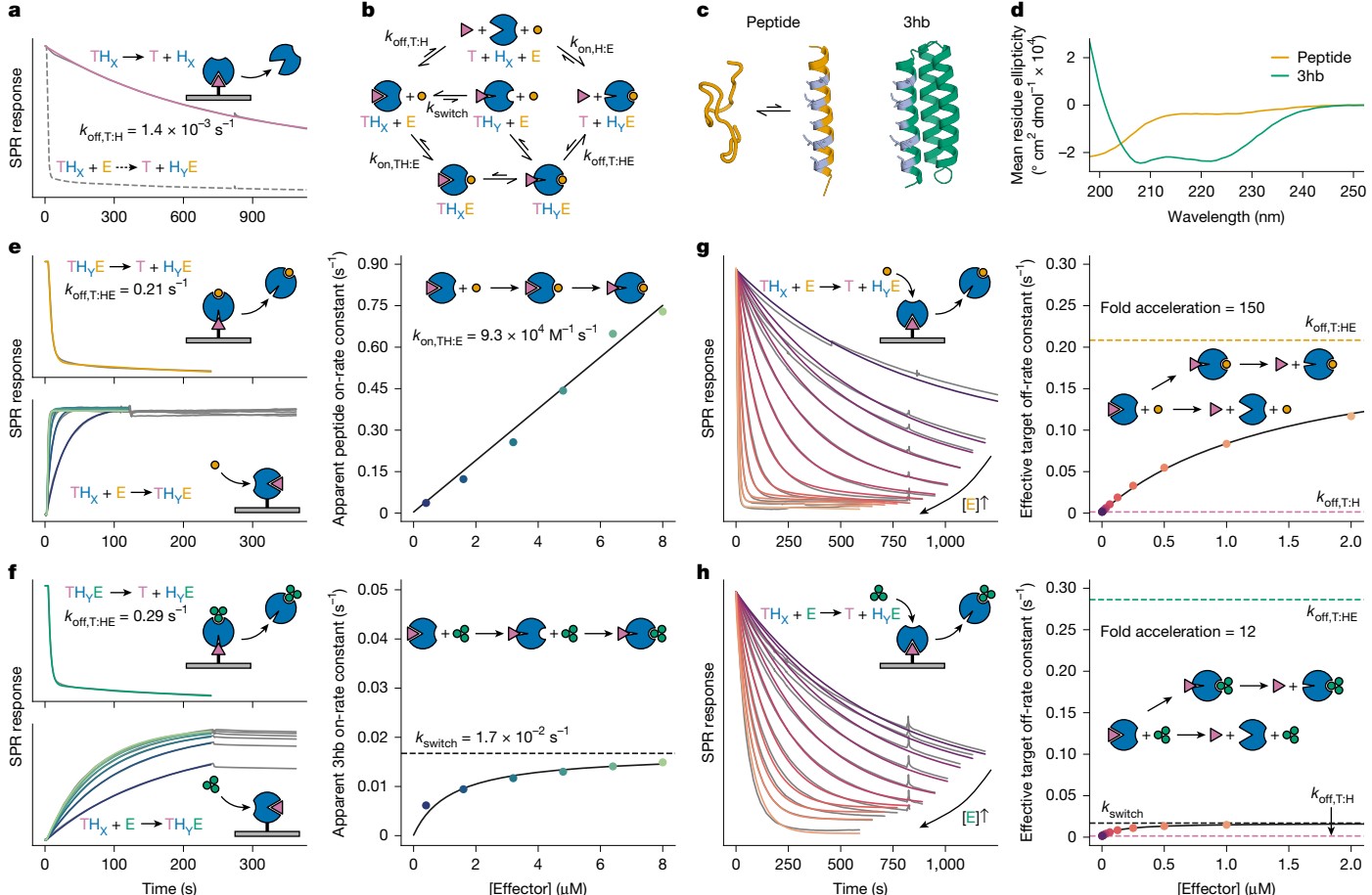

**Fig. 2 | Kinetic characterization of facilitated dissociation in AS1. a**, Slow dissociation of the target from the host in the absence of effector (solid line) and fast dissociation in the presence of 2 μM effector (dashed line) assessed by SPR. Slow dissociation data (solid grey) fitted with a double exponential (pink). **b**, Kinetic model describing pathways of competition. Top, mutually exclusive competition; middle, facilitated dissociation with effector binding rate-limited by conformational selection; bottom, facilitated dissociation with induced-fit effector binding. The $k$ labels are rate constants. **c**, Cartoon representations of the peptide (left) and 3hb (right) effectors; interface residues are shown in grey. **d**, Circular dichroism spectra of the peptide and 3hb effectors. **e,f**, Kinetic characterization of the formation and breakage of the ternary complex intermediate with the peptide (**e**) and 3hb (**f**) effectors. Top left, fast dissociation of the target from the ternary complex; data (grey) fitted with double exponentials (orange and green). Bottom left, effector association to form the ternary complex and extremely slow subsequent dissociation; data (grey) fitted with single exponentials (colours) in the association phase. Right, apparent effector on-rate constant plotted against effector concentration (circles) and a linear (**e**) or hyperbolic (**f**) fit. $k_{switch}$ is the saturating value of the hyperbolic fit. **g,h**, Kinetic characterization of the full facilitated dissociation pathway with the peptide (**g**) and 3hb (**h**) effectors. Left, effector-concentration-dependent dissociation of the target; data (grey) fitted (colours) as described in the Methods. Right, rate constant of facilitated target dissociation plotted against effector concentration (circles) and fitted with a hyperbolic equation (black line). In **a** and the left plots of **e–h**, schematics show the arrangement of proteins relative to the SPR chip (grey). In the right plots of **e–h**, schematics show the mechanism that can be inferred from the data.

we should observe allosteric coupling if the switch works as designed (Fig. 1e).

We obtained synthetic genes encoding ten such designs, and found that four tightly bound the effector (dissociation constant ($K_d$) < 1 nM; Supplementary Fig. 6d). To measure facilitated dissociation kinetics, we used surface plasmon resonance (SPR): with the target affixed to the SPR surface, we incubated with the host, then measured target–host dissociation under flow of various concentrations of effector. For these four designs, the target dissociates slowly in the absence of the effector; adding the effector increases the rate of dissociation markedly (Fig. 2a and Supplementary Fig. 7), but minimally affects the target off-rate from a control static binder fusion that lacks an effector-binding site (Supplementary Fig. 7).

To investigate the mechanism of facilitated dissociation (Fig. 2b), we focused on AS1, the design with the tightest effector binding ($K_{d,H:E} \approx 10$ pM). We directly measured the rate constant of target dissociation from the strained ternary complex ($k_{off,T:HE}$) by flowing pre-incubated host–effector complex at a high concentration to form

the strained ternary complex with the target on the SPR surface, then tracking target–host dissociation under continued flow of the effector (Fig. 2e, Supplementary Fig. 7 and Methods). In the full facilitated dissociation process, as the concentration of added effector increases, the rate constant of facilitated target dissociation approaches $k_{off,T:HE}$ (Fig. 2g and Supplementary Fig. 7), strongly suggesting that the ternary complex is an intermediate in the facilitated dissociation process (Fig. 2b, lower pathways).

To further analyse the mechanism of forming the strained ternary complex, we characterized facilitated dissociation with two different effectors: the peptide and a three-helix bundle '3hb' (3hb21; ref. 33) (Fig. 2c). The two effectors make nearly identical interactions with AS1, but when unbound, the peptide is disordered whereas the 3hb is structured (Fig. 2d). With AS1 affixed to the SPR surface, we measured the rate of effector association to form the strained ternary complex by first saturating AS1 with target, then flowing varying concentrations of effector mixed with constant excess target (to ensure that the target remains bound after effector association). The apparent on-rate for

binding to the target–AS1 complex increases linearly with concentration for the peptide (Fig. 2e) but hyperbolically for the 3hb, saturating at the rate of a concentration-independent step (Fig. 2f). Notably, with the 3hb, the rate of facilitated target dissociation saturates at this same value (Fig. 2h). The simplest explanation of these results is that the rigid 3hb can only bind to the fully open state Y of AS1 and that the $TH_X \rightarrow TH_Y$ conformational change is slow (owing to partial blocking by the bound target), so it becomes rate limiting for both the association with the target–AS1 complex and the overall facilitated dissociation process (Fig. 2b, middle pathway). In contrast to the 3hb, peptide effector binding (Fig. 2e) and resulting target destabilization (Fig. 2g) can both occur more rapidly than the $TH_X \rightarrow TH_Y$ conformational change, suggesting that the more flexible peptide effector can bind to AS1 in state X to accelerate this conformational change through an induced-fit mechanism[12,35,38] (Fig. 2b, bottom pathway).

We used X-ray crystallography to structurally characterize multiple states of our designed systems. For both the AS1 and the AS5 systems, the crystal structures of the hosts alone (Fig. 3a and Extended Data Fig. 1a) and of the host–effector complexes (Fig. 3b and Extended Data Fig. 1b) closely match the design models (maximum 1.3 Å Cα root mean square deviation (RMSD)). The unbound structures show an open hydrophobic cleft in the new designed state X poised to bind the effector, and the effector-bound structures show that binding the effector causes the two switch domains to register-shift into the designed state Y (Supplementary Note 2). Structures of the target–AS1 complex show that the target binds as designed (Fig. 3c, top), with some strain suggested by small variations among the structures (Fig. 3c, bottom, Extended Data Fig. 1c,d and Supplementary Note 3).

We next sought to determine how the target–host–effector ternary complex deforms to resolve the designed clash. At the concentrations of the components used in the above experiments, the AS1 ternary complex is an only transiently populated excited state, but at high concentrations it becomes the dominant state ($K_{d,T:HE}$ = 200 nM; Supplementary Fig. 9). This enabled us to solve structures of AS1 in the ternary complex intermediate with both target and effector. In structures from two different crystals, the switch region closely matches the state Y design model, and the rest of the structure strains to resolve the structural frustration from simultaneously binding the target and occupying state Y (Fig. 3d). This strain distributes across multiple locations in the structure: the binder fusion bends, the portion of the target that directly clashes with the switch becomes disordered and the target twists at its interface, disrupting the interfacial hydrophobic packing (Fig. 3e, top) and shearing the interfacial strand pairing (Fig. 3e, bottom). Double electron–electron resonance (DEER) spectroscopy (Extended Data Fig. 2) and molecular dynamics (MD) simulations (Extended Data Fig. 3) suggest that the ternary complex is dynamic, with varying amounts of strain at each location (Supplementary Note 3).

## Modulation of dissociation acceleration

We next investigated the factors that contribute to the dissociation kinetics, seeking to maximize the dissociation rate enhancement by (1) reducing the base rate of target dissociation and (2) increasing the rate of facilitated dissociation. For (1), the target dissociates 20-fold faster from AS1 than from an unhindered binder fusion (Supplementary Fig. 7), probably because of minor strain in the target–AS1 complex (Extended Data Fig. 1c). For (2), with AS1, effector association can occur at least five times faster than target dissociation from the ternary complex (Fig. 2e); thus, moderately increasing the energy of the ternary complex could accelerate the target dissociation step without making the effector association rate limiting, thereby increasing the overall rate of facilitated dissociation. Assuming a simple spring model of the ternary complex, this could be achieved by deforming with a greater magnitude or in a direction of higher stiffness (Supplementary Fig. 10a).

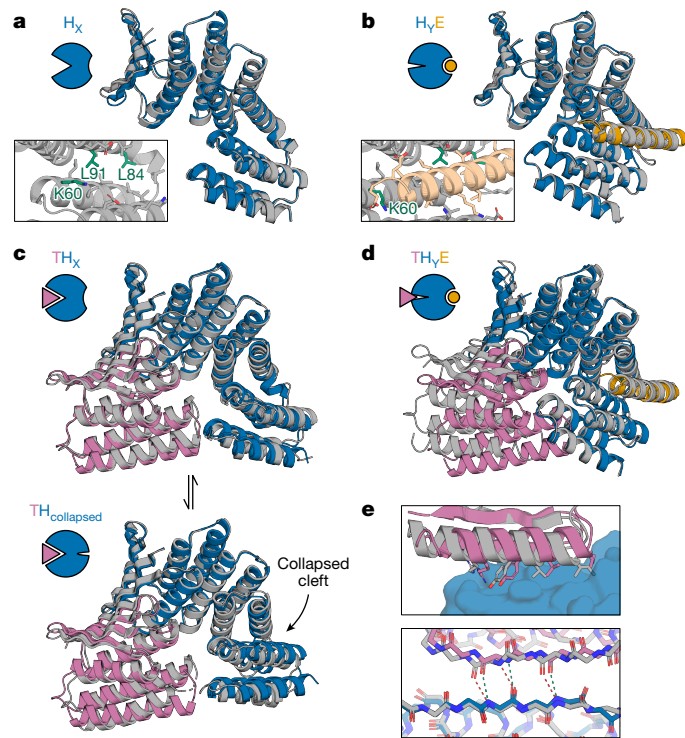

**Fig. 3 | Structural characterization of AS1. a**, Crystal structure of AS1 alone (grey) overlaid with the design model of AS1 in state X (blue). Inset, detailed view of side chains in the partially open effector-binding cleft. **b**, Cocrystal structure of AS1 and peptide effector (grey) overlaid with the design model of the AS1–effector complex in state Y (blue and orange). Inset shows the same view as in **a**. **c**, Top, cocrystal structure of AS1 (with intact cleft) and target (grey) overlaid with the design model of the target–AS1 complex in state X (blue and pink). Bottom, cocrystal structure of AS1 (with collapsed cleft) and target (grey) overlaid with the design model of the target–AS0 complex (blue and pink) whose state X resembles this collapsed state. **d**, Cocrystal structure of AS1 (with hydrophobic surface mutations), target and peptide effector (grey) aligned at the switch region with the design models of the target (pink) and AS1–effector complex in state Y (blue and orange) showing the designed clash. **e**, Top, detailed view of the target interface side chains in the ternary complex (grey) and the target–AS1 complex (pink) interacting with AS1 (blue). Bottom, detailed view of the backbone hydrogen bonding in the interfacial strand pairing. The target–AS1 complex (pink and blue) hydrogen bonds (green) are less strained than the ternary complex (grey) hydrogen bonds (red).

We sought to maximize strain energy in the ternary complex by modulating the magnitude and direction of the deformation required to resolve the designed clash (Fig. 4a). Avoiding any clash in state X while maintaining a strong clash in state Y, we sampled a variety of target + binder positions relative to the switch from AS1; rebuilt fusions between the switch and the newly located binder; and selected variants that were predicted by AlphaFold2 (AF2) to have substantial deformations spanning a variety of directions (AF2 predictions of the strained AS1 ternary complex were within 1.0 Å Cα RMSD of the target–AS1–effector crystal structure). These changes to the fusion region, although distant from both binding sites, caused considerable variation in the kinetics of target dissociation and their modulation by the effector (Supplementary Fig. 11). Most of these variants showed reduced target off-rates (compared with AS1) in the absence of effector that increased in the presence of effector (Supplementary Fig. 11 and Supplementary Table 1), and for the fastest variant (AS117), adding effector accelerated target dissociation by 2,400-fold, reaching a rate exceeding that of the original AS1 (Fig. 4b). By comparing the predicted deformations with the facilitated dissociation rates in the forward and reverse directions (Supplementary Note 4), we found that the global strain energy of the

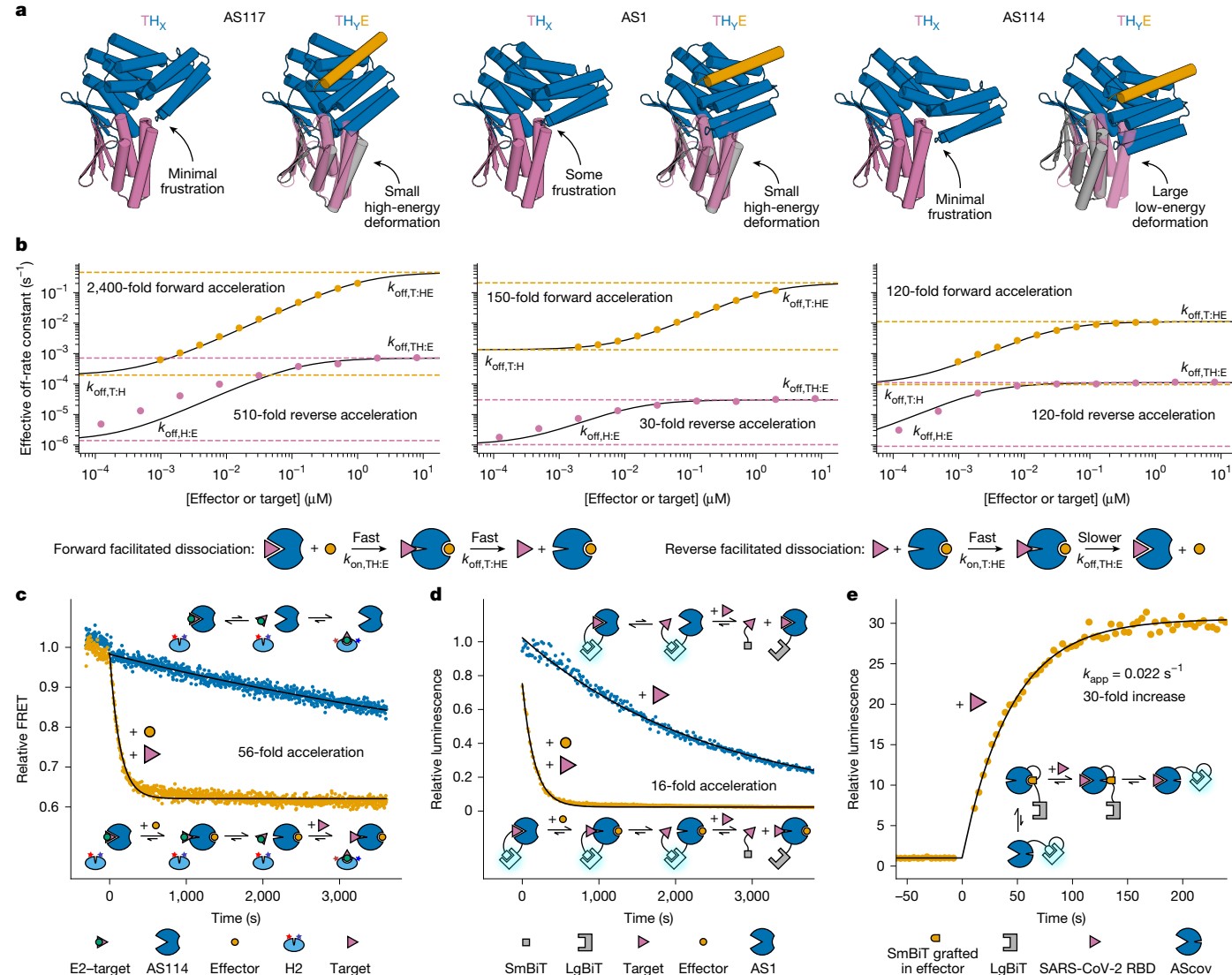

**Fig. 4 | Modulation and applications of facilitated dissociation.**
**a,b,** Comparison of three representative designs with different facilitated dissociation kinetics. **a,** For each design: left, model of host in state X (blue) aligned to the target (pink) to show any clash influencing the target off-rate in the absence of effector; right, model of host–effector complex in state Y (blue and orange) aligned to the target (pink) to show the designed clash, and (grey) AF2 prediction of the target position relative to the switch in the ternary complex to show how the clash resolves through global strain. **b,** Forward and reverse facilitated dissociation rates: target off-rate constants versus effector concentration (orange circles) and effector off-rate constants versus target concentration (pink circles) fitted with hyperbolics (black lines). Cartoons illustrate the forward and reverse facilitated dissociation pathways. Dashed lines mark the base and accelerated off-rate constants for forward (orange) and reverse (pink) facilitated dissociation. **c,** Chain reactions. FRET time courses showing slow transfer of a kinetically trapped effector (blue, top schematic) and accelerated transfer through facilitated dissociation (orange, bottom schematic). **d,** Breaking split enzymes. Luminescence time courses showing breakage of a reversible split luciferase through slow direct competition (blue, top schematic) and faster facilitated dissociation (orange, bottom schematic). **e,** Rapid sensing. Luminescence time course (orange) showing rapid sensing of SARS-CoV-2 through facilitated dissociation (schematic). **c–e,** Data fitted with single exponentials (black lines).

ternary complex depends on both the magnitude and the direction of the deformation (Supplementary Fig. 10) and that strain can distribute non-uniformly throughout the structure, leading to kinetic asymmetry[24] (Supplementary Fig. 12 and Extended Data Fig. 4).

## Applications of facilitated dissociation

We next set out to use facilitated dissociation to construct protein systems with kinetic behaviours that were previously inaccessible to design. First, inspired by toehold-mediated strand displacement in DNA[28], we sought to create a kinetically trapped system which, on stimulation, quickly reconfigures through a chain reaction. To investigate this, we fused the effector peptide (E2) for a second reporter hinge (H2) to the target such that E2 is occluded when this fusion (E2–target)

is bound to a host (Extended Data Fig. 5). As intended, the release of E2–target from AS114 and the subsequent switching of H2 is slow, but accelerates markedly upon addition of the original effector, which accelerates E2–target dissociation (we include excess original target to prevent E2–target rebinding[39]; Fig. 4c and Extended Data Fig. 5). In principle, multiple orthogonal hosts could be constructed and chained together through such target–effector fusions.

Second, we reasoned that our designs could complement split protein systems with high affinity and enable them to be switched off rapidly. To test this, we tagged AS1 and the target with NanoBiT split luciferase fragments, LgBiT and SmBiT (ref. 40). When combined, these components exhibit high luciferase activity that disappears much more rapidly upon addition of effector and excess untagged target than upon addition of excess untagged target alone (Fig. 4d).

Third, in previously designed thermodynamically controlled biosensors that operate through a conformational selection mechanism, there is a trade-off between dynamic range and response time (reducing background by lowering the energy of the 'off' state generally slows interconversion with the 'on' state)[41,42]. We reasoned that sensors based on facilitated dissociation could be limited by the rate of target association rather than by internal conformational switching, and found that by caging the SmBiT peptide within the effector bound to a host, facilitated dissociation (in the reverse direction compared with previously) upon binding the target uncaged the SmBiT rapidly to enable luciferase reconstitution (Methods). Using this strategy together with a designed SARS-CoV-2 receptor-binding domain (RBD) binder (LCB1; ref. 43), we generated 16 'AScov' sensor designs in which the RBD clashes with the switch in state Y but not in state X. After addition of SARS-CoV-2 RBD, the best sensor shows a 30-fold increase in luciferase activity, with a half-time of 30 s (Fig. 4e and Extended Data Fig. 6)−70 times faster than a previously designed LOCKR-based SARS-CoV-2 sensor that relies on conformational selection[41]. Thus, using this platform, a binder to almost any target can be turned into a single-component sensor that is sufficiently fast that, in most practical applications, its response time will be limited by target association rather than by a slow conformational change.

## Rapid modulation of IL-2 signalling

Finally, we investigated whether facilitated dissociation could be used to control cellular processes with high temporal resolution. In cellular signalling, the residence times of ligands on their cognate receptors is thought to modulate the signalling and cellular responses[44–46]. The central immune cytokine interleukin-2 (IL-2) activates the IL-2 receptor (IL-2Rβγ_c) by inducing heterodimerization of chains β and γ_c (ref. 47). The resulting complexes dissociate or degrade on timescales of hours[48,49], so controlling the temporal dynamics of IL-2 signalling is difficult: there is no off-switch (Fig. 5a).

We set out to construct a switchable IL-2 mimic that enables control over the lifetime of the IL-2Rβγ_c complex with a time resolution of seconds (Fig. 5b). Neo2, a previously designed IL-2 mimic, tightly binds to IL-2Rβγ_c to elicit downstream signalling[50]. Sampling a variety of γ_c positions relative to the switch from AS1, we rigidly fused Neo2 to the switch such that in state X, γ_c can bind, but in the effector-bound state Y, it would strongly clash (Fig. 5c). We identified several designs for which binding the effector markedly accelerates the dissociation of γ_c from the active signalling complex (Supplementary Fig. 13). For one of these, ASNeo2, binding the effector induces a 1,500-fold increase in the γ_c off-rate (Fig. 5d). We designed variants of ASNeo2 with different dissociation kinetics and topological safeguards against degradation (Supplementary Note 5); for one, the effector accelerated γ_c dissociation by 5,700-fold, the highest fold change of all our designed systems (Extended Data Fig. 8).

To investigate the switching capability of ASNeo2 in a physiological context, we quantified the dimerization of labelled IL-2Rβ and γ_c in the plasma membrane of live cells by single-molecule fluorescence microscopy. Dual-colour co-tracking and mobility analyses confirmed that ASNeo2 efficiently dimerizes IL-2Rβ and γ_c and that adding the effector reverses this association rapidly and completely, even at a high excess of γ_c (Fig. 5e,f, Extended Data Fig. 7c–i and Supplementary Video 1). By double labelling ASNeo2 on opposite sides, we could observe the effector-induced intramolecular conformational change at the plasma membrane using single-molecule Förster resonance energy transfer (smFRET) (Extended Data Fig. 7j,k). ASNeo2 activates signalling in human natural killer (NK) cells (YT-1 cell line), and its activity is greatly reduced in the presence of the effector (Fig. 5g). After the effector is added, STAT5 phosphorylation immediately stops accumulating and gradually decreases to a low level (Fig. 5h). The effector blocks ASNeo2 activity nearly as effectively as does ruxolitinib, a JAK1 inhibitor.

To examine how the duration of IL-2 signalling affects the downstream cellular response, we stimulated primary human T cells with ASNeo2 and induced dissociation at various time points. Whereas sustained stimulation was required for proliferation[51] (Fig. 5i), protection from apoptosis was evident after a short transient stimulation: cells treated with ASNeo2 and then with effector 5 min afterwards survived three days later at double the rate of an unstimulated control (Fig. 5j), despite the suppression of most downstream IL-2 target proteins (Extended Data Fig. 9e). This likely reflects reduced caspase-3 activity and increased *BCL2* expression after transient stimulation (Fig. 5k and Extended Data Fig. 9g).

To further investigate the dependence of IL-2 signalling on the lifetime of the signalling complexes, we compared RNA sequencing (RNA-seq) data from T cells that were treated with ASNeo2 transiently, continuously or not at all (Fig. 5l–n and Supplementary Fig. 14). Transient stimulation (through incubation with ASNeo2 followed by effector-induced dissociation) upregulated genes that are involved in suppressing apoptosis (*BCL2*, *NIBAN2* and *RNF157*), cell-cycle regulation (*CDKN2B*, *CDK6*, *CCND2* and *PIM1*), suppressing cytokine signalling (*SOCS2* and *CISH*) and immune activation (*TNFSF8*, *HAVCR2* and *PTGER2*) (Fig. 5m). On activation, T cells typically shift their energy production from oxidative phosphorylation to glycolysis[52], and we observed that transient stimulation downregulates oxidative-phosphorylation genes (Fig. 5n and Supplementary Fig. 15; glycolysis genes are not yet upregulated). Sustained stimulation activated genes associated with mTORC-driven metabolic changes and MYC- and E2F-driven cell-cycle progression (for example, *CDK4* and *POLD2*), but transient stimulation did not (Fig. 5n, Extended Data Fig. 9i and Supplementary Fig. 15), suggesting that IL-2 signalling must be sustained to pass the G1/S checkpoint. Transient stimulation did activate genes associated with formation of the mitotic spindle (for example, *DOCK2* and *KIF1B*) (Fig. 5n, Extended Data Fig. 9i and Supplementary Fig. 15), suggesting that preparations for mitosis are made immediately after T cell activation, before cell-cycle checkpoints. These results show how designed facilitated dissociation can be used to tackle unanswered questions in cell biology.

## Conclusions

We show here that by explicitly considering excited intermediate states when designing coupled protein systems, a broad range of facilitated dissociation systems can be designed. We use a switchable target binder (host) and a flexible effector that can rapidly bind to and switch the target–host complex to induce a large steric clash with the target, forming a strained excited ternary complex. By modulating the strain energy of the ternary complex, we can tune the resulting acceleration of target dissociation. Crystal structures throughout the facilitated dissociation process confirm our ability to design excited states and large register-shift conformational changes. Our designed dynamic systems show both a high dynamic range and a rapid stimulus response, demonstrating the kinetic advantage of facilitated dissociation over mutually exclusive competition.

Power-stroke mechanisms[8,31] can generate force more efficiently than can ratchet mechanisms (in which a single large step is rectified at the end of the conformational change)[37]. Induced folding underlies the kinesin power stroke[53], and our flexible effector likely also folds upon binding, accelerating the conformational transition by lowering the energy barrier: the energetic costs of uphill steps along the transition coordinate can be compensated by the formation of additional interactions with the folding peptide[35,36,54]. In contrast to this induced-fit mechanism of the flexible effector, we find that a rigid effector provides reduced rate acceleration even though it binds more tightly[38]. In kinesin and other biological systems, it is difficult to directly assess the role of flexibility and folding upon binding in overall function; by contrast, our designed model systems allow the direct comparison of flexible and rigid effectors and show that the former yield faster conformational transitions against loads.

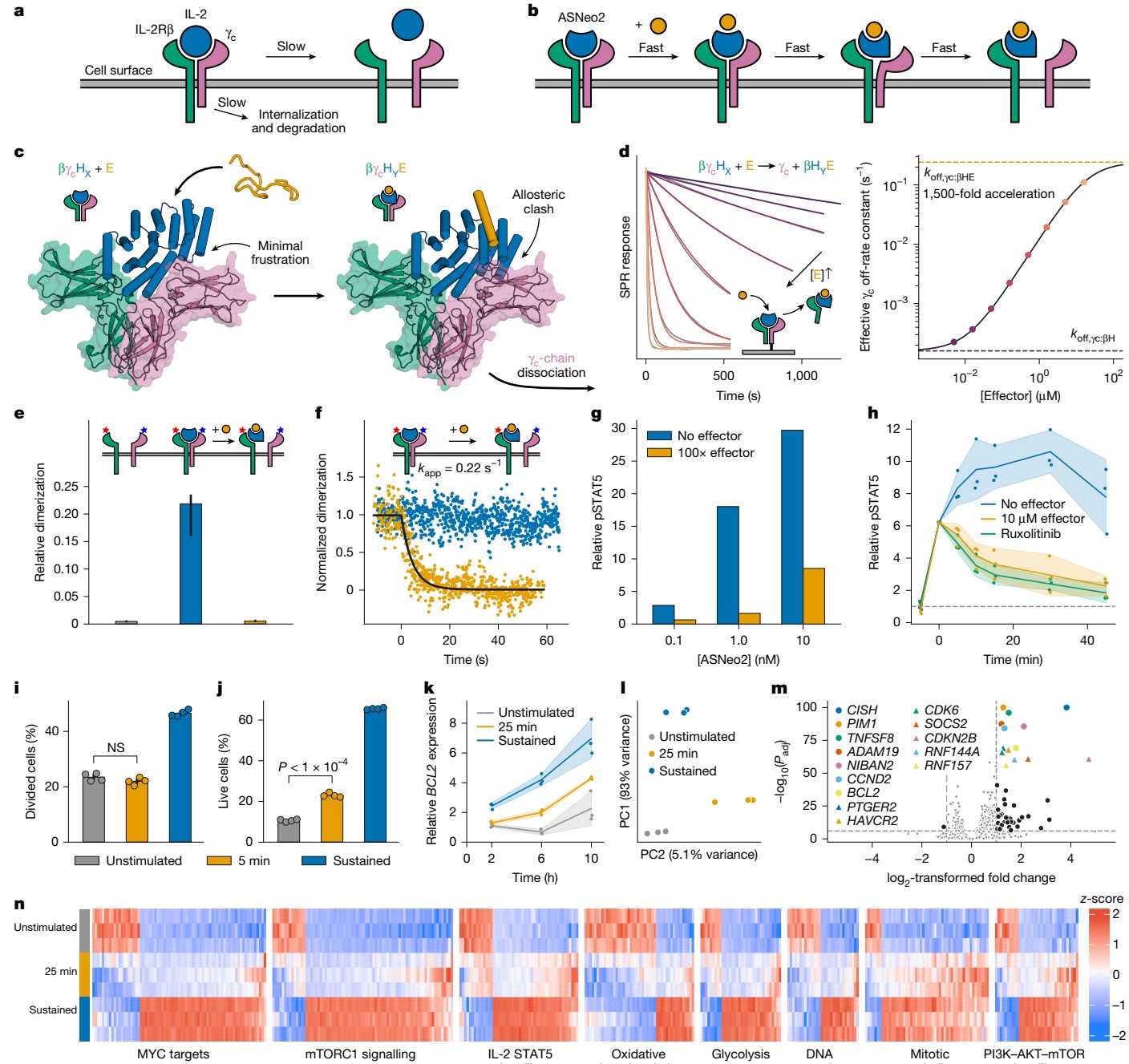

**Fig. 5 | Characterization of a rapidly switchable IL-2 mimic. a**, Natural pathways for terminating IL-2 signalling are slow. **b**, Through facilitated dissociation, signalling could be rapidly terminated. **c**, Model of ASNeo2 binding IL-2Rβγc to activate signalling (left), which quickly terminates after adding effector (right). **d**, Left, accelerated dissociation of γc; data (grey) fitted (colours) as described in the Methods. Right, γc dissociation rate constant versus effector concentration (circles) fitted with a hyperbolic (black line). **e**, Relative IL-2Rβ/γc dimerization on the cell surface at first (grey; $n = 37$), after adding ASNeo2 (blue; $n = 32$) and after subsequently adding effector (orange; $n = 33$). **f**, Time courses of IL-2Rβ/γc dimerization after pre-stimulation with ASNeo2 then adding nothing (blue) or effector (orange), fitted with an exponential (black) yielding the rate constant $k_{app}$. **g**, Dose–response of STAT5 phosphorylation from stimulation with ASNeo2 alone (blue) or with effector (orange) ($n = 1$). **h**, Time courses of STAT5 phosphorylation after stimulation

with ASNeo2 for 5 min then adding nothing (blue), effector (orange) or ruxolitinib (green) ($n = 3$). **i**–**n**, Human T cells were stimulated with ASNeo2 or left untreated as a control (grey). Signalling was sustained (blue) or terminated with effector (orange) after the indicated duration. **i,j**, Cell division (**i**, by carboxyfluorescein succinimidyl ester (CFSE) staining) and survival (**j**) three days after stimulation ($n = 4$); statistics from ANOVA with two-sided Tukey's post-test. NS, not significant. **k**, Time courses of *BCL2* expression (by quantitative PCR (qPCR); $n = 3$). **l**–**n**, RNA-seq analysis six hours after stimulation ($n = 3$). **l**, Principal component (PC) analysis. **m**, Changes in gene expression after transient stimulation. Points denote differentially expressed genes; the most significant are labelled. **n**, Heat map of differentially expressed genes from hallmark gene sets with high gene correlation. In **e,h**–**k**, lines and bars represent medians (**e**) or means (**h**–**k**); error bars and shaded areas represent 95% confidence intervals. $n$ refers to biologically independent samples.

Most known examples of facilitated dissociation couple the target and the effector through a combination of direct steric overlap[17–20,26,32] and intricate allostery[14,15,22–25]. By contrast, because our allosteric

mechanism of switching steric clashes places no specific requirements on the binder or target, our approach can be used to dynamically regulate protein–protein interactions quite generally: by fusing

to our switch, almost any binding interaction can be made to switch off rapidly in the presence of an effector. Our approach transferred immediately to rapidly switching active IL-2-like signalling complexes: we obtained several working designs on the first attempt (in which 24 designs were tested). Controlling the duration of IL-2 signalling with such switchable cytokines could enable investigation of early events in signalling or tuning of the cell response through timing-sensitive regulatory mechanisms further downstream[51] (for example, disrupting signalling complexes at the cell surface or later in the endosome[49] could be used to distinguish the cellular responses induced by signalling from each compartment[55]). Therapeutically, systemic administration of the effector and local administration of the switchable cytokine could elicit strong immune activation at only the site of administration (because any cytokine that escapes into circulation would be deactivated by the effector). More generally, our work provides a route to designing the rates and pathways of protein motion and change, which should ultimately enable the construction of complex, lifelike protein machinery.

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

## Methods

### Design of structured switch–binder fusions (hosts) allosterically coupling the target and the effector

In PyMOL, we manually positioned the switch relative to the binder–target complex subject to several constraints: there is no steric overlap between the target and switch state X; there is large steric overlap between the target and switch state Y; the smallest deformation that could be undergone by the switch and the target to resolve this clash is in the desired direction; and the switch C terminus and binder N terminus are relatively oriented such that sensible additional structure could be built between the switch and the binder to rigidly fuse them as positioned. To aid in visualizing this additional structure, we also included placeholder helices while manually positioning the switch and binder, effectively 'sketching' the fusion (Supplementary Fig. 4b).

We then refined these sketches into plausible backbones. For the initial fusions including AS0, we extracted the centre four residues of the placeholder helix, then used inpainting with RosettaFold[56] to scaffold that fragment between the switch and the binder. For later fusions, including the AS1 variants and ASNeo2 designs, we first used Rosetta FastDesign[57] to sample around the starting sketch for designable positions of the switch, binder and placeholder helices while keeping the region of the state X switch that clashes in state Y fixed relative to the target, then used RFDiffusion[7] (conditioned on the secondary structure and block adjacency of the sketches) to build structure between the sampled switch and binder positions. During both structure-generation approaches, we masked noncritical residues on the switch and binder interfacing with the fusion structure. After generation of the fusion backbone structure, we used ProteinMPNN[58] to optimize sequences for the fusion structure and the masked residues on the switch and binder. To filter designs, we used AF2[59] (with initial guess (AF2-IG)[60] for complex predictions) to predict the structure of the fusions alone, with the target, and with the effector, selecting designs for which each structural state is correctly and confidently predicted by a majority of the five model weight sets. Finally, for the AS1 variants and ASNeo2 designs, we used AF2-IG to predict the structure of the strained ternary complexes, selecting sets with a diversity of deformation directions.

### Design of induced-fit register-shift switches

Designs AS1, AS2, AS5 and AS7 were generated starting from design AS0, which contains the hinge switch cs221. When cs221 was designed, its state Y was generated by copying the N-terminal domain (helices 1–4) of its parent scaffold (DHR20; ref. 61), aligning helix 4 of the copy to the corresponding helix of DHR20 offset by three residues, and combining the transformed N-terminal domain and original C-terminal domain into a single protein[33]. Thus, relative to the C-terminal domain, the N-terminal domain is both rotated around and translated along the axis of helix 4, exposing a cleft between the domains for binding a helical peptide. Now, to generate a new state X for this switch that retains an open cleft, we repeated this procedure but aligned the N-terminal domain with a residue offset of −4 instead of 3, so that overall this position of the N-terminal domain is shifted from state Y along helix 4 by one heptad (Supplementary Fig. 4e). This introduces minimal rotation and thus maintains the open cleft. We combined this new position of the N-terminal domain with the entire C-terminal domain of AS0 (including the fusion to LHD101B), then used Rosetta FastDesign to further sample designable positions of the N-terminal domain around this starting point. In half of the design trajectories, we included a placeholder helix in the cleft to help ensure it remains open. We then used inpainting with RosettaFold to generate loops connecting the domains, then paired these complete state X backbones with the original effector-bound state Y of AS0 (also generating new loops between domains in state Y as necessary to match the loop lengths of each new state X).

To generate single sequences that support both states, we first performed multi-state design with Rosetta FastDesign (enforcing sequence symmetry between states) to further refine the paired backbones so that they are more mutually compatible, then used ProteinMPNN with residue probabilities tied at corresponding positions between states to optimize sequences simultaneously for both conformations[33]. During these sequence-design steps, the sequence of the effector and of the binder fusion was kept fixed. To filter designs, we used AF2-IG to predict the structure of the switches both with and without the effector, selecting designs for which the new state X (in the absence of effector) and state Y (in the presence of effector) are correctly and confidently predicted by a majority of the five model weight sets. To further ensure that these designs favour state X in the absence of effector and state Y in the presence of effector, we only selected designs that scored more favourably in Rosetta in state X than in state Y, but also scored more favourably in state Y with the effector bound than the sum of the scores of state X and the unbound effector.

### Design of rapid sensors

We first sought to cage the SmBiT peptide within the effector so that, through reverse facilitated dissociation, binding of the target would rapidly uncage the SmBiT to enable luciferase reconstitution. We grafted SmBiT onto the effector peptide at a range of positions, then screened these 'SmBiTgraft' variants for binding to AS0 and rapid dissociation after binding of the target. We used AS0 as the base design, reasoning that its closed state X would slow undesired peptide reassociation. For each working SmBiTgraft, we then generated a flexibly linked LgBiT–SmBiTgraft–AS0 construct. The best construct shows low luciferase activity that increases rapidly after addition of target, but only slowly after addition of effector peptide (which must compete directly with SmBiTgraft) (Extended Data Fig. 6). To demonstrate the modularity of this platform, using the methods described above for designing hosts, we rigidly fused a SARS-CoV-2 RBD binder (LCB1; ref. 43) to the switch in place of the target binder such that the RBD clashes with the switch in state Y but not in state X, and tested 16 of these AScov fusion designs.

### Recombinant expression and purification

Synthetic DNA fragments encoding each design were obtained from IDT as eBlocks and cloned into custom vectors using Golden Gate assembly[62]. Designs usually contained a C-terminal sequence-specific nickel-assisted cleavage (SNAC) tag[63] and a 6×His tag (MSG-Protein-GSGSHHWGSTHHHHHH). Proteins to be captured on the chip for SPR experiments contained an N-terminal AviTag and a C-terminal 6×His tag (MSGLNDIFEAQKIEWHESSG-Protein-GSGHHHHHH). For the size-exclusion chromatography (SEC) binding experiments shown in Supplementary Fig. 6 and for screening SmBiTgraft effector peptide variants with SPR, the effector was fused to superfolder GFP in a sfGFP-GSSG-Effector-GSHHHHHH construct. To rapidly break split luciferases, AS1 was fused to LgBiT in a MSG-AS1-linker-LgBiT-GSHHHHHH construct and the target was fused to SmBiT in a MSG-Target-linker-SmBiT-GSHHHHHH construct. In the rapid sensors, AS0 and AScov were fused to LgBiT and a SmBiT-containing effector in a MSGHHHHHHGS-LgBiT-linker-SmBiTgraft-linker-(AS0 or AScov)-GS construct. Sequences for all designs are available in Supplementary Table 4.

All proteins were expressed from NEB BL21(DE3) *E. coli* cells using TBII (MpBio) autoinduction medium with 0.5% (w/v) glycerol, 0.05% (w/v) glucose, 0.2% (w/v) lactose, 20 mM MgSO$_4$, trace metal mix and 50 µg ml$^{-1}$ kanamycin. Expression cultures (50 ml) were grown either at 37 °C for 16–20 h or at 37 °C for 6–8 h, then at 18 °C for 16–20 h. Cells were collected by centrifugation, resuspended in 5 ml lysis buffer (100 mM Tris HCl pH 8.0, 200 mM NaCl, 50 mM imidazole, 1 mM PMSF, and 1 Pierce Protease Inhibitor Mini Tablets, EDTA-free per 50 ml) and lysed by sonication. The lysate was clarified by centrifugation at 14,000$g$ for 30 min. Protein in the soluble lysate was bound to 1 ml Ni-NTA resin (QIAGEN), washed with 5 ml low-salt wash buffer (20 mM

Tris HCl pH 8.0, 200 mM NaCl and 50 mM imidazole), 5 ml high-salt wash buffer (20 mM Tris HCl pH 8.0, 1 M NaCl, 50 mM imidazole) and 5 ml low-salt wash buffer, and eluted in 1.2 ml elution buffer (20 mM Tris HCl pH 8.0, 200 mM NaCl and 500 mM imidazole) after a 0.4 ml pre-elution. The proteins were further purified by SEC on a fast protein liquid chromatography (FPLC) system with a Superdex 200 Increase 10/300 GL column in Tris-buffered saline (TBS; 20 mM Tris pH 8.0 and 100 mM NaCl) with 1-ml fractions. Where possible, fractions that probably corresponded to protein monomers were selected. Final protein concentrations were estimated using molar extinction coefficients predicted from the protein sequence and integrating the absorbance at 280 nm over the selected fractions. Correct protein molecular weights were confirmed using liquid chromatography–mass spectrometry (LC–MS).

### Peptide synthesis

The effector peptide cs221B was chemically synthesized by GenScript. The TAMRA-labelled effector used in fluorescence polarization (FP) experiments was synthesized as previously described[33].

### SEC binding assay

Individual host proteins, sfGFP–effector and 1:1 host:sfGFP–effector mixtures were prepared at 20 µM in TBS (20 mM Tris pH 8.0 and 100 mM NaCl). A 0.5-ml quantity of each solution was injected onto a Superdex 200 Increase 10/300 GL column in TBS and the absorbance at 230 nm was monitored for changes in the retention volumes of the mixture compared with the individual proteins.

### Fluorescence polarization

FP binding experiments with TAMRA-labelled effector were performed at 25 °C in TBS (20 mM Tris pH 8.0 and 100 mM NaCl) with 0.05% v/v TWEEN20 in 96-well plates (Corning 3686). Parallel and perpendicular fluorescence intensity was measured using a Synergy Neo2 plate reader with an FP 530/590 filter cube. Fluorescence polarization $P$ (in units of mP) was calculated by the following expression:

$$P = \frac{\{\text{parallel fluorescence intensity}\} - \{\text{perpendicular fluorescence intensity}\}}{\{\text{parallel fluorescence intensity}\} + \{\text{perpendicular fluorescence intensity}\}} \times 1,000.$$

For affinity measurements, host proteins were titrated by twofold serial dilution across TAMRA–effector (at a constant concentration between 0.1 and 1 nM) through 24 wells, with a final volume of 80 µl in each well. Plates were incubated for at least 12 h at room temperature to fully equilibrate before measurement. To determine affinities, the following binding isotherm function was fitted to the measured polarization values using nonlinear least-squares minimization:

$$P = P_0 + P_1 f_{\text{bound}}$$

$$f_{\text{bound}} = \frac{1}{2[E]}([H] + [E] + K_d - \sqrt{([H] + [E] + K_d)^2 - 4[H][E]}),$$

where $P$ is the modelled polarization, $P_0$ is the polarization of free effector, $P_1$ is the change in polarization after binding the host, $f_{\text{bound}}$ is the fraction of effector bound to the host, $[H]$ and $[E]$ are the total concentrations of host and effector, respectively, and $K_d$ is the affinity between the host and the effector. When the fit $K_d$ is lower than $[E]$, affinities are too strong to be accurately measured with this method, so '$K_d < [E]$' is reported.

For kinetic competition measurements, the target LHD101An1 was titrated by twofold serial dilution across the host protein (at a constant concentration of 22 nM) through five wells (a sixth well with just the host

was also included) with a final volume of 72 µl in each well. Plates were incubated for one hour to allow the host and the target to equilibrate. To each well, 8 µl of 200 nM TAMRA–effector was added and rapidly mixed using a multichannel pipette, and the measurement was started immediately afterwards. This resulted in a 20 nM final concentration of both host and effector in a final volume of 80 µl per well. The following single exponential decay function was fitted to the measured polarization time courses using nonlinear least-squares minimization:

$$P = P_0 + P_1(1 - e^{-k_{\text{app}}t}),$$

where $P$ is the modelled polarization, $P_0$ is the polarization of free effector, $P_1$ is the amplitude of the change in polarization, $k_{\text{app}}$ is the apparent rate constant and $t$ is the time after the start of the measurement.

To measure facilitated dissociation in the reverse direction, host and TAMRA–effector were incubated at a 2:1 ratio to fully saturate the effector with host, diluted to 4 nM host and 2 nM effector, and distributed across 11 wells (40 µl in each well). In nine separate wells, the target was titrated by a fourfold serial dilution with constant excess unlabelled effector (40 µM). Just excess unlabelled effector at 40 µM was prepared in the 10th separate well (to measure the base rate of effector dissociation), and buffer was prepared in the 11th (to confirm the baseline remains stable). Forty microlitres of these target + effector solutions were added to the 11 host + TAMRA–effector wells and rapidly mixed using a multichannel pipette, and the measurement was started immediately afterwards. This resulted in final concentrations of 2 nM host, 1 nM TAMRA–effector, 20 µM unlabelled effector (to make the dissociation of TAMRA–effector irreversible) and varying concentrations of target in a final volume of 80 µl per well. The following single exponential decay function was fitted to the measured polarization time courses using nonlinear least-squares minimization:

$$P = P_0 + P_1 e^{-k_{\text{app}}t}.$$

The following hyperbolic function was fitted to the apparent rate constants $k_{\text{app}}$:

$$k_{\text{app}} = \frac{k_{\text{off,TH:E}}[T]}{K_{1/2} + [T]} + k_{\text{off,H:E}},$$

where $k_{\text{off,H:E}}$ is the base effector off-rate constant, $k_{\text{off,TH:E}}$ is the off-rate constant for accelerated effector dissociation from the ternary complex, $K_{1/2}$ is the concentration at which half the rate increase from $k_{\text{off,H:E}}$ to $k_{\text{off,TH:E}}$ is reached and $[T]$ is the concentration of target present.

### SPR

Proteins to be captured on the SPR chip were expressed with an N-terminal AviTag and purified as described above, except that the proteins were biotinylated after elution from the Ni-NTA resin: to the elutions, 5 µg ml$^{-1}$ BirA (Avidity), 10 mM ATP, 10 mM Mg(OAc)$_2$ and 100 µM D-biotin were added and allowed to incubate at room temperature for at least 4 h before further purification by SEC. Successful biotinylation was confirmed using LC–MS. SPR measurements were performed at 25 °C in HBS-EP+ buffer (Cytiva) on a Biacore 8K instrument. Biotinylated proteins were immobilized on the chip using the Biotin CAPture system (Cytiva).

For measurements of target off-rate constants, biotinylated target protein was immobilized on the chip. To measure base off-rate constants $k_{\text{off,T:H}}$, host proteins at 50 nM were flowed over the chip for 60 s, then dissociation was measured for 2 h. To measure accelerated off-rate constants $k_{\text{off,T:HE}}$, pre-incubated host–effector complexes (host at 1 µM and effector at 5 µM to ensure that the host is saturated with effector) were flowed for 60 s to form the ternary complex on the chip, then dissociation was measured for 4–20 min under a constant flow of 5 µM effector. Most dissociation data were fitted with the following double

exponential decay function to account for populations of host protein with different dissociation kinetics:

$$S = S_0 + S_1 e^{-k_{app,1}(t-t_0)} + S_2 e^{-k_{app,2}(t-t_0)},$$

where $S$ is the modelled SPR response, $S_0$ is the baseline, $S_1$ and $S_2$ are amplitudes relating to the sizes of each host population, $k_{app,1}$ and $k_{app,2}$ are apparent rate constants corresponding to each host population, $t$ is the time and $t_0$ is the time at which dissociation initiates. The reported rate constant typically corresponds to the faster and higher amplitude exponential in the fit; instances in which other criteria are used to determine which rate constant corresponds to the change of interest are noted. When clearly only one host population is present (often indicated by $k_{app,1} \approx k_{app,2}$ or a large difference between $S_1$ and $S_2$ when fitting a double exponential), the following single exponential decay function was fitted instead, in which the parameters are the same as above (instances in which single exponentials were used are noted):

$$S = S_0 + S_1 e^{-k_{app}(t-t_0)}.$$

To measure the rate constant of effector association to form the ternary complex, biotinylated host protein was immobilized on the chip. The host protein was saturated with target by flowing 5 μM target over the chip for 4 min, then a varying concentration of effector and 5 μM target was flowed over the chip for 2–4 min to associate the effector, and finally 5 μM target was flowed over the chip for 4 min to monitor effector dissociation. Target was included at 5 μM (higher than its affinity to the ternary complex) throughout the experiment to ensure that the host remained saturated with target, preventing changes in target binding from convoluting the response from effector binding. Association data were fitted with the following single exponential decay function:

$$S = S_0 + S_1(1 - e^{-k_{app}(t-t_0)}),$$

where $S$ is the modelled SPR response, $S_0$ is the baseline, $S_1$ is the amplitude, $k_{app}$ is the apparent rate constant, $t$ is the time and $t_0$ is the time at which dissociation initiates.

With the peptide effector, the following linear function was fitted to the apparent rate constants:

$$k_{app} = k_{on,TH:E}[E] + k_{off,TH:E},$$

where $k_{on,TH:E}$ and $k_{off,TH:E}$ are on-rate and off-rate constants and [E] is the concentration of effector flowed over the chip.

With the 3hb effector, the following hyperbolic function was fitted to the apparent rate constants:

$$k_{app} = \frac{1}{2}(k_{switch} + k_{unswitch} + k_{on,TH:E}[E] + k_{off,TH:E})$$

$$- \frac{1}{2}\sqrt{(k_{switch} + k_{unswitch} - k_{on,TH:E}[E] - k_{off,TH:E})^2 + 4k_{unswitch}k_{on,TH:E}[E]},$$

where $k_{switch}$ is the rate constant for the $TH_X \rightarrow TH_Y$ conformational change, $k_{unswitch}$ is the rate constant for the $TH_Y \rightarrow TH_X$ conformational change, $k_{on,TH:E}$ and $k_{off,TH:E}$ are on-rate and off-rate constants and [E] is the concentration of effector flowed over the chip. This function describes the slow relaxation rate constant of binding by conformational selection[64]. The data can be fitted with only the slow relaxation because, as the conformational pre-equilibrium favours state X (that is, $k_{switch} \ll k_{unswitch}$), the effect of the fast relaxation on $k_{app}$ is minimal. During the fit, $k_{off,TH:E}$ was constrained to a low value ($<1 \times 10^{-4}$ s$^{-1}$), as observed.

To measure the effector-concentration-dependent rate constant of the full facilitated dissociation process, either biotinylated target protein or biotinylated common gamma $\gamma_c$ ectodomain (Acro Biosystems

ILG-H85E8) was immobilized on the chip. For experiments with ASNeo2 designs, the ASNeo2 hosts were pre-incubated with IL-2Rβ ectodomain (Acro Biosystems CD2-H5221) before association with $\gamma_c$. Each experiment involved multiple cycles of host association and induced dissociation under the flow of various concentrations of effector obtained by twofold serial dilution (Supplementary Fig. 8a). Throughout these cycles, a small population of host that is unresponsive to the effector ('Hn', owing to partial degradation or misfolding induced by the strain in the ternary complex) could accumulate on the chip (Supplementary Fig. 8a,b). The following system of differential equations describing the expected behaviour of the proteins on the chip (accounting for this accumulation) can be fitted to the dissociation curve of cycle $n$ (Supplementary Fig. 8c):

$$\frac{d[THE]}{dt} = -k_{off,T:HE}[THE] + k_{on,TH:E}[TH][E] - k_{off,TH:E}[THE]$$

$$\frac{d[TH]}{dt} = -k_{off,T:H}[TH] - k_{on,TH:E}[TH][E] + k_{off,TH:E}[THE]$$

$$\frac{d[THn]}{dt} = -k_{off,T:Hn}[THn],$$

with initial values computed for each cycle by

$$[TH]_{final,0} = 0$$

$$[THn]_{final,0} = 0$$

$$[T]_{final,n-1} = 1 - [TH]_{final,n-1} - [THn]_{final,n-1}$$

$$[THE]_{initial,n} = 0$$

$$[TH]_{initial,n} = [TH]_{final,n-1} + f_{responsive}[T]_{final,n-1}$$

$$[THn]_{initial,n} = [THn]_{final,n-1} + (1 - f_{responsive})[T]_{final,n-1}$$

and the modelled concentrations of each complex state on the chip is related to the SPR response by

$$S = f_n (a_{TH}[TH] + a_{THE}[THE] + a_{THn}[THn]),$$

where $S$ is the modelled SPR response, [THE], [TH] and [THn] are concentrations of complex states on the chip, $t$ is the time after the start of the dissociation cycle, [E] is the concentration of effector flowed over the chip, $k_{off,T:H}$, $k_{off,T:HE}$, $k_{on,TH:E}$, $k_{off,TH:E}$ and $k_{off,T:Hn}$ are on-rate and off-rate constants, $a_{TH}$, $a_{THE}$ and $a_{THn}$ are amplitudes relating the concentration of each state on the chip to an SPR response, $f_n$ is an amplitude fudge factor for cycle $n$ accounting for small differences in amplitude across cycles and $f_{responsive}$ is the fraction of host that is responsive to the effector. Varying all of these parameters, this model is then globally fitted to the dissociation curves of all cycles using nonlinear least-squares minimization. Note that with a sufficiently high value of $k_{on,TH:E}$, this model's dissociation kinetics are determined mainly by the target dissociation parameters $k_{off,T:H}$ and $k_{off,T:HE}$; meanwhile, the effector binding parameters $k_{on,TH:E}$ and $k_{off,TH:E}$ tend to tightly covary and cannot be accurately determined from these fits. Also note that for designs with low values for $k_{off,T:HE}$, this model may be less accurate because the assumption that [THE] = 0 at the beginning of each cycle may no longer be valid. To compute the effective rate constant of the full facilitated dissociation process for intact host, we used the following simplified system of differential equations, which no longer accounts for a small population of unresponsive host:

$$\frac{d[THE]}{dt} = -k_{off,T:HE}[THE] + k_{on,TH:E}[TH][E] - k_{off,TH:E}[THE]$$

$$\frac{d[TH]}{dt} = -k_{off,T:H}[TH] - k_{on,TH:E}[TH][E] + k_{off,TH:E}[THE]$$

$$[THE]_{initial} = 0$$

$$[TH]_{initial} = 1$$

For each effector concentration [E], this system was solved for the half-time of the target:host interaction $t_{1/2}$ using the rate parameters determined from the original model fitted to the data, and the effective rate constant of the full dissociation process $k_{eff}$ was computed from each half-time as follows:

$$[THE](t_{1/2}) + [TH](t_{1/2}) = 0.5$$

$$k_{eff} = \frac{\ln(2)}{t_{1/2}}$$

The following hyperbolic function was fitted to the effective rate constants (constrained to the values of $k_{off,T:HE}$ and $k_{off,T:H}$ obtained from the global fit):

$$k_{eff} = \frac{k_{off,T:HE}[E]}{K_{1/2} + [E]} + k_{off,T:H},$$

where $k_{off,T:H}$ is the base target off-rate constant, $k_{off,T:HE}$ is the off-rate constant for accelerated target dissociation from the ternary complex, $K_{1/2}$ is the concentration at which half the rate increase from $k_{off,T:H}$ to $k_{off,T:HE}$ is reached and [E] is the concentration of effector flowed over the chip.

The discrepancy in the 3hb $EC_{50}$ between the 3hb association experiment in Fig. 2f (right) and the target facilitated dissociation experiment in Fig. 2h (right) could result from the fusion tag that was used to fix AS1 to the surface competing with the 3hb for binding the cleft, reducing the apparent 3hb on-rate in the 3hb association experiment but not in the target facilitated dissociation experiment.

All SPR measurements in the main figures were repeated at least once, with similar results.

## Circular dichroism spectroscopy
Circular dichroism spectra were measured at 25 °C on protein samples at 0.2 mg ml⁻¹ in TBS (20 mM Tris pH 8.0, 100 mM NaCl) using a Jasco J-1500 spectrophotometer.

## X-ray crystallography
The AS1 TH and THE complexes required increased hydrophobicity to crystallize. This was accomplished by lysine methylation (for crystals AS1_TH 1, AS1_TH 2 and AS1_THE 1) or with the hydrophobic surface mutations K46L, E50W, K172W and E173Y (for crystal AS1_THE 2).

Protein was expressed from NEB BL21(DE3) *E. coli* cells using TBII autoinduction medium as above but at a larger scale: either 8 × 50 ml or 1–2 × 500 ml cultures. Cells were collected by centrifugation, resuspended in lysis buffer and lysed by sonication. The lysate was clarified by centrifugation at 14,000g for 30 min. Protein in the soluble lysate was bound to 8 ml Ni-NTA resin (QIAGEN), washed with 10 ml low-salt wash buffer, 30 ml high-salt wash buffer and 10 ml SNAC cleavage buffer (100 mM 2-(N-cyclohexylamino)ethanesulfonic acid (CHES), 100 mM acetone oxime, 100 mM NaCl and 500 mM guanidinium chloride, pH 8.6)[63] and incubated in 40 ml SNAC cleavage buffer + 2 nM NiCl₂ for 12 h at room temperature to cleave. Afterwards, the flowthrough was collected and the beads were washed with 40 ml lysis buffer (minus the protease inhibitors). The amount of cleaved protein in the flowthrough and wash was assessed with SDS–PAGE, and fractions with enough cleaved protein were concentrated and further purified using SEC on an FPLC system with either a Superdex 75 Increase 10/300 GL column or a HiLoad 20/600 Superdex 75 pg column in either TBS (20 mM Tris pH 8.0 and 100 mM NaCl) or lysine methylation buffer (50 mM HEPES pH 7.5 and 250 mM NaCl) if lysine residues in the protein were to be methylated. For protein complex cocrystallization, the purified proteins and/or chemically synthesized effector were mixed at equimolar ratios. For some samples (resulting in crystals AS1_TH 1, AS1_TH 2 and AS1_THE 1), lysine residues were methylated as previously described[65], and the reaction was quenched using SEC on an FPLC system to buffer-exchange into TBS. Finally, the samples were concentrated to crystallization levels.

Crystallization experiments were done using the sitting drop vapour diffusion method. Initial crystallization trials were set up in 200-nl drops using the 96-well plate format at 20 °C. Crystallization plates were set up using a Mosquito LCP from SPT Labtech, then imaged using UVEX microscopes and UVEX PS-256 from JAN Scientific. Diffraction-quality crystals formed in 0.2 M magnesium chloride hexahydrate, 0.1 M sodium cacodylate pH 6.5 and 50% v/v PEG 200 for AS1_H; in 0.2 M sodium chloride, 0.1 M Na/K phosphate pH 6.2 and 50% v/v PEG 200 for CS221B; in 0.1 M sodium acetate pH 5.0, 5% w/v γ-PGA (Na⁺ form, LM) and 30% v/v PEG 400 for LHD101An1; in 0.2 M magnesium chloride hexahydrate, 0.1 M Tris pH 8.5 and 25% w/v polyethylene glycol 3,350 for AS5_HE; in 0.1 M sodium acetate pH 5.0 and 20% (v/v) MPD for AS5_H; in 0.2 M 1,6-hexanediol, 0.2 M 1-butanol, 0.2 M 1,2-propanediol, 0.2 M 2-propanol, 0.2 M 1,4-butanediol, 0.2 M 1,3-propanediol, sodium HEPES, MOPS (acid) pH 7.5, 40% v/v PEG 500 MME and 20% w/v PEG 20000 for AS1_HE; in 0.1 M citric acid pH 3.5 and 2.0 M ammonium sulfate for AS1_THE 1; in 1.0 M lithium chloride, 0.1 M citrate pH 4.0 and 20% w/v PEG 6000 for AS1_THE 2; in 2.4 M sodium malonate pH 7.0 for AS1_TH 1 ($P6_1 2 2$); and in 1.8 M ammonium citrate tribasic pH 7.0 for AS1_TH 2 ($P2_1 2_1 2_1$).

Diffraction data were collected at the National Synchrotron Light Source II beamline 17-ID-1 (FMX/AMX), the Advanced Light Source beamline 821/822 or the Advanced Photon Source NECAT 24ID-C. X-ray intensities and data reduction were evaluated and integrated using XDS[66] and merged and scaled using Pointless or Aimless in the CCP4 program suite[67]. Structure determination and refinement starting phases were obtained by molecular replacement with Phaser[68], using the designed model for the structures. After molecular replacement, the models were improved using phenix.autobuild with rebuild-in-place to false and using simulated annealing. Structures were refined in PHE-NIX[69]. Model building was performed using Coot[70]. The final model was evaluated using MolProbity[71]. Data collection and refinement statistics are provided in Extended Data Tables 1–3.

## DEER spectroscopy
Spin-label modelling and distance distribution predictions were performed as previously described[33] using chiLife[72] with the off-rotamer sampling method[73]. Site pair selections were performed as previously described[33]. Host protein variants containing cysteine residues at the selected sites were purified as described above, except that 0.5 mM TCEP was included in the lysis buffer and the first two Ni-NTA resin washes, and the proteins were labelled immediately after elution from the Ni-NTA resin: to the elutions, 50 μl of 200 mM of the nitroxide spin label 1-oxyl-2,2,5,5-tetramethyl-3-pyrroline-3-methyl)methanethiosulfonate (MTSL) in dimethyl sulfoxide (DMSO) was added and allowed to incubate for at least 2 h at room temperature before further purification by SEC. Successful labelling was confirmed using LC–MS. DEER samples were prepared at 20 μM labelled host protein in deuterated solvent buffered by 20 mM Tris at pH 8.0 with 100 mM NaCl and 20% d₈-glycerol (Cambridge Isotope Laboratories) as a glassing agent. When appropriate, target and effector were added to a concentration of 100 μM each. Then, 15–30 μl of each sample was loaded into a quartz capillary

(Sutter Instrument, 1.1 mm inner diameter, 1.5 mm outer diameter) and flash-frozen with liquid nitrogen. Samples were stored at −80 °C until the DEER experiments were performed.

DEER experiments were performed as previously described[33]. An ELEXSYS E580 spectrometer (Bruker) at Q-band (around 34 GHz) with an EN5107D2 resonator (Bruker) was used for all experiments. The temperature was maintained at 50 K using a cryogen-free cooling system (ColdEdge). The four-pulse DEER sequence was used, using 60-ns Gaussian observer pulses with a full width at half maximum (FWHM) of 30 ns and a frequency near the centre of the field-swept spectrum and 150-ns sech/tanh probe pulses with centre 80 MHz above the observer frequency, 80 MHz bandwidth and a truncation parameter of 10. All shaped pulses were generated using the SpinJet arbitrary waveform generator (Bruker). Pulse shapes were calculated using PulseShape (https://github.com/stolllab/PulseShape), using both resonator compensation and transmitter nonlinearity compensation. All data were collected using a 2 ms shot repetition time, 8-step phase cycling and $\tau_1$ averaging from 400 ns to 528 ns in 16-ns steps. All other experimental parameters (including pump pulse time step, $\tau_2$ delays, number of scans and others) were chosen on a per-sample basis and are reported in Supplementary Table 3.

All DEER data were analysed using the DeerExp module of the eprTools Python package (https://github.com/mtessmer/eprTools). All data were fitted using separable nonlinear least squares[74]. The foreground signal was modelled using Tikhonov regularization with the second derivative operator. The regularization parameter was selected using generalized cross-validation. The background was modelled using a 3D-homogeneous spin distribution. An additional penalty restraining the modulation depth to be low was used to prevent the fitting of long-distance artefacts in the foreground, as was done previously[75]. Confidence intervals were estimated using bootstrap sampling with 100 samples using a fixed regularization parameter. Fit parameters such as the regularization parameter, the modulation depth and the signal-to-noise ratio are listed in Supplementary Table 3.

## MD simulations

Input files for the MD simulation were prepared with CHARMM-GUI[76,77], with AF2 structure as the initial structure. A rectangular water box of edge length 10 Å was placed around the protein. The water box contained potassium and chlorine ions of concentration 0.15 M that neutralized the protein's net charge. The ions were placed in the water box using the Monte Carlo method. To model the system, the CHARMM36m force field[78] was used. After the explicit solvent system was made, the system was minimized and equilibrated before the production MD run. All three steps were performed with the GROMACS 2020.2 MD engine[79,80].

The steepest descent method was used for energy minimization, for 5,000 steps with an energy tolerance of 1000 kJ mol$^{-1}$ nm$^{-1}$. The neighbour list was updated every ten steps with a cut-off distance of 1.2 nm. Cut-off was used to calculate Van der Waals interactions with a switch distance of 1.0 nm and a cut-off distance of 1.2 nm. The force was smoothly switched off between the switch distance and the cut-off distance. The fast smooth particle-mesh Ewald method was used to calculate electrostatics with a cut-off distance of 1.0 nm. All bonds with hydrogen atoms were treated as rigid using the linear constraint solver (LINCS) algorithm[81].

The equilibration step was performed with a leap-frog algorithm using a time step of 1 fs and a total simulation time of 125 ps. The system was propagated in the NVT ensemble. The temperature was maintained at 303.15 K using the velocity rescaling method. The solute and solvent were coupled with a time constant for coupling of 1 ps. The centre-of-mass translational velocity was removed every 100 steps to prevent the system from drifting. The cut-off schemes for Van der Waals and electrostatic interactions were the same as those used in the minimization step, except that the neighbour list was updated every

20 steps. The velocities were generated from a Maxwell distribution with a temperature of 303.15 K. The same LINCS method was used to constrain the hydrogen atoms during the equilibration step.

The production step was performed with a leap-frog algorithm using a time step of 2 fs and a total simulation time of 1 μs. The trajectories with the same initial equilibrated structure were obtained in triplicate. The system was propagated in the NPT ensemble. The temperature was maintained at 303.15 K using the velocity rescaling method. The solute and solvent were coupled with a time constant of 1 ps. Exponential relaxation pressure coupling and isotropic coupling with a time constant of 5.0 ps was used to maintain the pressure at 1.0 bar. The cut-off schemes for Van der Waals and electrostatic interactions were the same as those used in the minimization step, except that the neighbour list was updated every 20 steps and the Coulomb cut-off distance was set as 1.2 nm. The centre-of-mass velocity removal was the same as that used in the equilibration step. The same LINCS method was used to constrain the hydrogen atoms during the production step.

Analysis of the trajectories, such as RMSD and root mean square fluctuation (RMSF) calculations, was performed with MDAnalysis[82]. To simulate DEER distance distributions from these trajectories, structures along each trajectory were clustered using the Gromos clustering algorithm[83] in GROMACS, distance distributions were predicted (using chiLife[72] with the off-rotamer sampling method[73]) for the centre structure of each cluster and the resulting distributions were averaged weighted by the occupancies of their corresponding clusters.

## Chain reactions with FRET readout

Hinge protein cs201F_E249L (H2) was purified as described above, except that 0.5 mM TCEP was included in the lysis, wash and elution buffers, and the SEC buffer was phosphate-buffered saline (PBS; 20 mM sodium phosphate pH 7.0, 100 mM NaCl and 0.5 mM TCEP). To double label with dyes, 50 μM H2 was incubated with 250 μM Alexa Fluor 555 C$_2$ maleimide (donor; Thermo Fisher Scientific) and 250 μM Alexa Fluor 647 C$_2$ maleimide (acceptor; Thermo Fisher Scientific), shaking at room temperature for at least 2 h. The reaction was quenched by adding DTT to 10 mM, and the proteins were separated from excess dye by SEC in TBS (20 mM Tris pH 8.0, 100 mM NaCl).

FRET binding experiments were performed at 25 °C in TBS (20 mM Tris pH 8.0 and 100 mM NaCl) with 0.05% v/v TWEEN20 in 96-well plates (Corning 3686). Fluorescence intensity was measured using a Synergy Neo2 plate reader, exciting the donor at a 520-nm wavelength and reading acceptor emission at a 665-nm wavelength.

To measure the E2–target on-rate constant, 40 μl H2 at 10 nM was prepared in six wells. To each well, 40 μl E2–target at various concentrations was added and rapidly mixed using a multichannel pipette, and the measurement was started immediately afterwards. The following single exponential decay function was fitted to the measured FRET time courses using nonlinear least-squares minimization,

$$S = S_0 + S_1 e^{-k_{app}t},$$

where $S$ is the modelled fluorescence signal, $S_0$ is the fluorescence at equilibrium, $S_1$ is the amplitude of the change in fluorescence, $k_{app}$ is the apparent rate constant and $t$ is the time after the start of the measurement. The following linear function was fitted to the apparent rate constants:

$$k_{app} = k_{on}[E2] + k_{off},$$

where $k_{on}$ and $k_{off}$ are on-rate and off-rate constants and [E2] is the total concentration of E2–target.

To measure the accelerated transfer of E2 from AS114 to H2, 1,067 nM AS114 and 533 nM E2–target were incubated for 15 min to fully cage E2–target in AS114. A control solution of just 1,067 nM AS114 was also prepared. A 37.5-μl quantity of 42.7 nM H2 was prepared in eight wells;

37.5 µl of AS114 + E2–target was added to four wells and 37.5 µl of AS114 was added to the other four wells using a multichannel pipette, and the measurement was started immediately afterwards. Five minutes later, four solutions (5 µl buffer, 100 µM target, 16 µM effector or 100 µM target + 16 µM effector) were added to each set of four wells (each solution to a different well of each set) using a multichannel pipette, and the measurement was immediately continued for 1 h. Mixing components at these concentrations resulted in final concentrations of 500 nM AS114, 250 nM E2–target, 20 nM H2, 1 µM effector and 6 µM target. A baseline drift function of the following form was fitted to the AS114 + buffer data and subtracted from the other time courses:

$$S = \frac{S_1}{1 + e^{-k(t - t_{1/2})}} + S_0,$$

where $S$ is the modelled fluorescence signal, $S_0$ is the fluorescence at equilibrium, $S_1$ is the amplitude of the change in fluorescence, $k$ is a rate constant, $t$ is the time after the start of the measurement and $t_{1/2}$ is the time at which the fluorescence has changed by half the full amplitude. AS114 was used because, in state X, it does not clash with E2 extending past the target in E2–target.

### Rapid sensors and split enzymes with luminescence readout

Luminescence experiments were performed at 25 °C in TBS (20 mM Tris pH 8.0 and 100 mM NaCl) with 0.05% v/v TWEEN20 in 96-well plates (Corning 3686). Luminescence was measured using a Synergy Neo2 plate reader with a LUM filter cube.

To measure rapid breakage of a split luciferase, 111 pM AS1–LgBiT and 22 nM target SmBiT were incubated for one hour to load AS1 with the target and reconstitute the split luciferase. Next, 72 µl of this mixture was added to two wells, then 8 µl of 1/10-diluted Nano-Glo substrate (Promega N1130) and either 10 µM effector and 200 µM target or just 200 µM target was added to these wells using a multichannel pipette, and the measurement was started immediately afterwards. Excess target was included to fully outcompete target–SmBiT to enable measurement of the dissociation rate. Mixing components at these concentrations resulted in final concentrations of 100 pM AS1–LgBiT, 20 nM target–SmBiT, 1 µM effector, 20 µM target and 1/100-diluted Nano-Glo substrate.

To measure rapid analyte sensing, 64 µl of 12.5-pM sensor was added to eight wells, 8 µl of 1/10-diluted Nano-Glo substrate was added to these wells using a multichannel pipette and the measurement was started immediately afterwards. Five minutes later, 8 µl of various concentrations of analyte obtained by tenfold serial dilution was added using a multichannel pipette, and the measurement was immediately continued for 30–60 min. Mixing components at these concentrations resulted in final concentrations of 10 pM sensor and 1/100-diluted Nano-Glo substrate. The analytes were target, effector or SARS-CoV-2 RBD (Acro Biosystems SPD-C52H3). Figure 4e shows a time course with 800 nM SARS-CoV-2 RBD.

### Live-cell single-molecule imaging

For cell-surface labelling, receptors were N-terminally fused to suitable tags using a pSems vector including the signal sequence of Igκ (pSems leader). Common gamma chain (γc) was fused to the ALFA-tag[84] and IL-2Rβ was fused to nonfluorescent monomeric GFP (mXFP)[85]. HeLa cells (ACC 57, DSMZ) were cultured as previously described[86]. For transient transfection, cells were incubated for 4–6 h with a mixture of 150 mM NaCl, 10 µl of 1 mg ml⁻¹ polyethylenimine (PEI MAX, Polysciences 24765), 200 ng of DNA of pSems leader ALFA-tag-γc and 2,800 ng of pSems leader mXFPe1-IL-2Rβ[87]. Labelling, washing and subsequent imaging were performed after mounting the coverslips into custom-made incubation chambers with a volume of 1 ml. Cells were equilibrated in medium with fetal bovine serum (FBS) but lacking phenol red, supplemented with an oxygen scavenger and a redox-active photoprotectant

(0.5 mg ml⁻¹ glucose oxidase (Sigma-Aldrich), 0.04 mg ml⁻¹ catalase (Roche), 5% w/v glucose, 1 µM ascorbic acid and 1 µM methylviologene) to minimize photobleaching[88].

Selective cell-surface receptor labelling was achieved by using anti-GFP and anti-ALFA-tag nanobodies (NBs), which were site-specifically labelled by maleimide chemistry via a single cysteine residue at their C termini[88]. Anti-ALFA NB labelled with Cy3B (degree of labelling (DOL): 1.06) and anti-GFP NB labelled with ATTO 643 (DOL: 1.0) were added at concentrations of 3 nM each, at least 10 min before imaging. Coverslips were precoated with poly-L-lysine-graft-poly(ethylene glycol) to minimize unspecific binding of NBs and were functionalized with RGD peptide for efficient cell adhesion[89].

During the imaging experiments, ASNeo2 was used at 100 nM, and 10 µM effector was used to induce receptor dissociation.

Dual-colour imaging was performed by total internal reflection fluorescence (TIRF) microscopy using an inverted microscope (IX83, Olympus) equipped with a spectral image splitter (QuadView, Photometrics) and an EMCCD camera (iXon Ultra, Andor) as described in detail elsewhere[90]. Fluorophores were excited by sequential illumination with a 561-nm laser (2RU-VFL-P-2000-560-B1R, 2,000 mW, MPB Communications) and a 642-nm laser (2RU-VFL-P-2000-642-B1R, 2,000 mW, MPB Communications). Alternating laser excitation was achieved with a simple micro-controller (Arduino Uno) and open-source acquisition software[91] synchronizing laser shuttering with an acousto-optic tunable filter (AOTF; AA.AOTFnC-400.650-TN, AA Opto Electronic) and camera triggering. For long-term tracking experiments, 1,500 frames per channel were acquired at 40 fps. The resulting image stacks were divided into five-frame stacks and dimerization was determined for each stack. For Fig. 5f, three time courses, each normalized to its average initial relative dimerization, are overlaid for each condition. For all other tracking experiments, 200 frames per channel were acquired at 40 fps and dimerization was determined over the whole image stack.

Dual-colour single-molecule co-tracking time-lapse images were evaluated using an in-house-developed MATLAB software (SLIM-fast4C, https://zenodo.org/record/5712332)[88]. After channel registration based on calibration with fiducial markers, molecules were localized using the multi-target tracking algorithm[92]. Immobile emitters were filtered out by spatio-temporal cluster analysis[93]. Frame-by-frame co-localization within a cut-off radius of 150 nm was applied followed by tracking of co-localized emitters using the utrack algorithm[94]. Molecules co-diffusing for ten frames or more were then identified as co-localized. Relative levels of co-localization were determined on the basis of the fraction of co-localized particles relative to all localizations in the ALFA-γc channel (561 nm). Diffusion properties were determined from the pooled single trajectory using mean squared displacement analysis for all trajectories with a lifetime greater than ten frames. Diffusion constants were determined from the mean squared displacement by linear regression. Relative dimerization was estimated by:

$$\text{Relative dimerization} = \frac{\{\text{Co-localizations}\}}{\{\text{IL-2R}\beta \text{ localizations}\}}$$

FRET efficiencies were evaluated using an in-house-developed MATLAB software (provided and described in detail elsewhere as supplementary software[90]). In brief, alternating laser excitation FRET experiments provide three separated emission channels: directly excited donor $D_{D_{ex}}^{D_{em}}$ and acceptor $A_{A_{ex}}^{A_{em}}$ channels, as well as a sensitized FRET $F_{D_{ex}}^{A_{em}}$ channel. First, channels were aligned by calibration with fiducial markers. Then, after applying a single-molecule localization algorithm[95], single-molecule intensities were determined from background-subtracted images ($I_{D_{ex}}^{D_{em}}, I_{A_{ex}}^{A_{em}}, I_{D_{ex}}^{A_{em}}$). To evaluate FRET efficiencies, donor–acceptor pairs were co-localized with an optimized search radius. For these pairs, the apparent FRET efficiency was calculated by:

$$E_{\text{raw}} = \frac{I_{D_{\text{ex}}}^{A_{\text{em}}}}{I_{D_{\text{ex}}}^{D_{\text{em}}} + I_{D_{\text{ex}}}^{A_{\text{em}}}}.$$

To achieve accurate FRET efficiencies, standard further corrections were applied. These include the donor leakage coefficient, cross-talk-corrected proximity ratio and the correction factor γ (refs. 90,96).

## pSTAT5 signalling assay with human NK (YT-1) cells

Human NK (YT-1) cells were cultured in RPMI 1640 complete medium, supplemented with 10% FBS, 2 mM L-glutamine, minimum essential non-essential amino acids, sodium pyruvate, 25 mM HEPES and penicillin–streptomycin (Gibco). For the flow-cytometry-based pSTAT5 detection assay, $2–5 \times 10^5$ IL-2Rα-positive YT-1 cells were seeded in 350 µl of medium per well in a 96-well plate. The cells were stimulated with 1 nM ASNeo2 or Neo2 for 5 min at 37 °C. As a control, 50 µl of untreated YT-1 cells were set aside at the start of each experiment and evaluated after 45 min alongside the treated cells. After stimulation, all cells were transferred to three separate wells containing a control (no treatment), 10 µM effector or 40 µM ruxolitinib. One-seventh of the cells (50 µl) were resuspended in 17 µl of 16% paraformaldehyde (PFA) for immediate fixation. This process was repeated at 5-, 10-, 15-, 30- and 45-min intervals. After all of the time points were fixed, the cells were refixed in 4% PFA for 15 min at room temperature. After fixation, the cells were washed once with PBS containing 0.5% bovine serum albumin (BSA) (PBSA) and permeabilized with 100% methanol for 45 min at 4 °C. After permeabilization, the cells were washed twice with PBSA and stained for one hour at room temperature using Alexa Fluor 647-conjugated phospho-STAT5 (Tyr694) rabbit monoclonal antibody (Cell Signaling Technology, 9365, clone C71E5) diluted 1:100. After three washing steps, the cells were analysed using a CytoFlex S flow cytometer (Beckman Coulter). Data were analysed with CytExpert software, and cells were gated on SSC-A versus FSC-A. Each experiment was done in triplicate, and the results were analysed accordingly.

For Fig. 5g, pSTAT5 signal is normalized to the background level. For Fig. 5h, $n = 3$ time courses were normalized to the signal at time 0 (just before effector addition) and a baseline signal was determined by the average of this normalized pSTAT5 of the untreated cells, then each time course was renormalized to this baseline signal. Some batches of cells were unresponsive to stimulation; data from these batches were excluded from analysis.

## Cell-line sources

HeLa cells for single-molecule imaging were sourced from the DSMZ (ACC 57); YT-1 (CD25⁺) cells were sourced from ATCC. Cell lines were authenticated by short tandem repeat profiling. HeLa cells tested negative for mycoplasma contamination by PCR.

## Activation and stimulation of human T cells

Primary human peripheral blood mononuclear cells (PBMCs) isolated from a healthy donor by leukapheresis were thawed and resuspended at a cell density of $2 \times 10^6$ cells per ml in T cell medium, which contained RPMI 1640 (Gibco), FBS (10% v/v, Gibco, Thermo Fisher Scientific), HEPES (25 mM, Gibco, Thermo Fisher Scientific), penicillin–streptomycin (1% v/v, Gibco, Thermo Fisher Scientific), sodium pyruvate (1% v/v, Gibco, Thermo Fisher Scientific), MEM non-essential amino acids solution (1% v/v, Gibco, Thermo Fisher Scientific) and 2-mercaptoethanol (0.1% v/v, Gibco, Thermo Fisher Scientific), and supplemented with human IL-2 (100 U ml⁻¹). For human T cell activation, cells were activated with plate-bound anti-human CD3ε (1 µg ml⁻¹, clone OKT-3, BioXCell) and soluble anti-human CD28 (5 µg ml⁻¹, clone 9.3, BioXCell) for 48 h. Cells were then expanded in T cell medium with human IL-2 (100 U ml⁻¹) for eight days. On day 9, IL-2 was withdrawn for 36 h, after which cells were resuspended at $1 \times 10^6$ cells per ml and

stimulated with ASNeo2 (1 nM or 5 nM) for either 5 or 25 min, as indicated. Effector peptide (10 µM) was then added to terminate signalling. After 48 or 72 h, cells were collected for counting and phenotypic analysis by flow cytometry.

For Fig. 5i,j, cells were stimulated with 1 nM ASNeo2. For Fig. 5k–n, cells were stimulated with 5 nM ASNeo2.

## Flow-cytometry analyses of human T cells

For surface-marker staining, cells were collected into U-bottom 96-well plates (Thermo Fisher Scientific), blocked with Human TruStain FcX (BioLegend) and incubated with the indicated antibodies at 4 °C for 20 min, followed by live–dead staining by 4′,6-diamidino-2-phenylindole (DAPI; Thermo Fisher Scientific). Cells were then washed and resuspended with FACS buffer (PBS containing 0.2% BSA; Sigma-Aldrich) for flow-cytometry analyses. For phospho-STAT staining, primary human T cells were rested in T cell medium lacking IL-2 for 24 h before signalling assays. Cells were plated in a 96-well round-bottom plate in 50 µl T cell medium. Cells were stimulated with 50 µl ASNeo2 for 5 min or 25 min, followed by the addition of effector peptide and incubation for another 20 min at 37 °C, and the reaction was terminated by fixation with 1.5% PFA for 15 min at room temperature with agitation. Cells were washed and permeabilized with ice-cold 100% methanol for 60 min on ice. Afterward, cells were washed with FACS buffer before staining with pSTAT5 antibodies for one hour at 4 °C in the dark. Cells were washed and resuspended in FACS buffer for flow-cytometry analyses. For caspase-3 staining, cells were first stained for surface markers and Zombie Violet Fixable Dye (BioLegend), followed by staining with a FITC Active Caspase-3 Apoptosis Kit (BD Biosciences) according to the manufacturer's protocol. For transcription factor staining, cells were first stained for surface markers and Zombie Violet Fixable Dye, then fixed and permeabilized using a Foxp3/Transcription Factor Staining Buffer Set (eBioscience) as per the manufacturer's instructions. Cells were subsequently incubated with the indicated antibodies for intracellular staining. Detection was performed using a CytoFlex (Beckman Coulter), and data were analysed with FlowJo (v.10.10.0).

## Antibodies and reagents for flow cytometry of human T cells

The following antibodies or staining reagents were purchased from BioLegend: human CD3 (OKT3, 317324), human GATA3 (W19195B, 386906), human CD69 (FN50, 310932), human CD25 (BC96, 302611), human BCL2 (100, 658708), human Ki-67 (Ki-67, 350526), Human TruStain FcX (422302) and Zombie Violet Fixable Viability Kit (423114). The following antibodies or staining reagents were purchased from BD Biosciences: pSTAT5 (47/Stat5, 612599), and BD Pharmingen FITC Active Caspase-3 Apoptosis Kit (571606). DAPI was purchased from Thermo Fisher Scientific.

Antibodies were diluted 1:200 for surface markers (CD3, CD28, CD69 and CD25), 1:100 for intracellular proteins (GATA3, Ki-67 and BCL2) and 1:50 for pSTAT and caspase-3 staining.

Antibodies were validated by the manufacturers using flow cytometry.

## qPCR

Human T cells on day 9 after activation were subjected to IL-2 starvation for 36 h. The unstimulated control cells were maintained in culture without IL-2. For transient IL-2 signalling, cells were stimulated with ASNeo2 (5 nM) for 25 min, followed by the addition of effector peptide (10 µM) to terminate signalling. For sustained signalling, ASNeo2 (5 nM) was continuously maintained in the culture medium. Total RNA was extracted using the Quick-RNA Miniprep Kit (Zymo Research) following the manufacturer's instructions. cDNA was synthesized using the High-Capacity cDNA Reverse Transcription Kit (Thermo Fisher Scientific) with random primers. qPCR was performed using TaqMan master mix (Thermo Fisher Scientific) on a StepOnePlus Real-Time PCR System. *BCL2* expression was measured using the TaqMan BCL2 assay,

and gene expression was normalized to *GAPDH* (Thermo Fisher Scientific). Relative expression levels were calculated using the $\Delta\Delta Ct$ method.

## RNA-seq sample preparation and data analysis

Human T cells on day 8 after activation were subjected to IL-2 starvation for 36 h. The unstimulated control cells were maintained in culture without IL-2. For transient IL-2 signalling, cells were stimulated with ASNeo2 (5 nM) for 25 min, followed by the addition of effector peptide (10 μM) to terminate signalling. For sustained signalling, ASNeo2 (5 nM) was continuously maintained in the culture medium. After six hours, cells were collected, and total RNA was extracted using the Quick-RNA Miniprep Kit (Zymo Research). RNA libraries were prepared using a poly(A) enrichment-based mRNA library preparation kit following the manufacturer's instructions. Libraries were pooled and sequenced on the NovaSeq X Plus Series (PE150). Reads were aligned to the reference genome (GRCh38) using Rsubread, and gene expression was quantified with featureCounts. For analysing pathway enrichment, gene set co-regulation analysis (GESECA) was performed with hallmark gene sets from the Human Molecular Signatures Database (MSigDB). To reduce redundancy, we performed hierarchical clustering of hallmark gene sets with Jaccard's distance to yield eight gene-set clusters with minimal gene overlaps. Only one gene set with the lowest false discovery rate (FDR)-adjusted *P* value per cluster is shown in Fig. 5. Differential expression analysis was performed using DESeq2, with FDR-adjusted $P < 0.05$ as a threshold for differential expression. Heat maps were created using normalized counts of genes identified as differentially upregulated or downregulated in both sustained versus unstimulated and transient versus unstimulated comparisons.

## Statistics and reproducibility

The main-text SPR, circular dichroism, FP, FRET and luminescence experiments, and also a subset of the DEER and SEC experiments, were performed twice (SPR in Figs. 2d,h and 4b; FRET in Fig. 4c; and SEC in Supplementary Fig. 9a) or three times (SPR in Fig. 2a,e–g; FP in Fig. 4b and Supplementary Figs. 5c and 12; luminescence in Fig. 4d,e; SEC in Supplementary Fig. 6b; and DEER with AS1 in Extended Data Fig. 2) to ensure reproducibility and low variance, and one representative experiment was reported owing to low variance among replicates. Having established low variance for these experiments, some similar measurements reported in the Supplementary Information were performed once. For all microscopy, cell staining, qPCR and RNA-seq experiments, three or four independent biological replicates were performed unless otherwise noted in the figure legend. All attempted replications of all experiments were successful, with the exception that some batches of YT-1 cells were unresponsive to stimulation; the findings in Fig. 5h were not observed with those batches. Allocation was random. No sample size calculations were performed; rather the sample sizes were chosen based on experience and were sufficient for the important observed differences between groups to be strongly statistically significant.

## Reporting summary

Further information on research design is available in the Nature Portfolio Reporting Summary linked to this article.

## Data availability

All data generated during this study are included either in the main text or as Supplementary Information. Sequences for all designs are available in Supplementary Table 4. Atomic coordinates and structure factors for all crystal structures reported in this paper have been deposited in the Protein Data Bank (PDB; https://www.rcsb.org) with accession codes 9DCX, 9DCY, 9DCZ, 9DD0, 9DD1, 9DD2, 9DD3, 9DD4, 9DD5 and 9OLQ. PDB models and sequences for all designs and source data with analysis scripts have been deposited at Zenodo under

https://doi.org/10.5281/zenodo.16749448 (ref. 97). Single-molecule tracking data have been deposited at Zenodo: calibration beads, unstimulated samples, long-term measurements and labelled ligand experiments under https://doi.org/10.5281/zenodo.13957447 (ref. 98), Neo2 and Neo2 + effector under https://doi.org/10.5281/zenodo.13957498 (ref. 99) and ASNeo2 and ASNeo2 + effector, as well as smFRET data, under https://doi.org/10.5281/zenodo.13957540 (ref. 100). The raw RNA-seq data have been deposited in the NCBI Sequence Read Archive (SRA) under BioProject accession code PRJNA1302552. The SKEMPI database can be accessed at https://life.bsc.es/pid/skempi2/database/index. The reference genome GRCh38 (accession code GCF_000001405.26) can be accessed at https://hgdownload.soe.ucsc.edu/goldenPath/hg38/bigZips/. Hallmark gene sets can be accessed from MSigDB at https://www.gsea-msigdb.org/gsea/msigdb/collections.jsp.

## Code availability

Code used to generate designs and analysis scripts used to generate all figures has been deposited at Zenodo (design code: https://doi.org/10.5281/zenodo.16749263 (ref. 101); data analysis: https://doi.org/10.5281/zenodo.16749448 (ref. 97)).

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

**Acknowledgements** We thank P. J. Y. Leung, K. L. Shelley, A. Pillai, C. Demakis, M. Exposit, K. Thompson, C. Savvides, R. J. Ragotte, G. Ahn and M. Glögl for discussions and technical support; K. VanWormer and L. Goldschmidt for technical support; S. R. Gerben and A. Murray for protein production support; and X. Li, M. Lamb, Z. Taylor and V. Adebomi for LC–MS support. This work was supported by the Audacious Project at the Institute for Protein Design (A.J.B., A.K., J.D.L.C., E.B. and A.K.B.); by a gift from Microsoft (A.J.B.); by the Nordstrom Barrier Institute for Protein Design Directors Fund (M.H.A. and F.P.); by Bill and Melinda Gates Foundation OPP1156262 (A.K. and J.D.L.C.); by the Open Philanthropy Project Improving Protein Design Fund (E.B. and A.K.B.); by the National Institutes of Health (NIH) National Institute of Allergy and Infectious Disease grant R0AI160052 (A.K.B.); by CRI Irvington Postdoctoral Fellowship 315511 (Y.Z.); by National Cancer Institute K00 award 4K00CA274708 (M.O.); by National Science Foundation grant MCB 2119837 and NIH grant GM115805 (W.H.R. and D.M.Z.); by NIH grant GM151956 (S.S.); by NIH AI-51321 (K.C.G.); by the DFG grants PI 405/15 and SFB 1557 (C.P. and J.P.); and by the Howard Hughes Medical Institute (A.K.B., K.C.G. and D.B.). The EPR spectrometer used for the DEER experiments was in part supported by NIH grant S10OD021557. This research used resources (FMX/AMX) of the National Synchrotron Light Source II, a US Department of Energy (DoE) Office of Science User Facility operated for the DOE Office of Science by Brookhaven National Laboratory under contract DE-SC0012704. The Center for BioMolecular Structure (CBMS) is supported mainly by the NIH National Institute of General Medical Sciences (NIGMS) through a Center Core P30 Grant (P30GM133893), and by the DoE Office of Biological and Environmental Research (KP1607011). This work is based on research performed at the Northeastern Collaborative Access Team beamlines, which are funded by the NIGMS (P30 GM124165). The research used resources of the Advanced Photon Source, a US DoE Office of Science User Facility operated for the DoE Office of Science by Argonne National Laboratory under contract DE-AC02-06CH11357. The Berkeley Center for Structural Biology is supported by the NIH, NIGMS and the Howard Hughes Medical Institute. The Advanced Light Source is supported by the Director, Office of Science, Office of Basic Energy Sciences and US DoE (DE-AC02-05CH11231).

**Author contributions** A.J.B. developed the facilitated dissociation design concept. A.J.B. developed the computational design pipeline, designed the allosteric switches and characterized their binding kinetics with SPR, FP, FRET, luminescence and SEC experiments, with mentorship from F.P. C.P. performed and analysed single-molecule localization microscopy and NK cell signalling activity experiments. Y.Z. performed and analysed transient stimulation experiments on primary T cells. M.A.L. and A.J.B. designed and characterized the rapid sensors. M.D.J. and M.H.T. performed and analysed DEER experiments. W.H.R. and A.J.B. performed and analysed MD simulations. M.O. and A.J.B. analysed RNA-seq data. A.J.B. and M.H.A. performed initial signalling activity experiments. D.D.S. and A.A. performed site-saturation mutagenesis on LHD101. A.K., A.J.B., J.D.L.C., E.B., B.S. and A.K.B. determined and analysed crystal structures. D.B., J.P., F.P., K.C.G., S.S. and D.M.Z. supervised research. A.J.B. and D.B. wrote the manuscript. F.P., C.P., M.A.L., Y.Z., M.H.T. and W.H.R. contributed to the manuscript. F.P. and M.A.L. helped edit the manuscript. All authors read and commented on the manuscript.

**Competing interests** A.J.B., F.P., A.K.B. and D.B. are in the process of filing a provisional patent application that incorporates discoveries described in this article. The remaining authors declare no competing interests.

**Additional information**
**Correspondence and requests for materials** should be addressed to Adam J. Broerman, Florian Praetorius or David Baker.

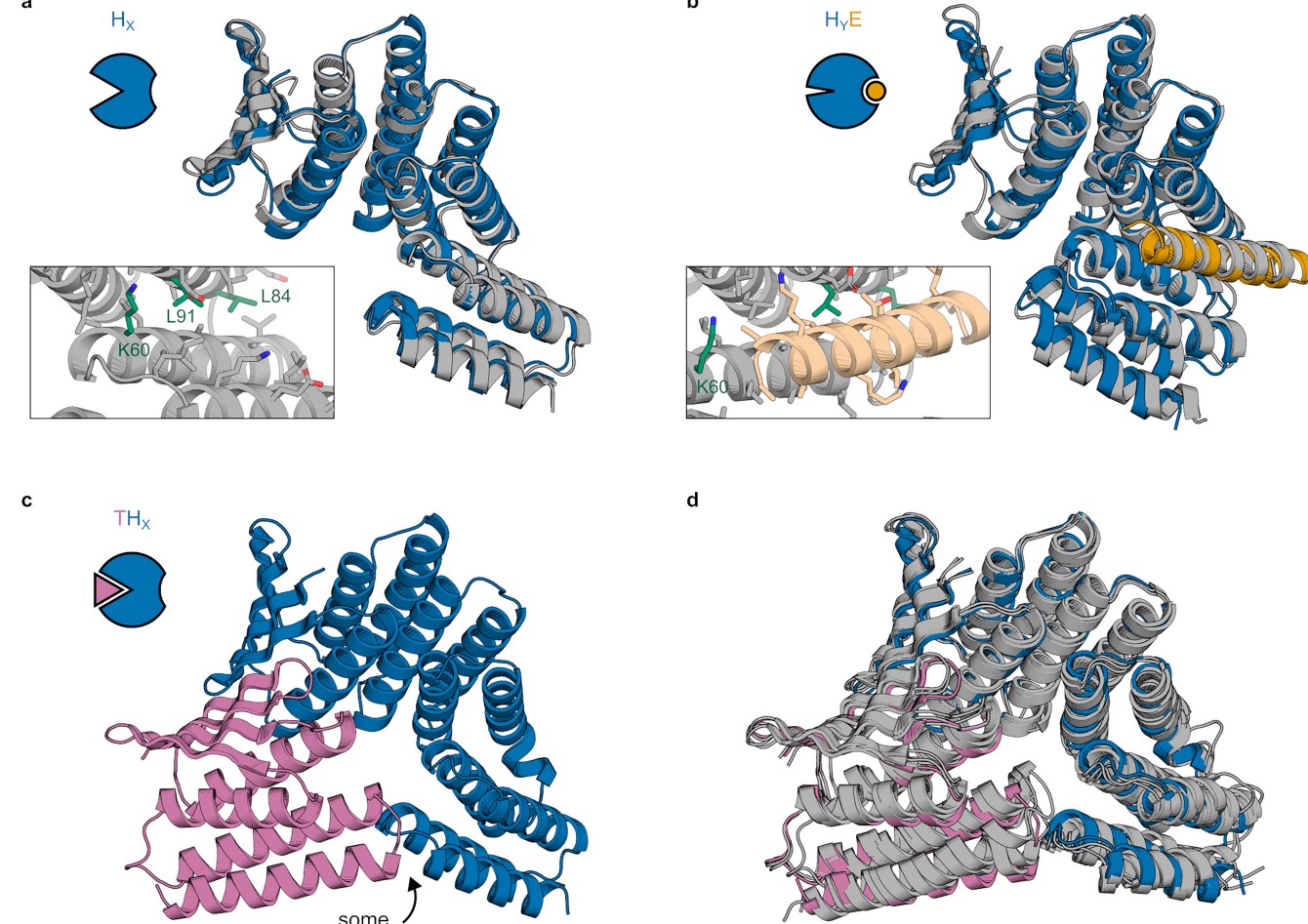

**Extended Data Fig. 1 | Structural characterization of AS5 and structural frustration of AS1 state TH$_X$. a**, Crystal structure of AS5 alone (grey) overlaid with the design model of AS5 in state X (blue). Inset shows a detailed view of side chains in the partially open effector-binding cleft. **b**, Cocrystal structure of AS5 and peptide effector (grey) overlaid with the design model of the AS5–effector complex in state Y (AS5 in blue, effector in orange). Inset shows the same view of the side chains in the effector-binding cleft as in **a**. **c**, Design model of AS1 in state X (blue) aligned to the target (pink), showing a minor clash. **d**, Three cocrystal structures of AS1 (with intact cleft) and target with methylated lysines (grey) overlaid with the AF2 model of the target–AS1 complex in state X (AS1 in blue, target in pink), showing fluctuation in the target binding conformation.

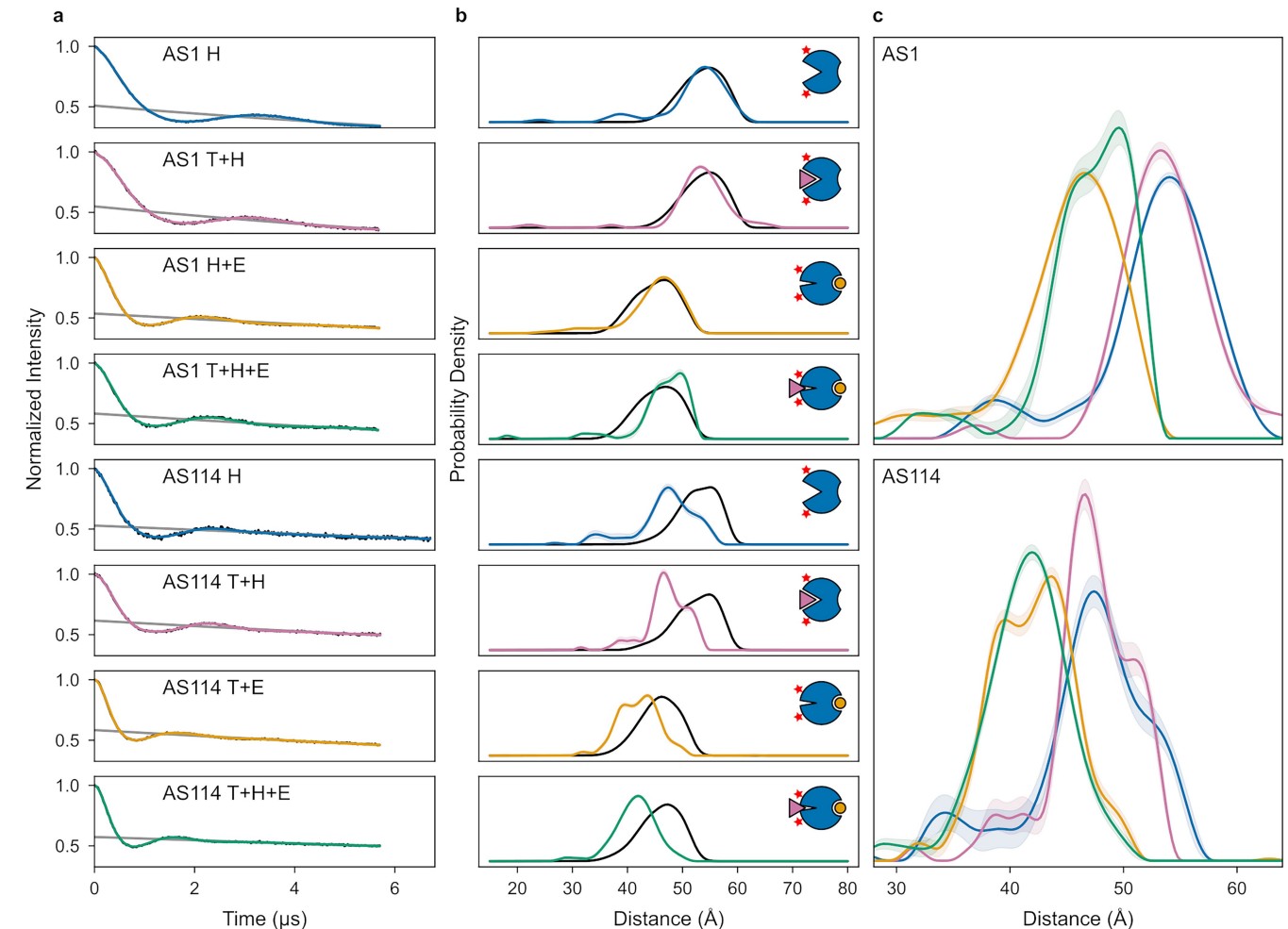

**Extended Data Fig. 2 | DEER characterization of AS1 and AS114. a**, Raw DEER traces (black), foreground fits (colours), and background fits (grey) for AS1 and AS114 with all combinations of target and effector. Experiments on complexes included the target, effector, or both in excess over the host at concentrations higher than required to fully form the complex (Supplementary Fig. 9), and spin labels were placed far from the target and effector binding sites. Thus, changes in the DEER distance distributions with different combinations of target and effector should only reflect changes in host conformation. **b**, Distance distributions from experiment (colours) and simulated from the structural state represented by the cartoons (black). For AS1, the simulated and experimental distance distributions agree well, further validating that each state adopts its

designed conformation. For AS114, the simulations consistently overestimate the experimental distribution by ~5 Å, but the shift in the distance distributions with the effector compared to those without validates the designed conformational change. **c**, Experimental distance distributions of all states, coloured corresponding to **b**. For both AS1 and AS114, the ternary complex distribution (green) aligns with the host–effector complex distribution (orange) and not with the host alone (blue) or target–host complex (pink) distributions, confirming that the ternary complex is primarily in state Y. **b,c**, Lines represent the distance distribution which best fits the time domain data; shaded regions represent 95% confidence intervals from bootstrapping (Methods).

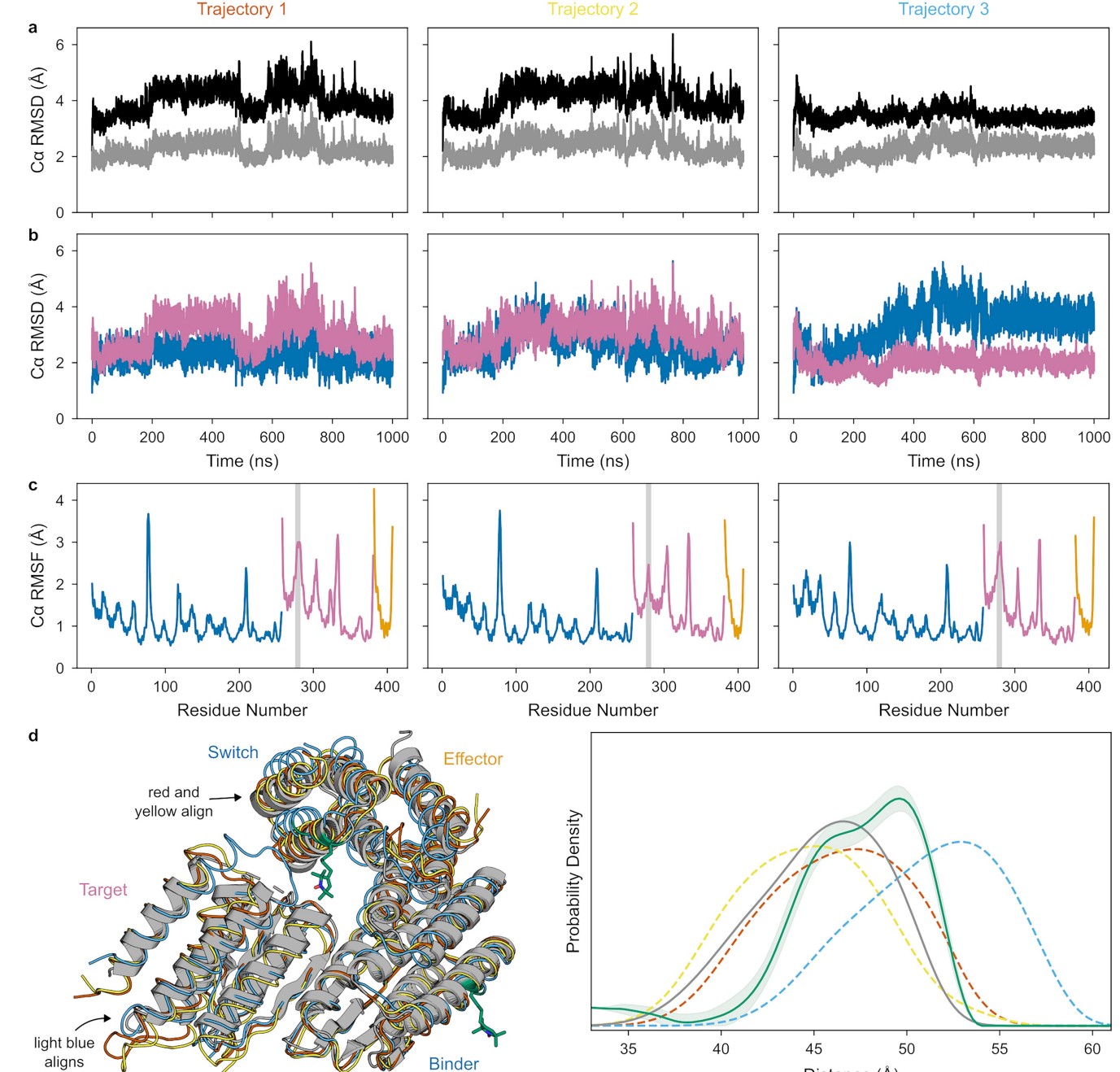

**Extended Data Fig. 3 | MD analysis of the AS1 ternary complex. a**, The Cα RMSD of the MD trajectories from the crystal structure (grey) is lower than from the aligned clashing design models of target and host–effector complex (black), showing that the MD simulations strain away from the clashing state in a manner similar to the crystal structure. **b**, Cα RMSD of the switch (blue) and target (pink) from their position in the crystal structure when the entire structures are aligned. Compared to trajectories 1 and 2, trajectory 3 shows reduced target deformation and increased switch deformation, showing that these trajectories differ in where they localize strain to resolve the clash. **c**, Per-residue Cα RMSF of the host (blue), target (pink), and effector (orange) in the ternary complex computed from each trajectory. The clashing region of the target (highlighted in grey) shows considerable flexibility, according with this region being disordered in the crystal structure. **d**, Comparison of MD simulations to experimental data. Left, crystal conformation of the ternary complex (grey) aligned to representative conformations from each MD trajectory (red, yellow, and light blue). DEER spin-label positions are shown in green. In the crystal structure, the clashing region on the target is disordered.

In the MD simulations, although flexible, this region remains mostly ordered, causing additional deformation compared to the crystal structure. Illustrating the differences in strain localization among trajectories shown in **b**, in the first two trajectories (red and yellow), the switch conformation aligns with the crystal structure and the target deforms more; in the third trajectory (light blue), the target conformation aligns with the crystal structure and the switch deforms instead. Right, experimentally measured DEER distance distribution of the ternary complex (green line representing the best fit to the time domain data and shaded region representing 95% confidence interval from bootstrapping (Methods)) and distance distributions simulated from the crystal structure (grey line) or MD trajectories (dashed lines, colours correspond to the conformations shown at left). The distance distribution simulated from the crystal structure aligns with the left peak in the experimental distance distribution, whereas the distance distributions simulated from the MD trajectories span the experimental distance distribution, suggesting that these trajectories more fully sample the space of ternary complex dynamics.

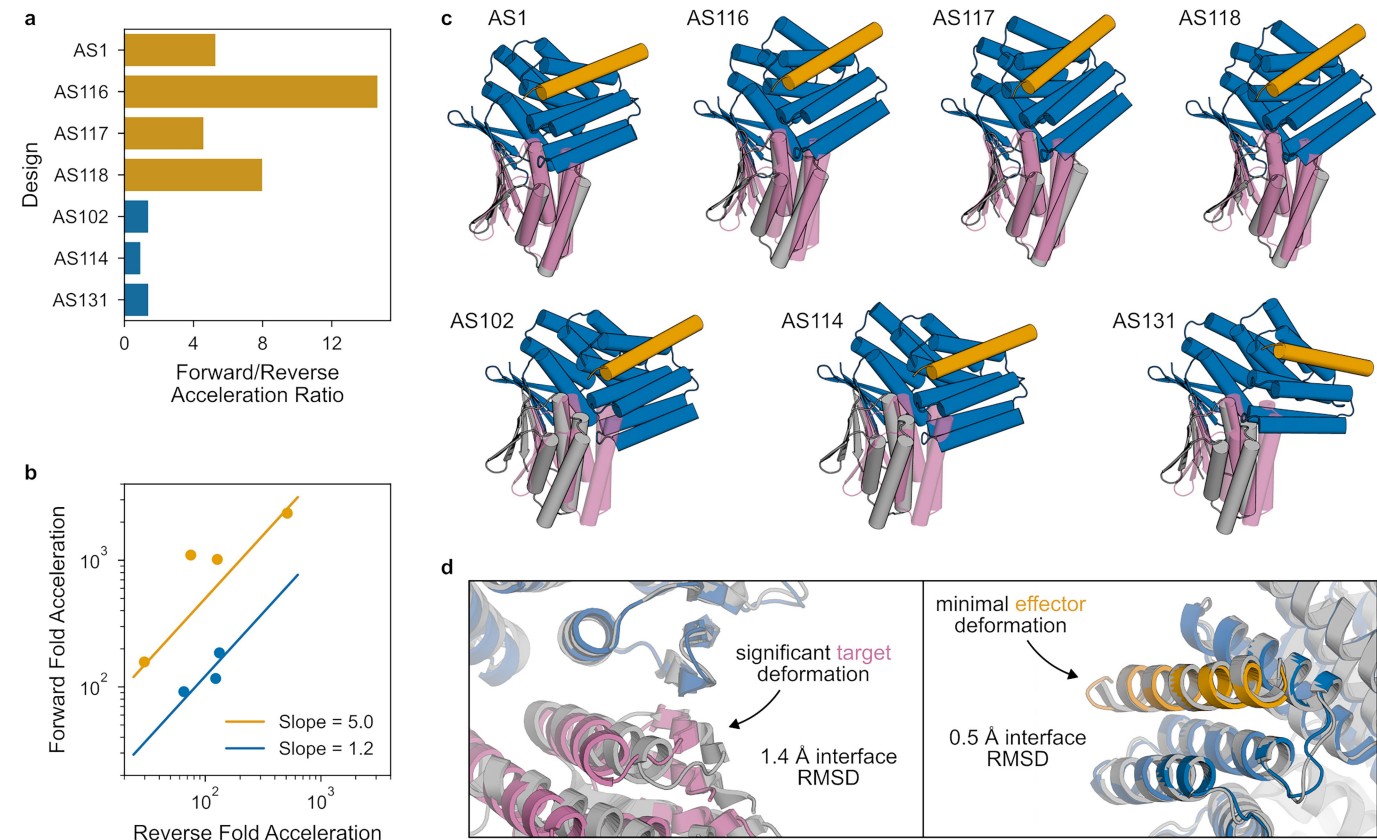

**Extended Data Fig. 4 | Nonuniform distribution of strain in the ternary complex. a**, Ratios of fold accelerations in the forward direction to those in the reverse direction, given by the equation: acceleration ratio = $(k_{off,T:HE}/k_{off,T:H})$ ÷ $(k_{off,TH:E}/k_{off,H:E})$. Acceleration ratios corresponding to designs with asymmetric distributions of strain are coloured orange, and to symmetric blue. **b**, Plot of forward vs reverse acceleration ratios, with linear fits for the symmetric and asymmetric groups. Note that on a log-log plot, the slope of a straight line passing through the origin becomes the y-intercept. **c**, For each design, design model of host–effector complex in state Y (blue and orange) aligned to the target (pink) to show the designed allosteric clash, and (grey) AF2 prediction of the target position relative to the switch in the ternary complex to show how the clash resolves through global strain. The target deforms downward in designs with an asymmetric distribution of strain, shearing the beta sheet, whereas it deforms outward in designs with a more symmetric distribution of strain, bending the beta sheet. **d**, Comparison of the AS1 binder–target interface (left) and switch–effector interface (right) (aligned at the host side) in the binary (colours) and ternary (grey) complex conformations, all from crystal structures, showing the binder–target interface deforms substantially more than the switch–effector interface in the ternary complex.

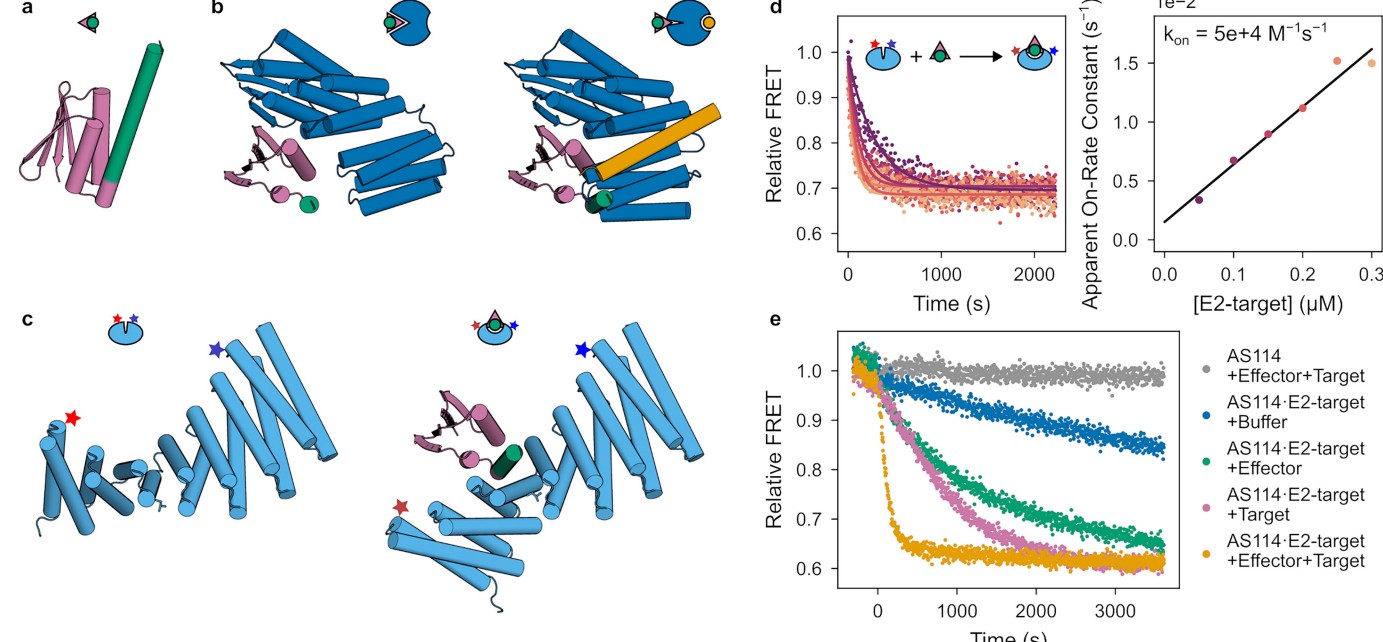

**Extended Data Fig. 5 | Construction and characterization of the chain reaction. a**, Design model of E2–target, comprising the target LHD101A (with mutations R43V and V69Q) fused to the effector peptide "E2" (cs201B) for hinge cs201. E2 is coloured green and LHD101A is coloured pink. **b**, Design models of E2–target (green/pink) bound to AS114 (blue) in state X showing no clash (left) and in state Y with the effector peptide (orange) showing a strong clash (right). **c**, Design models of the reporter hinge "H2" (cs201F with mutation E249L (sticks) which increases E2 on-rate and labelled with Alexa Fluors 555 and 647 at positions indicated by stars) in state X (left) and in state Y with E2–target (right). AS114 and H2 would overlap considerably if simultaneously bound to E2–target, so their binding should be mutually exclusive: AS114 should cage E2 until its release by the effector. **d**, Kinetics of E2–target and H2 association, measured by a change in FRET efficiency due to the conformational change in H2 upon binding. (Left) FRET time courses (normalized to the initial signal) with varying concentrations of E2–target and 5 nM H2; data (circles) fit with single exponentials (lines). (Right) apparent on-rate constants plotted against E2–target concentration (circles) and a linear fit. The value of the association rate constant ($5e + 4 M^{-1}s^{-1}$) is higher than the reported value ($4.5e + 3 M^{-1}s^{-1}$) for the original hinge cs201F with effector cs201B, suggesting that mutation E249L on H2 biases its conformational pre-equilibrium toward state Y to increase the apparent association rate. **e**, Additional data for the kinetically governed chain reaction shown in Fig. 4c. In the grey control time course, 500 nM AS114 was added to 20 nM H2, then 1 µM effector and 6 µM target was added at time 0, showing that none of these components bind to H2 to cause a change in FRET signal. In the other time courses, pre-incubated 500 nM AS114 and 250 nM E2–target was added to 20 nM H2, then buffer (blue), 1 µM effector (green), 6 µM target (pink), or both (orange) were added at time 0. A baseline drift (obtained from 500 nM AS114 after adding 20 nM H2 then at time 0 adding buffer) was subtracted from each time course, and time courses were normalized to the initial signal. The chain reaction proceeds faster when just excess target is added, probably due to blocking rebinding of E2–target to AS114 after transient dissociation[39], but this effect is insufficient to achieve full acceleration. The chain reaction also proceeds faster when just effector is added, but probably due to transient rebinding of E2–target to re-form the strained ternary complex, this also does not achieve full acceleration. Adding both effector (to accelerate E2–target dissociation from AS114) and excess target (to prevent E2–target rebinding to AS114) is required to fully accelerate the chain reaction. Note that if a single-chain effector is desired to fully accelerate the chain reaction, the effector and target could be flexibly fused into a single construct. Such a multivalent effector would be reminiscent of CITED2, whose multivalency enables rapid and unidirectional competition against HIF-1α (ref. 102).

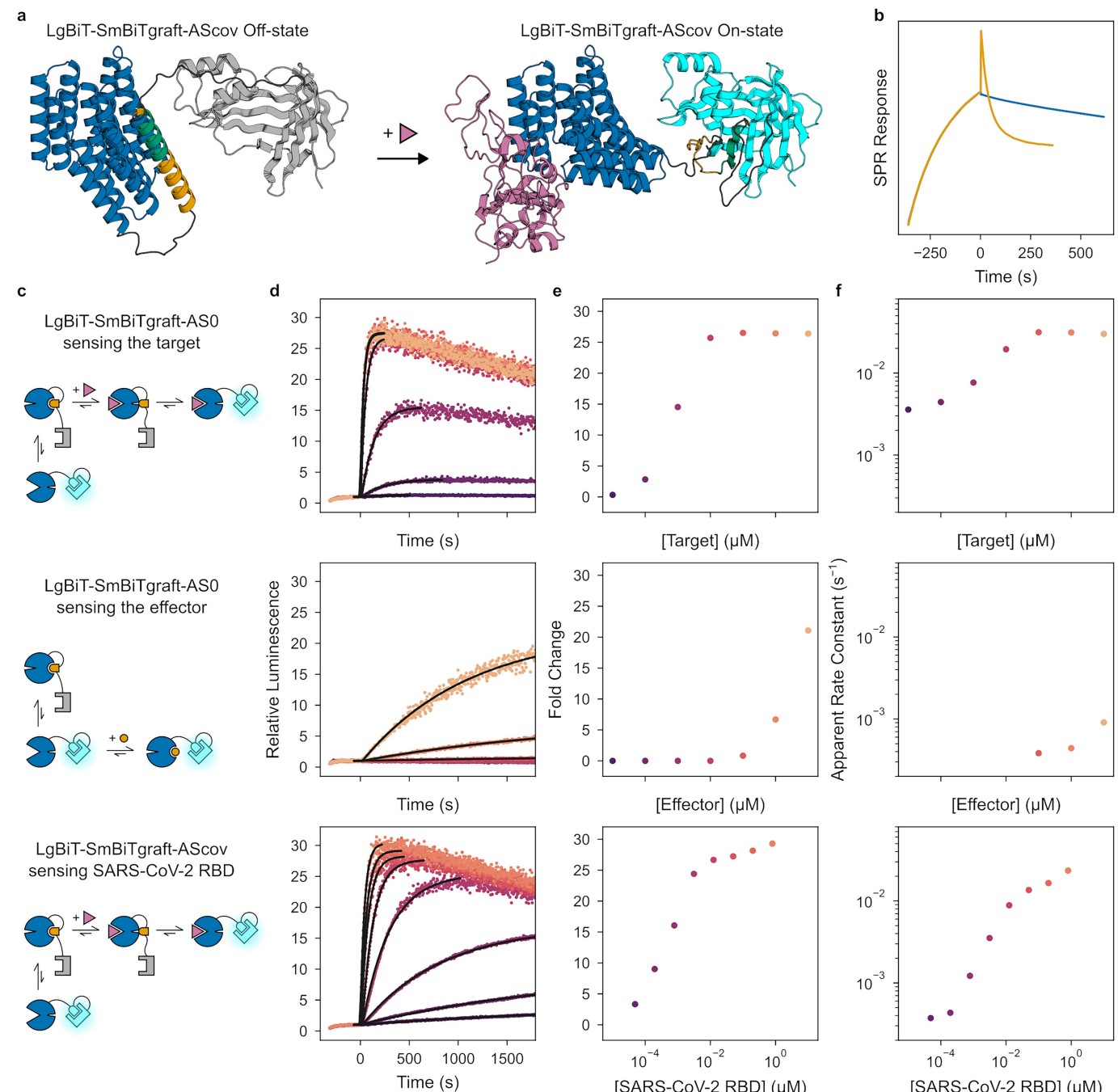

**Extended Data Fig. 6 | Construction and characterization of rapid sensors.**
**a**, Structural model of the best SARS-CoV-2 sensor construct, comprising AScov (blue), the SmBiTgraft peptide with the effector (orange) and grafted SmBiT (green), LgBiT (grey or cyan), and flexible linkers (black). In this design, SmBiT is caged in a helical conformation when SmBiTgraft is bound to the switch and is free to reconstitute the luciferase when SmBiTgraft is released. To form SmBiTgraft, SmBiT was grafted onto the effector peptide such that most of its hydrophobic residues are buried within the switch–SmBiTgraft interface when in the bound helical conformation. Because the original effector peptide binds so strongly to the switch, it could accommodate replacing some of its interface residues with residues from SmBiT without reducing its affinity so much that it no longer effectively cages the SmBiT. **b**, SPR data showing sfGFP-SmBiTgraft binding to AS0 (blue and orange, association phase), slow subsequent dissociation in the absence of target (blue), and rapid subsequent dissociation upon addition of 10 μM target (orange) caused by rapid target

binding to form a transient ternary complex, causing the spike at the beginning of the dissociation phase. **c**–**f**, Rapidly sensing the target through a facilitated dissociation mechanism (top), slowly sensing the effector limited by the slow base exchange rate of SmBiTgraft between binding AS0 and LgBiT (middle), and rapidly sensing the SARS-CoV-2 RBD with facilitated dissociation (bottom). **c**, Schematics showing the mechanism of sensing. **d**, Luminescence time courses (normalized to the initial signal) of 10 pM sensor construct then at time 0 adding varying concentrations of analyte (target, effector, or SARS-CoV-2 RBD); data (colours) fit (black) with single exponentials up to the maximum signal for time courses which showed appreciable signal increase. In some time courses, signal slowly decreases due to depletion of luciferase substrate. **e**, Luminescence signal fold change plotted against analyte concentration. **f**, Sensor response rate constant plotted against analyte concentration for time courses that showed appreciable signal increase.

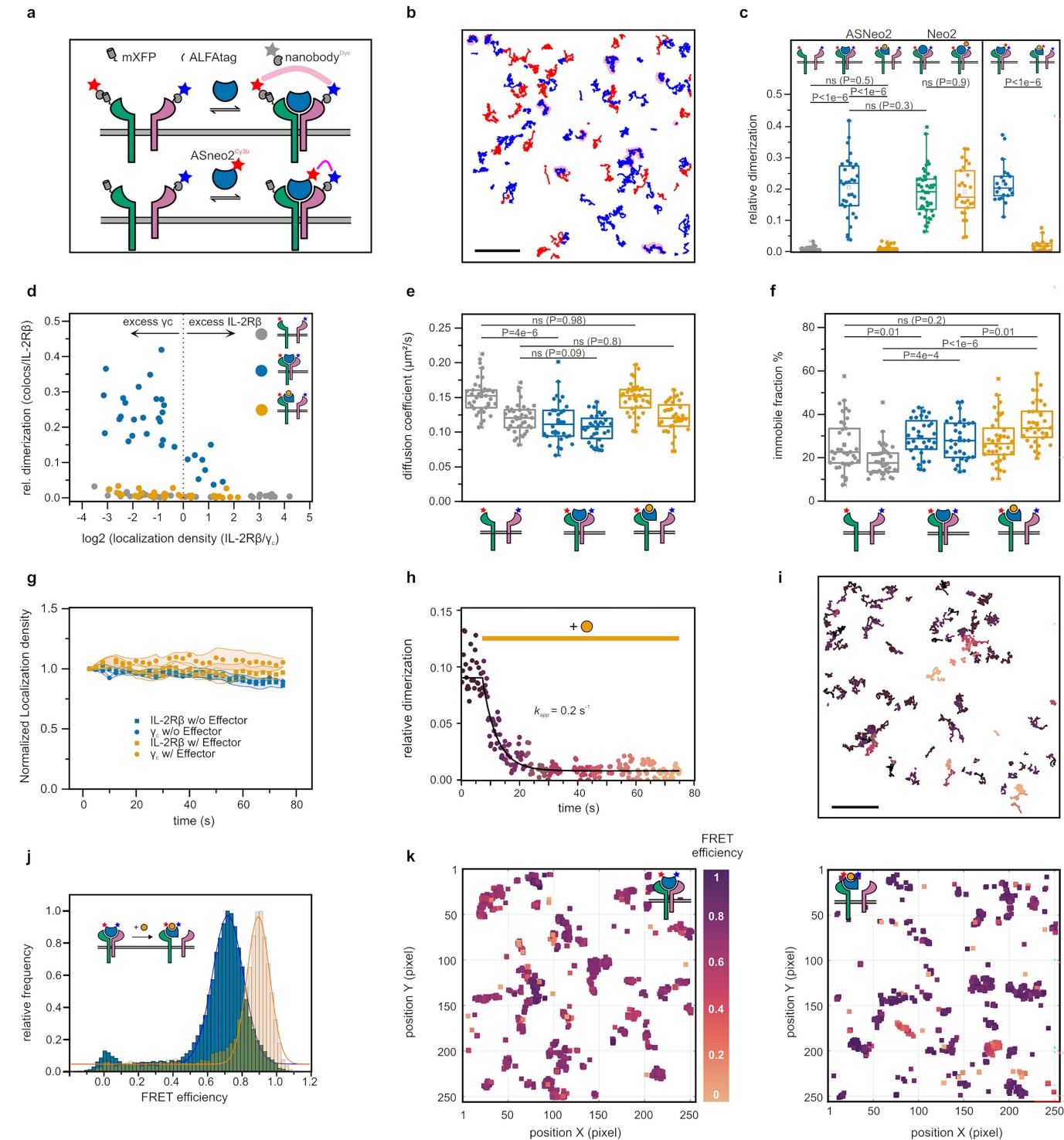

**Extended Data Fig. 7** | See next page for caption.

**Extended Data Fig. 7 | Detailed functional characterization of ASNeo2.**
**a**, Schematic depiction of the labelling strategy for single-molecule tracking experiments. **b**, Single-molecule trajectories of IL-2Rβ (red), $\gamma_c$ (blue) and ASNeo2-induced heterodimers (magenta). **c**, Data from Fig. 5e as box plots to display datapoint variation including Neo2 (+/- effector) (green and orange, left side) and single-molecule tracking experiments with labelled ASNeo2 and $\gamma_c$ (right side). Sample sizes and independent repeats are: unstimulated: 37 and 3; ASNeo2: 32 and 3; ASNeo2 + Effector: 33 and 3; Neo2: 44 and 4; Neo2 + Effector: 26 and 2; labelled ASNeo2: 21 and 2; labelled ASNeo2 + Effector: 18 and 2. **d**, Relative dimerization in relation to receptor cell surface density ratio indicates that high dimerization data variance is caused by differing IL-2Rβ to $\gamma_c$ ratios at the plasma membrane. Even at high $\gamma_c$ excess, effector-bound ASNeo2 shows no residual affinity for $\gamma_c$. **e**, Diffusion properties of IL-2Rβ and $\gamma_c$ are reverted to the ground state after addition of effector. **f**, Immobile particles are increased upon stimulation, but not decreased after effector addition, potentially indicating receptors internalizing in membrane proximal endosomes. For **e**,**f**, the left box always corresponds to IL-2Rβ and the right one to $\gamma_c$. Sample sizes for **d**–**f** are as in **c**. **g**, Normalized localization density over time confirms minimal single-molecule bleaching in long-term single-molecule tracking experiments. Sample sizes and independent repeats are: without Effector: 5 and 5; with Effector: 3 and 3. **h**,**i**, Dissociation of ASNeo2-induced IL-2Rβ/$\gamma_c$ dimers at the cell surface upon addition of 10 μM effector as detected by time-lapse single-molecule co-tracking (**h**) with colour-coded corresponding co-trajectories (**i**). **j**,**k**, Conformational change of ASNeo2 bound to the cell-surface receptor probed by smFRET. **j**, FRET efficiency histograms for ASNeo2 E4C/K211C labelled with Cy3B and ATTO643 bound to cells expressing IL-2Rβ and $\gamma_c$ in the absence (blue) and presence (yellow) of 10 μM effector. Sample sizes are: without Effector: 7; with Effector: 5. **k**, smFRET co-localizations of one individual cell before the effector was added (left) and after it was added (right) colour-coded for FRET efficiency, highlighting the observation of individual molecules. Statistics for **c**,**e**,**f** were performed using two-sided two-sample Kolmogorov–Smirnov tests (ns, not significant, *P* values noted). Box plots show the distribution of the dataset, highlighting the median, quartiles, and outliers, with whiskers extending to the range limits. Scale bars in **b**,**i**: 5 μm.

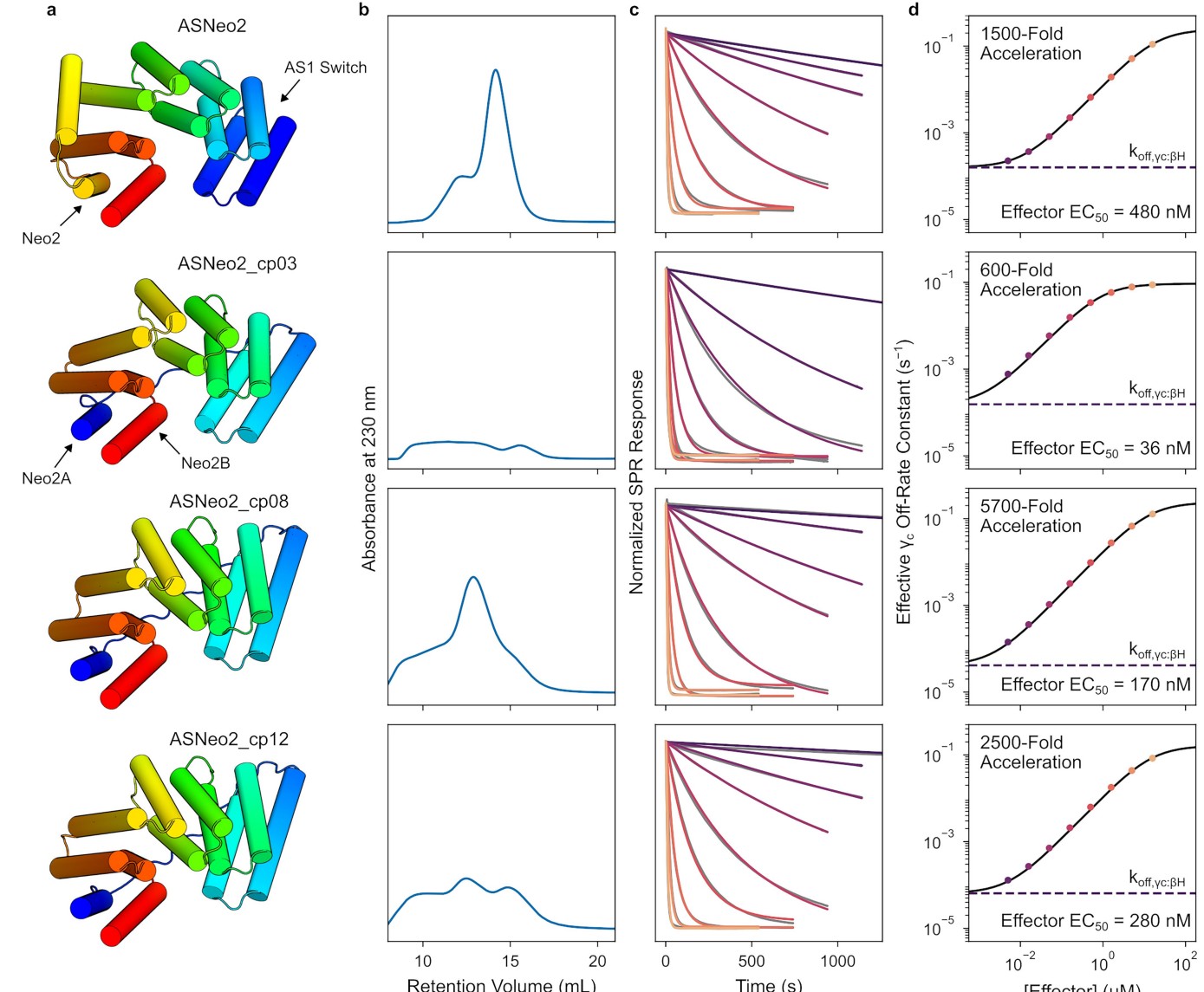

**Extended Data Fig. 8 | Characterization of cyclic permutations of ASNeo2.**
**a**, Design models of ASNeo2 and selected cyclic permutations in state X, rainbow-coloured from N terminus (blue) to C terminus (red) to illustrate the protein topology. In ASNeo2, the switch is at the N terminus and Neo2 is at the C terminus. In the cyclic permutations, although the relative position of the switch and Neo2 changes minimally, the switch is in the middle of the protein, part of Neo2 is at the N terminus, and the other part is at the C terminus. This way, the regulatory switch cannot degrade without also breaking Neo2. **b**, SEC purifications performed on a Superdex 200 Increase 10/300 GL column. The

cyclic permutations are prone to aggregation during expression, but distinct monomer peaks can be picked out. **c**, Fast effector-concentration-dependent dissociation of $\gamma_c$ from the ASNeo2−IL-2Rβ−$\gamma_c$ complex upon addition of peptide effector. Data (grey) fit (colours) as described in methods (neglecting the accumulation modelling because accumulation on the SPR surface was negligible with these proteins). **d**, Rate constants of facilitated $\gamma_c$ dissociation computed from the model fit by $\ln(2) \div \{$half-time of $\gamma_c$−host interaction$\}$ plotted against effector concentration (circles) and fit with hyperbolic equations (black lines).

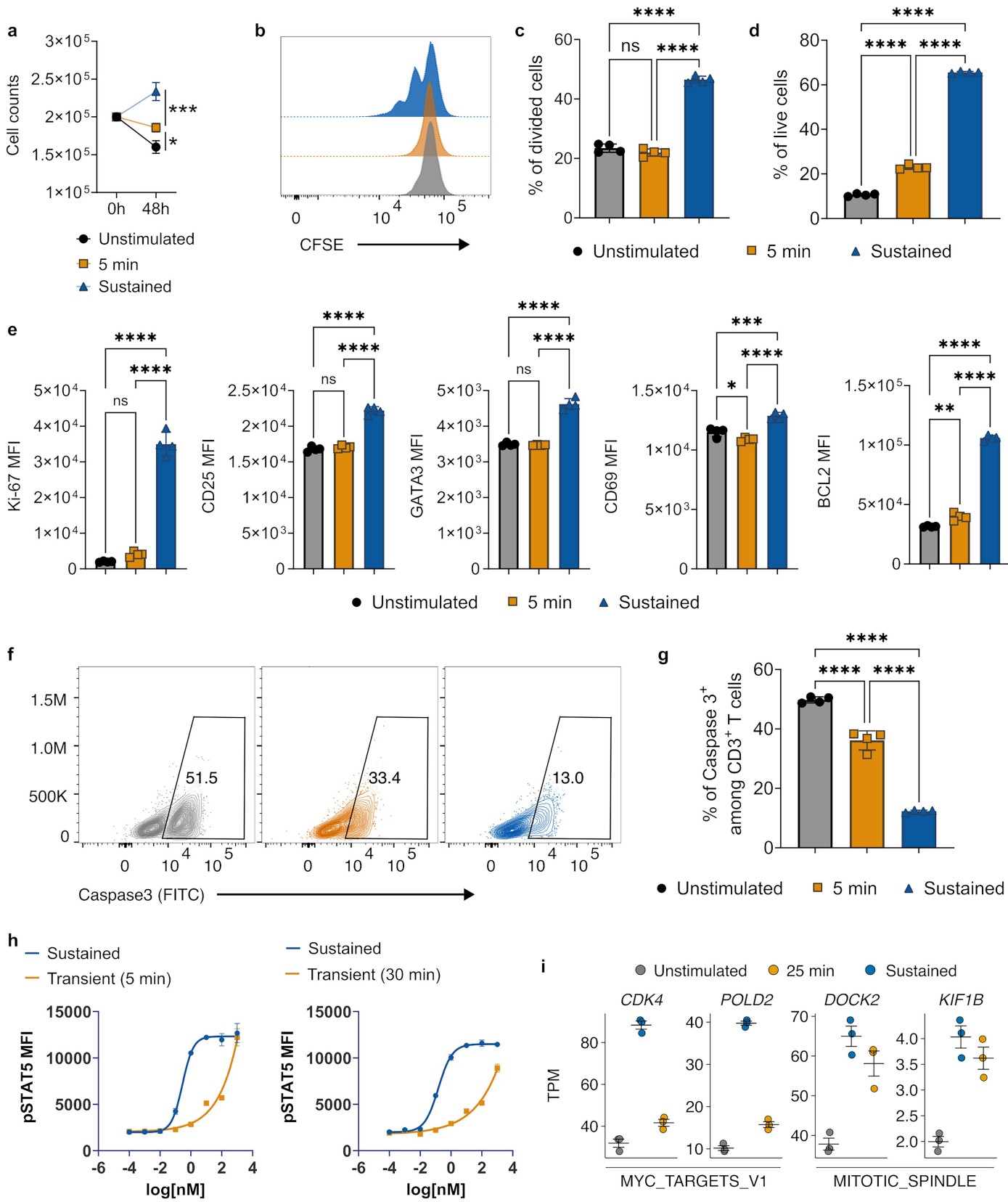

**Extended Data Fig. 9 |** See next page for caption.

**Extended Data Fig. 9 | Additional characterization of differential effects of transient ASNeo2 stimulation on T cell behaviour.** Human T cells were stimulated with 1 nM (**a**–**g**) or 5 nM (**i**) ASNeo2 for 5 min (**a**–**h**), 30 min (**h**), or 25 min (**i**) or left untreated as a control. Signalling was either sustained by continued ASNeo2 treatment or terminated by the addition of 10 μM effector. Cells were collected for counting and phenotypic analysis by flow cytometry after 72 h (**b**,**c**; n = 4) or 48 h (**a**,**e**–**g**; n = 4), for pSTAT signalling analysis by flow cytometry after 20 min (**h**, n = 3), or for RNA-seq analysis after 6 h (**i**, n = 3). **a**, Changes in viable T cell counts from 0 h to 48 h across each group. **b**,**c**, Representative flow-cytometry histogram of CFSE (**b**) and quantitative analysis of divided cells (**c**). **d**, Frequencies of live T cells. **e**, Mean fluorescence intensity (MFI) of Ki-67, CD25, GATA3, CD69, and BCL2. **f**,**g**, Representative flow-cytometry plots (**f**) and quantitative analysis of caspase-3+ cells (**g**; n = 4). **h**, Dose-dependent pSTAT5 curves. **i**, Transcripts per million (TPM) of *CDK4* and *POLD2* in the MYC_TARGETS_V1 gene set (left); TPM of *DOCK2* and *KIF1B* in the MITOTIC_SPINDLE gene set (right). **a**,**c**–**e**,**g**, Statistics were obtained from ANOVA with two-sided Tukey's post-test (ns, not significant ($P > 0.05$), *$P = 0.04$, **$P = 0.001$, ***$P = 0.0002$, ****$P < 0.0001$). Lines or bars represent means; error bars represent s.e.m. (**a**,**h**,**i**) or SD (**c**–**e**,**g**). *n* refers to biologically independent samples.

**Extended Data Table 1 | Crystallographic data collection and refinement statistics for structures AS1_H, AS1_HE and AS1_TH 1**

|  | AS1_H (PDB: 9DCX) | AS1_HE (PDB: 9DCY) | AS1_TH #1 (PDB: 9DCZ) |
|---|---|---|---|
| **Data collection** | | | |
| Space group | $P\ 61\ 2\ 2$ | $P\ 2_1$ | $P\ 6_1\ 2\ 2$ |
| Cell dimensions | | | |
| $a, b, c$ (Å) | 61.54, 61.54, 236.10 | 78.44, 35.98, 88.90 | 80.71, 80.71, 546.85 |
| $\alpha, \beta, \gamma$ (°) | 90, 90, 120 | 90. 109.63, 90 | 90, 90, 120 |
| Resolution (Å) | 48.58 - 1.80 (1.85 - 1.80) | 47.98 - 2.35 (2.47 - 2.35) | 34.75 - 2.90 (2.94 - 2.90) |
| $R_{sym}$ or $R_{merge}$ | 0.084 (0.945) | 0.090 (0.221) | 0.369 (2.128) |
| $I / \sigma I$ | 22.53 (8.59) | 8.70 (1.88) | 10.5 (2.1) |
| Completeness (%) | 99.31 (100.00) | 99.66 (99.89) | 99.90 (100) |
| Redundancy | 12.7 (13.3) | 4.8 (5.1) | 31.4 (34.2) |
| | | | |
| **Refinement** | | | |
| Resolution (Å) | 48.58 - 1.80 (1.85 - 1.80) | 47.98 - 2.35 (2.47 - 2.35) | 34.75 - 2.90 (2.94 - 2.90) |
| No. reflections | 25499 (1764) | 19956 (2831) | 24845 (960) |
| $R_{work} / R_{free}$ | 0.2135 (0.2862) / 0.2552 (0.3298) | 0.1937 (0.2226) / 0.2437 (0.3186) | 0.2094 (0.3779) / 0.2550 (0.4580) |
| No. atoms | | | |
| Protein | 1965 | 4340 | 6010 |
| Ligand/ion | n/a | n/a | n/a |
| Water | 92 | 59 | 7 |
| *B*-factors | | | |
| Protein | 38 | 42 | 84 |
| Ligand/ion | n/a | n/a | n/a |
| Water | 44 | 37 | 56 |
| R.m.s. deviations | | | |
| Bond lengths (Å) | 0.008 | 0.001 | 0.003 |
| Bond angles (°) | 0.81 | 0.34 | 0.52 |

Values in parentheses are for the highest-resolution shell.

**Extended Data Table 2 | Crystallographic data collection and refinement statistics for structures AS1_TH 2, AS1_THE 1 and AS1_THE 2**

| | AS1_TH #2 (PDB: 9DD0) | AS1_THE #1 (PDB: 9DD1) | AS1_THE #2 (PDB: 9OLQ) |
|---|---|---|---|
| **Data collection** | | | |
| Space group | $P\,2_1\,2_1\,2_1$ | $P\,43\,2_1\,2$ | $C\,2$ |
| Cell dimensions | | | |
| $\quad a, b, c$ (Å) | 62.98, 74.07, 211.77 | 110.85, 110.85, 65.53 | 285.58, 96.40, 74.69 |
| $\quad \alpha, \beta, \gamma$ (°) | 90, 90, 90 | 90, 90, 90 | 90, 90.39, 90 |
| Resolution (Å) | 30.25 - 3.88 | 28.21 - 3.70 | 41.16 - 3.48 |
| | (4.34 - 3.88) | (3.93 - 3.70) | (3.67 - 3.48) |
| $R_{sym}$ or $R_{merge}$ | 0.270 (0.717) | 0.190 (0.951) | 0.316 (2.305) |
| $I / \sigma I$ | 5.80 (3.5) | 10.70 (2.65) | 4.9 (1.2) |
| Completeness (%) | 99.80 (100.00) | 100.00 (100.00) | 99.6 (99.4) |
| Redundancy | 9.6 (9.4) | 11.8 (12.1) | 6.9 (7.1) |
| | | | |
| **Refinement** | | | |
| Resolution (Å) | 30.25 - 3.88 | 28.21 - 3.70 | 41.16 - 3.48 |
| | (4.34 - 3.88) | (3.93 - 3.70) | (3.57 - 3.48) |
| No. reflections | 9667 (1344) | 4667 (756) | 25996 (1868) |
| $R_{work} / R_{free}$ | 0.2261 (0.2743) / | 0.2472 (0.3015) / | 0.2475 (0.3276) / |
| | 0.2789 (0.3296) | 0.2772 (0.3010) | 0.2871 (0.3802) |
| No. atoms | | | |
| $\quad$ Protein | 5891 | 3062 | 15510 |
| $\quad$ Ligand/ion | n/a | n/a | n/a |
| $\quad$ Water | n/a | n/a | n/a |
| $B$-factors | | | |
| $\quad$ Protein | 115 | 92 | 111 |
| $\quad$ Ligand/ion | n/a | n/a | n/a |
| $\quad$ Water | n/a | n/a | n/a |
| R.m.s. deviations | | | |
| $\quad$ Bond lengths (Å) | 0.003 | 0.002 | 0.002 |
| $\quad$ Bond angles (°) | 0.53 | 0.40 | 0.40 |

Values in parentheses are for the highest-resolution shell.

**Extended Data Table 3 | Crystallographic data collection and refinement statistics for structures AS5_H, AS5_HE, LHD101An1 and CS221B**

| | AS5_H (PDB: 9DD2) | AS5_HE (PDB: 9DD3) | LHD101An1 (PDB: 9DD4) | CS221B (PDB: 9DD5) |
|---|---|---|---|---|
| **Data collection** | | | | |
| Space group | *P 43* | *C 2* | *P 2₁2₁2₁* | *C 2* |
| Cell dimensions | | | | |
| *a, b, c* (Å) | 62.37, 62.37, 78.50 | 90.17, 35.66, 75.52 | 64.97, 70.66, 76.39 | 54.67, 25.09, 34.67 |
| α, β, γ (°) | 90, 90, 90 | 90, 116.2, 90 | 90, 90, 90 | 90, 102.77, 90 |
| Resolution (Å) | 48.84 - 1.72 (1.75 - 1.72) | 31.69 - 1.64 (1.68 - 1.64) | 51.88 - 2.11 (2.17 - 2.11) | 33.81 - 1.5 (1.62 - 1.50) |
| $R_{sym}$ or $R_{merge}$ | 0.047 (0.709) | 0.062 (0.537) | 0.037 (0.911) | 0.060 (0.788) |
| $I / \sigma I$ | 18.50 (2.2) | 6.62 (1.14) | 21.25 (1.80) | 12.57 (1.71) |
| Completeness (%) | 100.00 (100.00) | 91.01 (95.00) | 99.63 (99.82) | 99.71 (99.38) |
| Redundancy | 9.0 (9.2) | 1.9 (1.9) | 6.4 (6.5) | 6.7 (6.8) |
| | | | | |
| **Refinement** | | | | |
| Resolution (Å) | 48.84 - 1.72 (1.75 - 1.72) | 31.69 - 1.64 (1.68 - 1.64) | 51.88 - 2.11 (2.17 - 2.11) | 33.81 - 1.5 (1.62 - 1.50) |
| No. reflections | 31945 (2281) | 24347 (1919) | 20777 (1711) | 7489 (1451) |
| $R_{work}$ / $R_{free}$ | 0.2035(0.2755)/ 0.2431(0.3195) | 0.2018(0.3285)/ 0.2509(0.3520) | 0.2222(0.3295)/ 0.2617(0.3301) | 0.2091(0.2512)/ 0.2494(0.3124) |
| No. atoms | | | | |
| Protein | 2001 | 2149 | 1976 | 430 |
| Ligand/ion | n/a | n/a | n/a | n/a |
| Water | 125 | 70 | 16 | 45 |
| *B*-factors | | | | |
| Protein | 40 | 35 | 80 | 31 |
| Ligand/ion | n/a | n/a | n/a | n/a |
| Water | 46 | 41 | 69 | 46 |
| R.m.s. deviations | | | | |
| Bond lengths (Å) | 0.006 | 0.003 | 0.002 | 0.004 |
| Bond angles (°) | 0.68 | 0.53 | 0.35 | 0.60 |

Values in parentheses are for the highest-resolution shell.

# Reporting Summary

## Statistics

For all statistical analyses, confirm that the following items are present in the figure legend, table legend, main text, or Methods section.

| n/a | Confirmed | |
|---|---|---|
| ☐ | ☒ | The exact sample size (*n*) for each experimental group/condition, given as a discrete number and unit of measurement |
| ☐ | ☒ | A statement on whether measurements were taken from distinct samples or whether the same sample was measured repeatedly |
| ☐ | ☒ | The statistical test(s) used AND whether they are one- or two-sided *Only common tests should be described solely by name; describe more complex techniques in the Methods section.* |
| ☒ | ☐ | A description of all covariates tested |
| ☒ | ☐ | A description of any assumptions or corrections, such as tests of normality and adjustment for multiple comparisons |
| ☐ | ☒ | A full description of the statistical parameters including central tendency (e.g. means) or other basic estimates (e.g. regression coefficient) AND variation (e.g. standard deviation) or associated estimates of uncertainty (e.g. confidence intervals) |
| ☐ | ☒ | For null hypothesis testing, the test statistic (e.g. *F*, *t*, *r*) with confidence intervals, effect sizes, degrees of freedom and *P* value noted *Give P values as exact values whenever suitable.* |
| ☒ | ☐ | For Bayesian analysis, information on the choice of priors and Markov chain Monte Carlo settings |
| ☒ | ☐ | For hierarchical and complex designs, identification of the appropriate level for tests and full reporting of outcomes |
| ☐ | ☒ | Estimates of effect sizes (e.g. Cohen's *d*, Pearson's *r*), indicating how they were calculated |

*Our web collection on statistics for biologists contains articles on many of the points above.*

## Software and code

Policy information about availability of computer code

| | |
|---|---|
| Data collection | RFdiffusion (https://github.com/RosettaCommons/RFdiffusion), inpainting with RosettaFold (https://github.com/RosettaCommons/RFDesign), ProteinMPNN (https://github.com/dauparas/ProteinMPNN), and AlphaFold2 were used along with custom code to design proteins, which has been deposited at Zenodo (https://doi.org/10.5281/zenodo.16749263). PyMOL (v2.5.5) was used during some protein design steps. Pulse shapes for DEER experiments were calculated with PulseShape (https://github.com/StollLab/PulseShape). MD simulations were performed using GROMACS 2020.2. SPR data was collected on a Biacore 8k with Biacore Insight software version 5.0.18.22405. FP, FRET, and luminescence data was collected on a Synergy Neo2 plate reader with BioTek Gen5 software version 3.11 or 3.14. |
| Data analysis | Proteins were visualized with PyMOL (v2.5.5). Kinetic data from SPR, FP, FRET, and luminescence measurements was analyzed with custom code. Nonlinear least squares optimization was performed with LMFIT version 1.3.1. Crystallography data was analyzed using CCP4 program suite (v8.0, particularly the modules Pointless, Aimless, Phaser, and Coot v0.9.8), Phenix (v1.21.1), MolProbity (https://github.com/rlabduke/MolProbity/tree/641e0eb52e88fc5f4b99409f297b82020620177c), and XDS (20230630). DEER data was analyzed using eprTools (https://github.com/mtessmer/eprTools). MD data was analyzed with GROMACS 2020.2 and MDAnalysis version 2.4.3. Single molecule co-tracking was analyzed using SLIMfast4C available at https://zenodo.org/record/5712332. The Matlab R2022b script for FRET efficiency analysis accompanied with a demo dataset is provided as Supplementary Software in https://doi.org/10.1038/s41467-024-49876-9. Flow cytometry data was analyzed with CytExpert (v2) or FlowJo (v10.10.0). Statistics were performed using GraphPad Prism (v10.5.0). RNA-seq data was analyzed with Rsubread (v2.16.1, particularly with the module featureCounts after alignment), DESeq2 (v1.42.1), and FGSEA (v1.28.0). Code to analyze all data and to generate all figures has been deposited at Zenodo (https://doi.org/10.5281/zenodo.16749448). |

For manuscripts utilizing custom algorithms or software that are central to the research but not yet described in published literature, software must be made available to editors and reviewers. We strongly encourage code deposition in a community repository (e.g. GitHub). See the Nature Portfolio guidelines for submitting code & software for further information.

# Data

Policy information about availability of data

All manuscripts must include a data availability statement. This statement should provide the following information, where applicable:

- Accession codes, unique identifiers, or web links for publicly available datasets
- A description of any restrictions on data availability
- For clinical datasets or third party data, please ensure that the statement adheres to our policy

All data generated during this study are included in either the main text or as Supplementary Information. Data deposition, atomic coordinates, and structure factors for all crystal structures reported in this paper have been deposited in the Protein Data Bank (PDB), https://www.rcsb.org/ with accession codes 9DCX, 9DCY, 9DCZ, 9DD0, 9DD1, 9DD2, 9DD3, 9DD4, 9DD5, and 9OLQ. PDB models and sequences for all designs and source data with analysis scripts has been deposited at Zenodo under https://doi.org/10.5281/zenodo.16749448. Single molecule tracking data has been deposited at Zenodo: calibration beads, unstimulated samples, long term measurements and labeled ligand experiments under https://doi.org/10.5281/zenodo.13957447, Neo2 and Neo2 + effector under https://doi.org/10.5281/zenodo.13957498 and ASNeo2 and ASNeo2 + effector, as well as smFRET data under https://doi.org/10.5281/zenodo.13957540. The raw RNA-seq data have been deposited in the NCBI Sequence Read Archive (SRA) under BioProject accession code PRJNA1302552 at https://ncbi.nlm.nih.gov/bioproject/PRJNA1302552. The SKEMPI database can be accessed at https://life.bsc.es/pid/skempi2/database/index. The reference genome GRCh38 (accession code GCF_000001405.26) can be accessed at https://hgdownload.soe.ucsc.edu/goldenPath/hg38/bigZips/.

# Research involving human participants, their data, or biological material

Policy information about studies with human participants or human data. See also policy information about sex, gender (identity/presentation), and sexual orientation and race, ethnicity and racism.

| | |
|---|---|
| Reporting on sex and gender | Sex and gender information was not collected in this study. |
| Reporting on race, ethnicity, or other socially relevant groupings | Race and ethnicity information was not collected in this study. |
| Population characteristics | This study included no human participants. |
| Recruitment | This study included no human participants. |
| Ethics oversight | This study included no human participants requiring ethics oversight. |

Note that full information on the approval of the study protocol must also be provided in the manuscript.

# Field-specific reporting

Please select the one below that is the best fit for your research. If you are not sure, read the appropriate sections before making your selection.

☒ Life sciences ☐ Behavioural & social sciences ☐ Ecological, evolutionary & environmental sciences

For a reference copy of the document with all sections, see nature.com/documents/nr-reporting-summary-flat.pdf

# Life sciences study design

All studies must disclose on these points even when the disclosure is negative.

| | |
|---|---|
| Sample size | Sample sizes were chosen to obtain confident enough measurements to demonstrate significant differences between groups. The main text SPR, CD, FP, FRET, and luminescence experiments, and also a subset of DEER and SEC experiments, were performed multiple times, and one representative sample was reported due to low variance between replicates. Having established low variance for these experiments, similar experiments in the supplement were performed multiple times or once. All microscopy, cell staining, and qPCR/RNA-seq measurements were conducted with sample size >=3. No sample size calculations were performed; rather the sample sizes were chosen based on experience and were sufficient for the important observed differences between groups to be strongly statistically significant. |
| Data exclusions | In Fig 5h, we excluded from analysis signaling activity data obtained from batches of YT-1 cells that were unresponsive to stimulation. |
| Replication | All main text SPR, CD, FP, FRET, and luminescence measurements were performed multiple times to ensure reproducibility and low variance. Having established low variance for these experiments, similar measurements in the supplement were performed multiple times or once. To establish reproducibility of SEC experiments in the supplement, designs AS0, AS1, AS2, AS5, AS116, AS117, and AS118 were tested 3 times from 3 separate expressions with similar results, so other SEC experiments were performed once. A subset of DEER experiments were performed multiple times with similar results, so other DEER experiments were performed once. In particular, SPR in Fig. 2d,h, 4b; FRET in Fig. 4c; SEC in Fig. S9a were performed twice. SPR in Fig. 2a,e,f,g; FP in Fig. 4b, S5c, S12; luminescence in Fig. 4d,e; SEC in Fig. S6b; DEER with AS1 in Extended Data Fig. 2 were performed three or more times. For all microscopy, cell staining, and qPCR/RNA-seq experiments, three or four independent biological replicates were performed unless otherwise noted in the figure caption. All attempted replications of all experiments were successful, with the exception that some batches of YT-1 cells were unresponsive to stimulation; the findings in Fig. 5h were not observed with those batches. |

| Randomization | Allocation was random. |
|---|---|
| Blinding | Blinding was not relevant because all groups in each experiment were analyzed the same way. |

# Reporting for specific materials, systems and methods

We require information from authors about some types of materials, experimental systems and methods used in many studies. Here, indicate whether each material, system or method listed is relevant to your study. If you are not sure if a list item applies to your research, read the appropriate section before selecting a response.

### Materials & experimental systems

| n/a | Involved in the study |
|---|---|
| ☐ | ☒ Antibodies |
| ☐ | ☒ Eukaryotic cell lines |
| ☒ | ☐ Palaeontology and archaeology |
| ☒ | ☐ Animals and other organisms |
| ☒ | ☐ Clinical data |
| ☒ | ☐ Dual use research of concern |
| ☒ | ☐ Plants |

### Methods

| n/a | Involved in the study |
|---|---|
| ☒ | ☐ ChIP-seq |
| ☐ | ☒ Flow cytometry |
| ☒ | ☐ MRI-based neuroimaging |

## Antibodies

| Antibodies used | Alexa Fluor® 647-conjugated Phospho-STAT5 (Tyr694) rabbit monoclonal antibody (Cell Signaling Technology, #9365, C71E5), nanobodies for Single-molecule tracking experiments: Anti-GFP and anti-ALFAtag (produced in-house). The following antibodies were purchased from BioXCell: human CD3 (OKT3, BE0001-2) and human CD28 (9.3, BE0248). The following antibodies or staining reagents were purchased from BioLegend: human CD3 (OKT3, 317324), human GATA3 (W19195B, 386906), human CD69 (FN50, 310932), human CD25 (BC96, 302611), human BCL2 (100, 658708), human Ki-67 (Ki-67, 350526), and Human TruStain FcX™ (422302). The following antibodies were purchased from BD Biosciences: pSTAT5 (47/Stat5, 612599), and BD Pharmingen™ FITC Active Caspase-3 Apoptosis Kit (571606). |
|---|---|
| Validation | Surface marker antibodies purchased from BioLegend (human CD3 (OKT3, 317324), human CD69 (FN50, 310932), human CD25 (BC96, 302611)) were validated by the manufacturer: "Each lot of this antibody is quality control tested by immunofluorescent staining with flow cytometric analysis (https://www.biolegend.com/protocols/cell-surface-flow-cytometry-staining-protocol/4283/)." <br><br>BioLegend human GATA3 antibody (W19195B, 386906) was validated by the manufacturer: "Each lot of this antibody is quality control tested by intracellular flow cytometry using our True-Nuclear™ Transcription Factor Staining Protocol." <br><br>BioLegend human BCL2 antibody (100, 658708) was validated by the manufacturer: "Each lot of this antibody is quality control tested by intracellular immunofluorescent staining with flow cytometric analysis (https://www.biolegend.com/protocols/intracellular-flow-cytometry-staining-protocol/4260/)." <br><br>BioLegend human Ki-67 antibody (Ki-67, 350526) was validated by the manufacturer: "Each lot of this antibody is quality control tested by our Ki-67 staining protocol (https://www.biolegend.com/en-gb/products/pe-cyanine7-anti-human-ki-67-antibody-9084)." <br><br>BD Biosciences pSTAT5 antibody (47/Stat5, 612599) was validated by the manufacturer: "This purified or conjugated mAb was characterized by flow cytometry" yielding positive staining for human PBMC and whole blood cells treated with IL-2. <br><br>BD Pharmingen™ FITC Active Caspase-3 Apoptosis Kit (571606) was validated by the manufacturer by flow cytometry. <br><br>Cell Signaling Technology pSTAT5 antibody (Tyr694, C71E5, #9365) was validated by the manufacturer: "This Cell Signaling Technology antibody is conjugated to Alexa Fluor® 647 fluorescent dye and tested in-house for direct flow cytometric analysis of human cells." <br><br>Antibodies purchased from BioXCell (human CD3 (OKT3, BE0001-2) and human CD28 (9.3, BE0248)) were not validated for binding by the manufacturer, but we found they stimulated the proliferation of human T cells as expected. |

## Eukaryotic cell lines

Policy information about cell lines and Sex and Gender in Research

| Cell line source(s) | HeLa cells for single molecule imaging: German Collection of Microorganisms and Cell Cultures GmbH (ACC 57), YT (CD25+) cells: American Type Culture Collection (ATCC) previously used in [Gaggero et al., Science Immunol 2022, PMID 36459543] |
|---|---|
| Authentication | HeLa cells were authenticated by the manufacturer: "STR analysis according to the global standard ANSI/ATCC ASN-0002.1-2021 (2021) resulted in an authentic STR profile of the reference STR database." YT cells were authenticated by Eurofins using STR profiling. |

| Mycoplasma contamination | HeLa cells were tested negatively for mycoplasma (PCR). |
| Commonly misidentified lines (See ICLAC register) | No commonly misidentified cell lines were used in this study. |

## Plants

| Seed stocks | *Report on the source of all seed stocks or other plant material used. If applicable, state the seed stock centre and catalogue number. If plant specimens were collected from the field, describe the collection location, date and sampling procedures.* |
| Novel plant genotypes | *Describe the methods by which all novel plant genotypes were produced. This includes those generated by transgenic approaches, gene editing, chemical/radiation-based mutagenesis and hybridization. For transgenic lines, describe the transformation method, the number of independent lines analyzed and the generation upon which experiments were performed. For gene-edited lines, describe the editor used, the endogenous sequence targeted for editing, the targeting guide RNA sequence (if applicable) and how the editor was applied.* |
| Authentication | *Describe any authentication procedures for each seed stock used or novel genotype generated. Describe any experiments used to assess the effect of a mutation and, where applicable, how potential secondary effects (e.g. second site T-DNA insertions, mosiacism, off-target gene editing) were examined.* |

## Flow Cytometry

### Plots

Confirm that:

☒ The axis labels state the marker and fluorochrome used (e.g. CD4-FITC).

☒ The axis scales are clearly visible. Include numbers along axes only for bottom left plot of group (a 'group' is an analysis of identical markers).

☒ All plots are contour plots with outliers or pseudocolor plots.

☒ A numerical value for number of cells or percentage (with statistics) is provided.

### Methodology

| Sample preparation | For the YT signaling assay (Fig. 5h), cells were stimulated with 1 nM ASNeo2 or Neo2 for 5 minutes at 37°C and then either immediately fixed in 4% PFA at 37°C or transferred to a new well containing Effector, Ruxolitinib or nothing. Cells were then fixed in 4% PFA at 37°C at different timepoints and finally re-fixed in 4% PFA for 15 minutes at room temperature. Following fixation, the cells were washed once with PBS containing 0.5% BSA (PBSA) and permeabilized with 100% methanol for 45 minutes at 4°C. After permeabilization, the cells were washed twice with PBSA and stained for 1 hour at room temperature using Alexa Fluor® 647-conjugated Phospho-STAT5 (Tyr694) rabbit monoclonal antibody (Cell Signaling Technology, #9365). The cells were washed three times, then analyzed by flow cytometry. <br> For T cell surface marker staining (Ext. Data Fig. 9e), cells were collected into U-bottom 96-well plates (Thermal Fischer Scientific), blocked with Human TruStain FcX™ (BioLegend), and incubated with indicated antibodies at 4 °C for 20 min, followed by live/dead staining by 4',6-diamidino-2-phenylindole (DAPI, Thermo Fisher Scientific). Cells were then washed and resuspended with FACS buffer (PBS containing 0.2 % BSA, Sigma-Aldrich) for flow cytometry analyses. For T cell phospho-STAT staining (Ext. Data Fig. 9h), primary human T cells were rested in T cell medium lacking IL-2 for 24h before signaling assays. Cells were plated in a 96-well round bottom plate in 50µl T cell medium. Cells were stimulated with 50 µL of ASNeo2 for 5 or 25 minutes, followed by the addition of effector peptide and incubation for another 20 minutes at 37°C, and the reaction was terminated by fixation with 1.5% paraformaldehyde (PFA) for 15 min at room temperature with agitation. Cells were washed and permeabilized with ice-cold 100% methanol for 60min on ice. Afterward, cells were washed with FACS buffer before staining with pSTAT5 antibodies for 1h at 4°C in the dark. Cells were washed and resuspended in FACS buffer for flow cytometry analyses. For T cell caspase-3 staining (Ext. Data Fig. 9g), cells were first stained for surface markers and Zombie Violet Fixable Dye (BioLegend), followed by staining with a FITC Active Caspase-3 Apoptosis Kit (BD Biosciences) according to the manufacturer's protocol. For transcription factor staining, cells were first stained for surface markers and Zombie Violet Fixable Dye, then fixed and permeabilized using a Foxp3/Transcription Factor Staining Buffer Set (eBioscience) as per the manufacturer's instructions. Cells were subsequently incubated with the indicated antibodies for intracellular staining. Detection was performed using a CytoFlex (Beckman Coulter), and data were analyzed with FlowJo (v10.10.0). |
| Instrument | CytoFlex S flow cytometer (Beckman Coulter) |
| Software | CytExpert, FlowJo |
| Cell population abundance | For the YT signaling assay, no cell sorting was performed. For the primary T cell assays, the proportion of CD3+ T cells among live singlets ranged from 90% to 100%. |
| Gating strategy | YT cell populations were identified based on forward/side scatter profiles. Primary T cells were gated sequentially on singlets, live cells, and then CD3+ cells. Gating examples for YT cells and primary T cells are shown in Supp. Fig. 17. |

☒ Tick this box to confirm that a figure exemplifying the gating strategy is provided in the Supplementary Information.

