## [Peer Review File · Nature]

Design of facilitated dissociation enables timing of cytokine signaling

Corresponding Author: Mr Adam Broerman

Version 1:

Reviewer comments:

Referee #1

(Remarks to the Author)

1. It is a good idea (Fig. 1), and it works.

2. The kinetic and binding data shows that they have built what they wanted to build (Fig. 2), and the structural data (Fig. 3) shows that the logic was right (namely, that they built what they designed). Given that I am out of the field, I will leave it to other reviewers to evaluate the rigor of the technical aspects of the experiments.

3. The second half of the paper (applications, Figs 4 and 5) adds bulk, but not much substance (it is additional to, but not necessary, the main result).

Referee #2

(Remarks to the Author)

Broerman et al. report the generation of an allosteric modulator of protein dissociation and demonstrate its utility. The work is innovative, and the conclusions are well-supported by the presented data.

The authors may want to consider addressing the following points:

Amino acid residue numbering in the coordinate files should be consistent between the same protein in different crystal structures. It should also match descriptions in the manuscript text.

The effector peptide shows noticeable structural/positional variation between the AS1 AC and ABC structures, especially toward the C-terminus of the helix, which is partially unfolded in both molecules in the asymmetric unit of the ABC structure. Is this reflecting a conformational strain, or could it be an artifact caused by lysine methylation?

Some graphs showing experimental data (SPR, CD, etc.) are missing an axis label with the unit.

Referee #3

(Remarks to the Author)

This is an interesting manuscript by Broerman et al. describing the development of engineered protein switches that are designed to rapidly dissociate stable protein complexes of interest. The authors adapt a previously described allosteric switch protein as a modular component for triggering effector-induced dissociation of protein complexes – initially focusing on a synthetic protein:protein binding pair to explore design parameters of the approach and then applying this to demonstrate kinetic control over IL-2 cytokine signaling. Overall, the described approach is quite novel and in general

rigorously pursued, although I have specific questions about some issues that I feel need further clarification and justification. The results presented are in some cases quite dramatic in demonstrating the ability to regulate protein complex stabilities and kinetics in a systematic manner.

However, the current manuscript is very dense and overly difficult to follow, detracting from the presentation of the results. In particular, the initial introduction expends far too much text on abstract discussion of principles and does not thoroughly or effectively introduce the explicit design strategies. In particular, I think the authors should focus on more carefully introducing the components of their specific partner:host pair and the allosteric switch protein that they incorporate into their new designs, providing more background information for the readers. Conceptually the approach is not that complicated but the introduction as written leaves the reader in an abstract space trying to imagine the experimental details, when focusing on the actual experimental design and introducing the specific protein components would eliminate most of that unnecessary complexity. I would really encourage the authors to focus primarily on a simplified description of the experimental design and results, while separating out conceptual interpretations, which in some cases may not be entirely supported experimentally and may not be particularly useful practically. In addition, I would also encourage the authors to simplify the nomenclature as much as possible and to keep this consistent throughout the manuscript.

Lines 53-55: Allostery and facilitated dissociation are two different concepts and should be clearly delineated. An allosteric mechanism may speed up dissociation but that is distinct from facilitated dissociation. The approach taken here incorporates a bit of both concepts.

Lines 58-63: This section is not clear. "Treating the ternary complex..." needs more thorough description. What fusion are we talking about here? Is the incorporation of Hooke's law into this helpful? I did not see this as particularly helpful in understanding the experimental approach and design.

Lines 84-90: This section is very abstract yet it also clearly is referring to a specific model that the authors have in mind that has not been introduced to the readers (e.g. "each helix contacts several others"). I think it would be better to introduce and focus on the concrete model to be engineered - keeping this section at an abstract conceptual level just makes the ideas more difficult to follow and seems somewhat artificial. Explaining the actual experimental system better will help convey the conceptual approach much more clearly.

Lines 93-105: This entire section needs to present background information on the initial design components clearly and thoroughly. Where does the LHD101A-LHD101B heterodimer pair come from? Introduce its properties - K_d, flexibility, X and Y state conformational change and any other relevant information. Explain why this was chosen as the host:partner pair. Clarify the naming/nomenclature and tie these more explicitly (and repeatedly) to the host:partner nomenclature used in the manuscript.

For the cs221 switch: explain where state X and Y for this system are coming from by providing background on the peptide effector induced conformational changes. Be clear and concrete so that the reader doesn't need to refer to earlier papers to understand the specific designs developed here.

The nomenclature, experimental design and cartoon representations could also be changed throughout the manuscript to unify and clarify the presentation. For example, use H and P instead of A and B in the kinetic schemes. Include S as the modular switch/hinge (cs221) which is integrated with H and generate cartoons that more faithfully represent the domain-based design of the allosterically controlled H-S fusions. Why call the effector "C" – when it could be "E"? In the current format, switching between the various descriptions of the system creates unnecessary overhead and barriers to following the work.

Fig. 1: This is a specific model - annotate it with the binder pair and hinge that are used as components in the design. Highlight the newly designed components that have been added to better explain how the modular fusion generates steric clashes when the effector is bound. Color the molecules by the starting components and newly designed structures/interfaces. Show the two states of the hinge plus/minus effector peptide more clearly. Unify the nomenclature as much as possible as described above. I did not find Fig. 1d. helpful – I think showing a cartoon schematic of host+partner+hinge/switch domain that more closely mimics the structural components would be more helpful. Other figures (e.g. Fig. S4 and elsewhere) could also be annotated with the models of the components - e.g. cs221, effector peptide, LHD101A and LHD101B.

Lines 111-113: This was not clear. Maybe the sFig. 5 graphics could show more clearly the impact on the effector peptide binding site in the redesign. How is the register shift promoted?

Lines 126-128: What are the concentrations of effector used in these experiments, and how do these compare to the K_d values for effector binding alone?

Lines 129-130: This sentence was not clear to me.

Fig. 2a: What happens in the presence of excess partner? Given results presented later, this would be a useful/important control to see what the impact of a direct competitor has on the dissociation kinetics.

Line 176 and Fig. 3a,b insets: The insets are too small to be of much use.

Lines 187-188: The authors should address here whether AS1 has lower affinity for partner as a result of this clash.

Lines 190-192: At high concentrations, the ternary complex becomes the dominant state, which is an important consideration for the design of dissociable systems. How much is partner affinity reduced by effector binding? How does this potentially limit the utility of this approach and the concentration ranges over which an effector will be effective? I think this is something the authors could expand on here and also address in the conclusions of the paper.

Lines 251-252: A table of dissociation rates would be useful for comparisons here. Is the primary advantage that the X state of the redesigns has higher affinity vs AS1? I think ratios of Kds or koff's for the two X/Y states for the variable designs would be helpful.

Lines 264-269: The AF2 predictions suggest how the strain may be resolved, but is this deeper analysis really justified given that these are just predictions that are subject to errors that could be on a similar magnitude as "real" structural changes? The concern is that this is potentially an overinterpretation of what is governing the functional differences. At least there is some discussion of the AF2 errors in their text describing local/global deformations, but I am not convinced that the emphasis on this analysis adds substantially to the paper.

Fig. 4 and elsewhere: The cartoons should be redesigned to more closely correspond to the actual structural design. The designs append an allosterically modulated domain onto a binder such that the allosteric conformational change (induced by the effector) generates a clash with the partner. I would show the modularity of this design more clearly in all of the figures.

Fig. 4a: What are the baseline dissociation rates in the presence of excess partner? The authors should also clarify how the measurements are made in the figure legend. Fig. 4b: The significant impact of adding partner alone should be noted here and in the text - that acceleration is presumably strictly competitive and a reflection of the intrinsic kinetic dynamics of the E2-partner:AS114 complex. Fig. 4d: Analyte? Keep nomenclature consistent.

Lines 289-295: The authors should better explain the value and details of this design. In the main text, refer to H2 as the "reporter", as is done in Ext. Data 5 legend. The new "reporter hinge" H2 responds to release of E2-partner upon E2 triggering of trapped complex. Does E2-partner have two potential modes of interaction with AS114 – why not? This needs to be explained a bit more clearly.

Lines 300-304: The slowness of the split luciferase reporter is not due to the luciferase - but just due to the AS1/partner complex. Are the authors implying that there is a kinetic role for the LgBiT/SmBiT components here?

Lines 312-313: How was the smBiT caged? What is the kinetic stability of this modified effector peptide complex?

Fig. 5d: The authors should include data on a control competitor here as a comparison to the effector peptide.

Lines 398-406: What are the practical downsides to using a flexible peptide as an effector? What are use cases that the authors are imagining here?

Lines 423-426: "...with complexity rivaling that powering life." is unnecessarily excessive here.

Referee #4

(Remarks to the Author)

This manuscript describes the design of three-component protein systems in which a protein that can undergo a hinge-bending conformational transition (referred to variously as a "binder") is coupled to the binding of a second "partner" protein and a third "effector" protein that binds to a remote site and promotes dissociation of the partner by promoting an allosteric shift in the binder toward the low affinity (with respect to the partner) state. An overarching goal is to develop a binder that binds the partner protein with high affinity, but that can be rapidly triggered to drive dissociation. To achieve rapid dissociation, the authors attempt to design effectors that bind with an induced fit mechanism. They achieve this by choosing an alpha helical effector that is disordered in the unbound state, which they posit will allow it to bind to a narrow effector channel and force open the hinge in a folding-on-binding mechanism. Fig. 2 demonstrates using SPR that the authors have achieved their goal of achieving fast facilitated dissociation quite well, and Fig. 3 provides a structural view that corroborates their design strategy as well as their functional data. The authors also expand their strategy to other systems, including controlling a split luciferase construct and in modulating interferon-based cell signaling. On the whole, this is an impressive piece of protein design. One weakness of the manuscript is that it is not that easy to read, due to the omission of (or burial of) a lot of key information. However, I do think this can be improved, and some suggestions for revision that would make this a more readable manuscript are provided below.

1. There are a lot of important details left out of the main text. One notable area for this is the methods used to design the proteins. Although some info is provided in the methods section, including a little description would help the reader a lot. All it would take is a couple of sentences saying that, e.g., RFdiffusion methods were used with whatever constraints were applied, MPNN was used to design sequences, etc. The level of success the authors have achieved with their design is impressive, and while it may seem routine to the authors, to the general readers of Nature it is not. For most readers, AI methods for protein design seem a bit like pulling rabbits out of hats, and not describing the methods will not help bring general readers up to speed. Another area where more detail would be helpful is in describing the SPR experiments. For example, in Fig. 2e, the authors measure the rate of dissociation of C from the BAYC complex. From the main text and

legend it is not clear how this reaction is triggered; I had assumed C is flowed in at high concentrations at time zero, which seemed like it might introduce other rate-limiting steps (C association, X->Y isomerization) in addition to the intended dissociation. It was only when I read the methods that I learned that the authors drove the ternary complex by flowing C at high concentration, and then initiated dissociation by flowing buffer. Just adding a single sentence here (and for other SPR experiments) would clear this up.

2. It is not clear to me what the authors mean by "power stroke" in this context. Please define in the main text.

3. It would be helpful if the names of different proteins in the ternary complex were used more consistently. In particular, the protein that binds and switches conformations is sometimes called a "binder", sometimes a "switch", sometimes a "host", and sometimes "A". What is wrong with staying with the original nomenclature of "binder, partner, and effector" as originally defined? Or a better biochemical connection could be made using "Receptor", "Ligand" and "Effector" and abbreviating R, L, and E.

4. In Fig. 2e, since the progress curve is biphasic, which fitted rate constant is selected to represent $k_{offB:AYC}$ and what is the rationale?

5. Several of the progress curves in Fig. 2 would benefit from being plotted with the log of time. For example, in Fig. 2a it is clear that adding effector increases the rate of dissociation (dashed curve), but the fold-change cannot be interpreted because the curve is smashed against the y-axis. Using $\log t$ would allow this to be assessed quantitatively from the raw data. Likewise for both panels of 2e,f (in the lower panels, it is unclear that the velocity continues to increase with C concentration in 2e, and not clear it is plateauing in 2f).

6. The CD spectra in Fig. 2 have no y-axis label.

7. Why on the right panel of Fig. 2f is only 1/10th of the vertical scale used? Perhaps they want the same scale as in 2g,h, but the saturation effect is not well represented at this scale.

8. The authors refer to "rates" of reaction when they should be referring to "rate constants". For example, y-axis labels of right-hand panels of Fig. 2e-h. The accepted nomenclature is that rates of reaction are akin to velocities (with units of molar per second), whereas rate constants multiply concentrations to give those rates (units sec^{-1} , or $M^{-1} \text{sec}^{-1}$).

9. It is a little unclear how rate constants for various steps are determined. For example, in Fig. 2f right panel, is k_{switch} simply determined by fitting a hyperbola to the apparent rate constants and taking the limiting value? And for Fig. 2g,h, are the values of k_{switch} and k_{off} taken from the limits of the hyperbolae? And are k_{off} values for the ternary complexes taken from Fig. 2e,f top? If so, they seem a bit off from the dotted lines (e.g., $2e^{-1}$ vs $2.1e^{-1}$, $3e^{-1}$ vs what looks like $2.8e^{-1}$). A better way to get these rate constants would be to write out the differential equations for all the kinetic transformations, and solve numerically in a global fit of all the data. This appears to be described for some of the systems described later in the manuscript, but not for the main system described in Fig. 2. Such an approach would not only lead to more accurate estimates of rate constants, it would avoid misassignment of rate constants, which is a common pitfall in analysis of multistep kinetics, where apparent rate constants have contributions from multiple kinetic steps.

Version 2:

Reviewer comments:

Referee #2

(Remarks to the Author)

All my comments have been addressed.

Referee #3

(Remarks to the Author)

The authors have addressed my previous comments in the revised manuscript and this version is significantly easier to follow. I have only a few minor comments/suggestions:

Lines 240-242: This sentence (" For AS1, target dissociation...") is not clear to me - what is the link between the effector kinetics and limit on target dissociation here?

Line 262: This section is interesting but a bit difficult to follow. how should the reader understand "deformation magnitude" - is this the extent of the expected X-Y conformational change? How does this related to the extent of clashes with the target in the different designs?

Lines 273-277: I didn't see these points illustrated in Fig. S11b or S11e but such a figure could be helpful to the reader.

Line 328: "tha" should be "that"

Line 364: Fig. 5d says 1500 fold while the text says 2000-fold.

Referee #4

(Remarks to the Author)

The authors have adequately addressed my concerns and suggestions. I am in favor of publishing.

(Remarks on code availability)

I would not be able to provide useful comments on the code.

Reviewer Response

We deeply thank the reviewers for their insightful and constructive comments! We noticed two main concerns: first that the manuscript is dense and hard to read, and second that the nomenclature is complex and confusing.

We agree that, especially in the introduction, the initially submitted manuscript focused too much on abstract principles, detracting from a ready understanding of the actual approach we took to explore the design of facilitated dissociation. Thus, we rewrote the majority of the introduction and updated Figure 1 to introduce the concepts more concretely alongside describing the actual proteins we designed and experiments we conducted.

We also agree that the nomenclature is complex and thought hard about how it could be simplified or clarified. To improve clarity, we renamed the “partner” to the “target”; we think this word better evinces the modularity of the system (that we could in principle generate hosts which can be rapidly dissociated from any *target*), and also works when describing the rapid sensors, so we no longer need to use the word “analyte.” As the reviewers suggested, we also changed our single-letter names to abbreviate the full names of each part: A→H (host), B→T (target), and C→E (effector). We ultimately concluded we could not further simplify the nomenclature without introducing unacceptable vagueness. In this system, we have five components: the “target,” the “binder” to the target, the “effector,” the “switch” which changes conformation in response to the effector, and the “host” which is formed by fusing the switch to the binder. The binder and switch refer to parts of the host, not the full host, and having names for each part and their fusion enables the text to refer to each specifically as needed. We have included a note on nomenclature explaining this in the supplementary information.

In addition, we have included further experiments which we think substantially add to the paper. First, we have solved an additional crystal structure of the strained ternary complex which does not suffer from artifacts of lysine methylation. Second, we have characterized facilitated dissociation in the reverse direction for several designs. Comparing the fold acceleration in the forward and reverse directions for each design, we find that the topology of the protein affects the distribution of strain throughout the ternary complex (Fig. 4b, Ext. Data Fig. 4, S12). Third, we have modified our rapid sensor platform to detect SARS-CoV-2, rather than our original *de novo* target (Fig. 4d, Ext. Data Fig. 6). This further demonstrates the modularity of these designs and the kinetic advantage of facilitated dissociation. Fourth, in collaboration with Dr. Chris Garcia’s lab, we have used transient stimulation with ASNeo2 to study early events in IL-2 signaling (Fig. 5i-n).

We have addressed each specific comment below (original reviewer comments in blue, our responses in black).

Thank you once again,
Adam, Florian, and David

Referee #1:

1. It is a good idea (Fig. 1), and it works.
2. The kinetic and binding data shows that they have built what they wanted to build (Fig. 2), and the structural data (Fig. 3) shows that the logic was right (namely, that they built what they designed). Given that I am out of the field, I will leave it to other reviewers to evaluate the rigor of the technical aspects of the experiments.
3. The second half of the paper (applications, Figs 4 and 5) adds bulk, but not much substance (it is additional to, but not necessary, the main result).

We sincerely thank the reviewer for their assessment. Regarding point 3, as supported by the other reviewers, we do believe that the applications we show are important, as they demonstrate the wide applicability of this mechanism to achieve behaviors that were previously challenging to engineer.

Referee #2:

Broerman et al. report the generation of an allosteric modulator of protein dissociation and demonstrate its utility. The work is innovative, and the conclusions are well-supported by the presented data.

We sincerely thank the reviewer for this assessment and for the points they raised below.

The authors may want to consider addressing the following points:

Amino acid residue numbering in the coordinate files should be consistent between the same protein in different crystal structures. It should also match descriptions in the manuscript text. We have updated the numbering in the coordinate files to be consistent with the manuscript text and among structures.

The effector peptide shows noticeable structural/positional variation between the AS1 AC and ABC structures, especially toward the C-terminus of the helix, which is partially unfolded in both molecules in the asymmetric unit of the ABC structure. Is this reflecting a conformational strain, or could it be an artifact caused by lysine methylation?

We believe this is an artifact of the lysine methylation, as there is a lysine present in the effector peptide interface near the location of strain. To address this, we solved a higher-resolution crystal structure of the ternary complex which used four hydrophobic surface mutants rather than lysine methylation to increase hydrophobicity enough to crystallize. In this structure (now shown in Fig. 3d), the effector peptide conformation agrees well with the unstrained AC structure.

Some graphs showing experimental data (SPR, CD, etc.) are missing an axis label with the unit.

We thank the reviewer for catching this. We have added the axis label for the CD plot in Figure 2. For the SPR plots, the units of response on the y-axis are essentially arbitrary, so we opted to simplify the figure by not explicitly showing them.

Referee #3:

This is an interesting manuscript by Broerman et al. describing the development of engineered protein switches that are designed to rapidly dissociate stable protein complexes of interest. The authors adapt a previously described allosteric switch protein as a modular component for triggering effector-induced dissociation of protein complexes – initially focusing on a synthetic protein:protein binding pair to explore design parameters of the approach and then applying this to demonstrate kinetic control over IL-2 cytokine signaling. Overall, the described approach is quite novel and in general rigorously pursued, although I have specific questions about some issues that I feel need further clarification and justification. The results presented are in some cases quite dramatic in demonstrating the ability to regulate protein complex stabilities and kinetics in a systematic manner.

However, the current manuscript is very dense and overly difficult to follow, detracting from the presentation of the results. In particular, the initial introduction expends far too much text on abstract discussion of principles and does not thoroughly or effectively introduce the explicit design strategies. In particular, I think the authors should focus on more carefully introducing the components of their specific partner:host pair and the allosteric switch protein that they incorporate into their new designs, providing more background information for the readers. Conceptually the approach is not that complicated but the introduction as written leaves the reader in an abstract space trying to imagine the experimental details, when focusing on the actual experimental design and introducing the specific protein components would eliminate most of that unnecessary complexity. I would really encourage the authors to focus primarily on a simplified description of the experimental design and results, while separating out conceptual interpretations, which in some cases may not be entirely supported experimentally and may not be particularly useful practically. In addition, I would also encourage the authors to simplify the nomenclature as much as possible and to keep this consistent throughout the manuscript. We sincerely thank the reviewer for their assessment, for this insightful critique, and for all the specific points they raised below. As suggested by the reviewer, we have rewritten the introduction and updated Figure 1 to more concretely introduce the concepts, prioritizing description of the actual proteins we designed and experiments we conducted over abstract principles. We have also improved the consistency and clarity of the nomenclature as described above.

Lines 53-55: Allosteric and facilitated dissociation are two different concepts and should be clearly delineated. An allosteric mechanism may speed up dissociation but that is distinct from facilitated dissociation. The approach taken here incorporates a bit of both concepts.

We have more clearly distinguished this in the revised introduction by explaining how facilitated dissociation uses the strained ternary intermediate and how binding the effector affects the target (indicative of allostery):

“We set out to design protein systems which undergo facilitated dissociation. Given an interacting protein binder·target pair, we reasoned that we could construct host proteins by fusing an effector-responsive conformational switch to the binder such that when the effector is not bound, the target can bind normally, but in the alternate effector-bound conformation, the switch clashes with the target, leading to strain in the target·host·effector ternary complex which resolves when the target dissociates (Fig. 1d,e). Thus, the effector and target would be allosterically coupled, and facilitated dissociation could proceed through the strained ternary complex intermediate faster than spontaneous dissociation of the target in mutually exclusive competition (Fig. 1c).”

Lines 58-63: This section is not clear. “Treating the ternary complex...” needs more thorough description. What fusion are we talking about here? Is the incorporation of Hooke’s law into this helpful? I did not see this as particularly helpful in understanding the experimental approach and design.

In the revised introduction, we have clarified the more general point that by varying the structure of the switch-binder fusion, we could control the energy of the ternary intermediate and thus the rate of facilitated dissociation:

“To set the ternary intermediate energy within this optimal range (Fig. S2), we reasoned that we could control the level of strain in the ternary complex by varying the geometry of the switch-binder fusion.”

We now wait to introduce Hooke’s law as a way to model this until around Fig. 4, when we actually apply this model, as we agree that describing the specific energy function does not contribute to understanding in the introduction.

Lines 84-90: This section is very abstract yet it also clearly is referring to a specific model that the authors have in mind that has not been introduced to the readers (e.g. “each helix contacts several others”). I think it would be better to introduce and focus on the concrete model to be engineered - keeping this section at an abstract conceptual level just makes the ideas more difficult to follow and seems somewhat artificial. Explaining the actual experimental system better will help convey the conceptual approach much more clearly.

In the revised introduction, we now explain these concepts alongside describing the actual experimental system, rather than listing abstract requirements:

“We began by using a previously designed effector-responsive conformational switch (hinge protein cs221 from ref.30) to test the concept of allosteric coupling via switching steric clashes between the host and target. This switch can undergo a rigid-body hinge motion to switch from a closed state (X) to an open state (Y), and in the open state Y can bind an effector peptide quite tightly ($k_{\text{off}} = 5e-6 \text{ s}^{-1}$). For a model binder·target interaction, we chose a designed heterodimer pair (LHD101 from ref.31, modified as described in Fig. S3) with slow enough dissociation (k_{off}

= $9e-5 \text{ s}^{-1}$) that we could easily measure significant effector-induced acceleration of target dissociation, but not slower than the effector dissociation so that effector binding could still provide sufficient driving force to expel the target.”

Lines 93-105: This entire section needs to present background information on the initial design components clearly and thoroughly. Where does the LHD101A-LHD101B heterodimer pair come from? Introduce its properties - K_d , flexibility, X and Y state conformational change and any other relevant information. Explain why this was chosen as the host:partner pair. Clarify the naming/nomenclature and tie these more explicitly (and repeatedly) to the host:partner nomenclature used in the manuscript.

For the cs221 switch: explain where state X and Y for this system are coming from by providing background on the peptide effector induced conformational changes. Be clear and concrete so that the reader doesn't need to refer to earlier papers to understand the specific designs developed here.

The nomenclature, experimental design and cartoon representations could also be changed throughout the manuscript to unify and clarify the presentation. For example, use H and P instead of A and B in the kinetic schemes. Include S as the modular switch/hinge (cs221) which is integrated with H and generate cartoons that more faithfully represent the domain-based design of the allosterically controlled H-S fusions. Why call the effector “C” – when it could be “E”? In the current format, switching between the various descriptions of the system creates unnecessary overhead and barriers to following the work.

We have simplified the nomenclature here, referring to the target and binder using those names exclusively (rather than interchanging with LHD101A and LHD101B) and using single-letter names which abbreviate the full names. We have also introduced relevant properties of the hinge switch and target:binder pair, explaining why they were chosen (see the quoted text for the previous point). As we mention in the next point, we now show the switch and binder · target pair individually in Fig. 1d, so the reader can easily understand their structure without having to refer to previous papers.

Fig. 1: This is a specific model - annotate it with the binder pair and hinge that are used as components in the design. Highlight the newly designed components that have been added to better explain how the modular fusion generates steric clashes when the effector is bound. Color the molecules by the starting components and newly designed structures/interfaces. Show the two states of the hinge plus/minus effector peptide more clearly. Unify the nomenclature as much as possible as described above. I did not find Fig. 1d. helpful – I think showing a cartoon schematic of host+partner+hinge/switch domain that more closely mimics the structural components would be more helpful. Other figures (e.g. Fig. S4 and elsewhere) could also be annotated with the models of the components - e.g. cs221, effector peptide, LHD101A and LHD101B.

We have replaced the original panel d (the spring model) with a panel introducing the switch and binder individually and showing how they are fused to make hosts. We have colored the new fusion structure separately, and think this is enough to clearly delineate the different regions of the host protein. We have added a new supplementary figure (Fig. S4) which uses more

detailed cartoons paired with structural models of the parts to illustrate how we design the switch-binder fusions and induced-fit switches.

Lines 111-113: This was not clear. Maybe the sFig. 5 graphics could show more clearly the impact on the effector peptide binding site in the redesign. How is the register shift promoted? This is now explained more clearly in the main text:

“We constructed this new state X by shifting the two domains from state Y relative to each other by 1 nm along the axis of a helix at the cleft, then refining this new dock between domains (Fig. S4e, methods).”

and in the Fig. S4 caption:

“Our two-state design process generates sequences for the switch considering the state X alone and the effector-bound state Y. Thus, the switch is only designed to make strong interactions with the effector when in state Y, causing effector binding to promote the register shift to state Y.”

The new cartoons in Fig. S4 illustrate the difference in the effector binding site between the hinge and register shift switches.

Lines 126-128: What are the concentrations of effector used in these experiments, and how do these compare to the K_d values for effector binding alone?

The AS1:effector K_d has been noted in the text:

“To probe the mechanism of facilitated dissociation (Fig 2b), we focused on AS1, the design with the tightest effector binding ($K_{D,HE} \approx 10$ pM).”

The effector concentration used in Fig. 2a has been noted in the caption:

“a, Slow dissociation of the target from the host in the absence of effector (solid) and fast dissociation in the presence of 2 μM effector (dashed) as assessed by SPR.”

Lines 129-130: This sentence was not clear to me.

We have revised this paragraph to more clearly describe how different SPR configurations enable measurement of different kinetic parameters of the system, and this also clarifies the point in this sentence.

“We directly measured the rate constant of target dissociation from the strained ternary complex ($k_{off,T,HE}$) by flowing pre-incubated host·effector complex at high concentration to form the strained ternary complex with the target on the SPR surface, then tracking target:host dissociation under continued flow of the effector (Fig. 2e, S7; methods). In the full facilitated dissociation process, as the concentration of added effector increases, the rate constant of facilitated target dissociation approaches $k_{off,T,HE}$ (Fig. 2g, S7), strongly suggesting that the ternary complex is an intermediate in the facilitated dissociation process (Fig. 2b, lower pathways).”

Fig. 2a: What happens in the presence of excess partner? Given results presented later, this would be a useful/important control to see what the impact of a direct competitor has on the dissociation kinetics.

While the later bulk assays in Fig. 4 require excess partner to prevent rebinding of the host after dissociation slowing the apparent dissociation kinetics, SPR uses flow to ensure that dissociation is mostly permanent. To confirm this, we compared target:host dissociation in all combinations of excess target (as a direct competitor) and effector, finding that the dissociation rates in the presence and absence of direct competitor agree within 2-fold:

Importantly, the presence of direct competitor increases the observed dissociation rates in the presence and absence of effector by the same factor (1.3–1.4), so even though the raw rates are slightly different with the direct competitor, the fold accelerations we measure are unaffected.

Line 176 and Fig. 3a,b insets: The insets are too small to be of much use.

We have zoomed in and enlarged these.

Lines 187-188: The authors should address here whether AS1 has lower affinity for partner as a result of this clash.

This is indeed the case, but we thought this made more sense to mention later (section Modulation of dissociation rate enhancement) as explanation for why we reasoned that repositioning the binder and switch to reduce this clash could also reduce the base off-rate of the target.

“We also hypothesized that the minor strain in the target·AS1 complex (Extended Data Fig. 1c) may increase the base rate of target dissociation, and indeed saw that the target dissociates 20-fold faster from AS1 than from an unhindered binder fusion (Fig. S7).”

Lines 190-192: At high concentrations, the ternary complex becomes the dominant state, which is an important consideration for the design of dissociable systems. How much is partner affinity reduced by effector binding? How does this potentially limit the utility of this approach and the concentration ranges over which an effector will be effective? I think this is something the authors could expand on here and also address in the conclusions of the paper.

We have directly measured the target affinity to the ternary complex to be 200 nM. Effector addition provides a quantitative fold change in target affinity (400-fold for AS1, 4000-fold for AS117). The concentration ranges over which effector can promote target dissociation at equilibrium can in principle be tuned upward or downward by varying the base affinity of the target. However, even at concentrations above the target affinity, effector addition will increase the target off-rate, and excess unlabeled target can be used in combination with the effector to drive rapid exchange of a caged labeled target (as we show for example in Fig. 4c). We have expanded on this in the conclusions:

“The effector provides a quantitative reduction in target affinity (up to four orders of magnitude); effector binding will always increase target exchange rate, and by tuning the basal target affinity, the concentration range over which effector induces target dissociation can in principle be controlled.”

Lines 251-252: A table of dissociation rates would be useful for comparisons here. Is the primary advantage that the X state of the redesigns has higher affinity vs AS1? I think ratios of Kds or koff's for the two X/Y states for the variable designs would be helpful.

We have included a reference to the supplementary table of dissociation rates here. Most designs have increased state X affinity to the target as the reviewer describes, and AS117 additionally has a faster target off rate from the ternary complex.

Lines 264-269: The AF2 predictions suggest how the strain may be resolved, but is this deeper analysis really justified given that these are just predictions that are subject to errors that could be on a similar magnitude as “real” structural changes? The concern is that this is potentially an overinterpretation of what is governing the functional differences. At least there is some discussion of the AF2 errors in their text describing local/global deformations, but I am not convinced that the emphasis on this analysis adds substantially to the paper.

In our new data comparing facilitated dissociation in the forward and reverse directions, the clear relationship between the AF2-predicted direction of deformation and experimental asymmetry of facilitated dissociation (Ext. Data Fig. 4) validates that the AF2 predictions are somehow meaningful, supporting our use of AF2 predictions in predicting facilitated dissociation kinetics. AF2 recapitulates our crystal structures to ~1 Å RMSD accuracy, suggesting AF2 has a ~1 Å RMSD resolution for these proteins. The differences between the unstrained binary and strained ternary complexes it predicts are equal to or greater than this resolution, around 1.0–1.5 Å RMSD. We share the reviewer’s concern that our model relating deformation direction

to kinetic behavior is more speculative, so we have deemphasized it in the main text, moving it from an extended data figure to the supplement (Fig. S10) and relegating most of the description to the figure caption. We do still believe its inclusion in the paper is warranted since it is potentially interesting and may spark a more detailed investigation in the future.

Fig. 4 and elsewhere: The cartoons should be redesigned to more closely correspond to the actual structural design. The designs append an allosterically modulated domain onto a binder such that the allosteric conformational change (induced by the effector) generates a clash with the partner. I would show the modularity of this design more clearly in all of the figures.

We explored cartoons which more clearly convey the modularity of the designs, but could not find a cartoon that incorporated this added complexity without detracting from the clarity of other points where we use these cartoons to illustrate mechanisms or experimental configurations (especially in Figure 2). We instead kept the same cartoons for most of the paper, and added a supplementary figure (Fig. S4) with the more complicated cartoons the reviewer suggests which display the modularity of the design.

Fig. 4a: What are the baseline dissociation rates in the presence of excess partner? The authors should also clarify how the measurements are made in the figure legend. Fig. 4b: The significant impact of adding partner alone should be noted here and in the text - that acceleration is presumably strictly competitive and a reflection of the intrinsic kinetic dynamics of the E2-partner:AS114 complex. Fig. 4d: Analyte? Keep nomenclature consistent.

Fig. 4a: The flow in SPR makes excess partner generally unnecessary to obtain accurate dissociation rates, as described above.

Fig. 4c: We have noted in the text that preventing rebinding is a significant reason the base dissociation rate of E2-target is slow:

“Due to a combination of slow dissociation and rebinding, the release of E2-target from AS114 and subsequent switching of H2 is slow, but this dramatically accelerates upon adding original effector (to switch AS114 to accelerate E2-target dissociation) and excess original target (to prevent E2-target rebinding) (Fig. 4c, Extended Data Fig. 5).”

This is further detailed in the Ext. Data Fig. 5 caption:

“The chain reaction proceeds faster when just excess target is added, likely due to blocking rebinding of E2-target to AS114 after transient dissociation, but this effect is insufficient to achieve full acceleration. The chain reaction also proceeds faster when just effector is added, but likely due to transient rebinding of E2-target to re-form the strained ternary complex, this also does not achieve full acceleration. Adding both effector (to accelerate E2-target dissociation from AS114) and excess target (to prevent E2-target rebinding to AS114) is required to fully accelerate the chain reaction.”

Fig. 4e: “target” now makes sense in place of analyte in the main text, and we can also refer to the Covid RBD explicitly with the new sensors.

Lines 289-295: The authors should better explain the value and details of this design. In the main text, refer to H2 as the “reporter”, as is done in Ext. Data 5 legend. The new “reporter hinge” H2 responds to release of E2-partner upon E2 triggering of trapped complex. Does E2-partner have two potential modes of interaction with AS114 – why not? This needs to be explained a bit more clearly.

This system is intended to demonstrate how facilitated dissociation can be used to construct systems with kinetic behaviors that were previously challenging to design using proteins. This system—rapid release of a kinetically trapped bioactive protein—is reminiscent of DNA systems which rely on toehold-mediated strand displacement; many quite complex and useful functions have been engineered using this behavior.

We now refer to H2 as the reporter in the main text. Like the original target with AS1 and AS114, we expect E2-Target to have two modes of interaction with AS114; this is illustrated in the Fig. 4c cartoon and now is better explained in the text:

“the release of E2-target from AS114 and subsequent switching of H2 is slow, but this dramatically accelerates upon adding original effector (to switch AS114 to accelerate E2-target dissociation)”

Lines 300-304: The slowness of the split luciferase reporter is not due to the luciferase - but just due to the AS1/partner complex. Are the authors implying that there is a kinetic role for the LgBiT/SmBiT components here?

Indeed- by itself, the LgBiT/SmBiT components dissociate quite quickly; here they are only slow to reconfigure due to the high affinity between AS1 and the target. We did not intend to imply a kinetic role for the luciferase components. To avoid confusion, we rephrased this:

“Second, we reasoned our designs could complement split protein systems with high affinity and enable them to be rapidly switched off.”

Lines 312-313: How was the smBiT caged? What is the kinetic stability of this modified effector peptide complex?

We now explain this better in the Ext. Data Fig. 6 caption:

“In this design, SmBiT is caged in a helical conformation when SmBiTgraft is bound to the switch and is free to reconstitute the luciferase when SmBiTgraft is released. To form SmBiTgraft, SmBiT was grafted onto the effector peptide such that most of its hydrophobic residues are buried within the switch:SmBiTgraft interface when in the bound helical conformation. Since the original effector peptide binds so strongly to the switch, it could accommodate replacing some of its interface residues with residues from SmBiT without reducing its affinity so much that it no longer effectively cages the SmBiT.”

The kinetic stability of the SmBiTgraft peptide is shown in Ext. Data Fig. 6b.

Fig. 5d: The authors should include data on a control competitor here as a comparison to the effector peptide.

As described above, the flow in SPR makes excess direct competitor generally unnecessary to obtain accurate dissociation rates.

Lines 398-406: What are the practical downsides to using a flexible peptide as an effector? What are use cases that the authors are imagining here?

Since the peptide effector is unfolded in isolation, it may more easily degrade when used *in vivo*. This could be a benefit or disadvantage, depending on how long the application requires the effector to persist. Otherwise, we have not noticed practical problems when working with the flexible peptide: it is quite soluble and does not significantly aggregate. Here in the conclusions, we are suggesting that flexibility in general helps smooth energy landscapes for conformational transitions. This could be useful generally when designing rapid conformational transitions, and suggests that folding-upon-binding could be a useful molecular mechanism for generating force. One tradeoff with using a flexible effector is that once bound, it slightly less effectively localizes strain to the target interface (Fig. 2g,h, Table S2). We have added a sentence to this paragraph describing this:

“Once bound, however, this more deformable effector less effectively strains the target interface (the target off-rate from the ternary complex is higher with the rigid effector than with the peptide).”

Lines 423-426: “...with complexity rivaling that powering life.” is unnecessarily excessive here. We agree; we have pared this down:

“More generally, our work provides a route to designing not only protein structures, but also the rates and pathways of protein motion and change, which will ultimately enable construction of complex lifelike protein machinery.”

Referee #4:

This manuscript describes the design of three-component protein systems in which a protein that can undergo a hinge-bending conformational transition (referred to variously as a “binder”) is coupled to the binding of a second “partner” protein and a third “effector” protein that binds to a remote site and promotes dissociation of the partner by promoting an allosteric shift in the binder toward the low affinity (with respect to the partner) state. An overarching goal is to develop a binder that binds the partner protein with high affinity, but that can be rapidly triggered to drive dissociation. To achieve rapid dissociation, the authors attempt to design effectors that bind with an induced fit mechanism. They achieve this by choosing an alpha helical effector that is disordered in the unbound state, which they posit will allow it to bind to a narrow effector channel and force open the hinge in a folding-on-binding mechanism. Fig. 2 demonstrates using SPR that the authors have achieved their goal of achieving fast facilitated dissociation quite

well, and Fig. 3 provides a structural view that corroborates their design strategy as well as their functional data. The authors also expand their strategy to other systems, including controlling a split luciferase construct and in modulating interferon-based cell signaling. On the whole, this is an impressive piece of protein design. One weakness of the manuscript is that it is not that easy to read, due to the omission of (or burial of) a lot of key information. However, I do think this can be improved, and some suggestions for revision that would make this a more readable manuscript are provided below.

We sincerely thank the reviewer for their assessment and for their insightful critiques raised below.

1. There are a lot of important details left out of the main text. One notable area for this is the methods used to design the proteins. Although some info is provided in the methods section, including a little description would help the reader a lot. All it would take is a couple of sentences saying that, e.g., RFdiffusion methods were used with whatever constraints were applied, MPNN was used to design sequences, etc. The level of success the authors have achieved with their design is impressive, and while it may seem routine to the authors, to the general readers of Nature it is not. For most readers, AI methods for protein design seem a bit like pulling rabbits out of hats, and not describing the methods will not help bring general readers up to speed. Another area where more detail would be helpful is in describing the SPR experiments. For example, in Fig. 2e, the authors measure the rate of dissociation of C from the BAYC complex. From the main text and legend it is not clear how this reaction is triggered; I had assumed C is flowed in at high concentrations at time zero, which seemed like it might introduce other rate-limiting steps (C association, X->Y isomerization) in addition to the intended dissociation. It was only when I read the methods that I learned that the authors drove the ternary complex by flowing C at high concentration, and then initiated dissociation by flowing buffer. Just adding a single sentence here (and for other SPR experiments) would clear this up.

We thank the reviewer for this critique. With our new streamlined introduction in which we introduce design concepts alongside the actual designs, we feel that simultaneously introducing additional concepts (design methods) would detract from the clarity of the other two. Instead, we have created a new supplementary figure (Fig. S4) which explains the design process, and we have explicitly referred the reader to the methods when introducing new designed proteins in the main text. To explain the SPR experiments more clearly, we have updated the main text to explicitly describe how each experiment was conducted:

“To measure facilitated dissociation kinetics, we used surface plasmon resonance (SPR): with the target affixed to the SPR surface, we associated the host, then measured target:host dissociation under flow of various concentrations of effector.”

“We directly measured the rate constant of target dissociation from the strained ternary complex ($k_{\text{off,T:HE}}$) by flowing pre-incubated host·effector complex at high concentration to form the strained ternary complex with the target on the SPR surface, then tracking target:host dissociation under continued flow of the effector (Fig. 2e, S7; methods).”

“With AS1 affixed to the SPR surface, we measured the rate of effector association to form the strained ternary complex by first saturating AS1 with target, then flowing varying concentrations

of effector mixed with constant excess target (to ensure target remains bound after effector association).”

2. It is not clear to me what the authors mean by “power stroke” in this context. Please define in the main text.

We now define this:

“Such a “driven” motion would be akin to the power strokes of motor proteins: a large conformational change that is both thermodynamically and kinetically favored (a low-barrier descent down a steep energy gradient).”

3. It would be helpful if the names of different proteins in the ternary complex were used more consistently. In particular, the protein that binds and switches conformations is sometimes called a “binder”, sometimes a “switch”, sometimes a “host”, and sometimes “A”. What is wrong with staying with the original nomenclature of “binder, partner, and effector” as originally defined? Or a better biochemical connection could be made using “Receptor”, “Ligand” and “Effector” and abbreviating R, L, and E.

As described above, we simplified the nomenclature as much as possible without, in our view, sacrificing clarity. From the reviewer’s suggestion, single letter names of the three components are now abbreviations (“Target”: “T,” “Host”: “H,” “Effector”: “E”). We kept separate names for binders, switches, and hosts (the fusions between switch and binder which allosterically couple them) as we think it is important to be able to refer to each component specifically, and in the supplement we included a note on nomenclature explaining this distinction.

4. In Fig. 2e, since the progress curve is biphasic, which fitted rate constant is selected to represent koffB:AYC and what is the rationale?

Within this double exponential fit, one exponential corresponded to 85% of the amplitude and the other to 15%, so we chose the rate constant corresponding to the higher amplitude exponential. Generally, the reported rate constant corresponds to the faster and higher amplitude exponential in the fit (for both $k_{\text{off,T:HE}}$ and $k_{\text{off,T:H}}$), and we note instances where other criteria are used to determine which rate constant corresponds to the change of interest. This is described in the SPR fitting methods:

“The reported rate constant typically corresponds to the faster and higher amplitude exponential in the fit; instances where other criteria are used to determine which rate constant corresponds to the change of interest are noted. When clearly only one host population is present (often indicated by $k_{\text{app},1} \approx k_{\text{app},2}$ or a large difference between S_1 and S_2 when fitting a double exponential), the following single exponential decay function was fitted instead, where the parameters are the same as above. Instances where single exponentials were used are noted.

$$S = S_0 + S_1 e^{-k_{\text{app}}(t-t_0)},$$

5. Several of the progress curves in Fig. 2 would benefit from being plotted with the log of time. For example, in Fig. 2a it is clear that adding effector increases the rate of dissociation (dashed

curve), but the fold-change cannot be interpreted because the curve is smashed against the y-axis. Using $\log t$ would allow this to be assessed quantitatively from the raw data. Likewise for both panels of 2e,f (in the lower panels, it is unclear that the velocity continues to increase with C concentration in 2e, and not clear it is plateauing in 2f).

We have generated the following figure containing the Fig. 2 data plotted with the log of time. While we agree that plotting progress curves with the log of time would better enable evaluation of fast kinetics, we think that for the general reader, the appearance of exponential progress curves plotted with the log of time is not intuitive, and so doing so would detract from the general clarity of the figure. We also find it less intuitive to represent the start of the time courses at a $t > 0$ (since $t=0$ cannot be shown on a log scale).

6. The CD spectra in Fig. 2 have no y-axis label.

We thank the reviewer for catching this. We have added the axis label.

7. Why on the right panel of Fig. 2f is only 1/10th of the vertical scale used? Perhaps they want the same scale as in 2g,h, but the saturation effect is not well represented at this scale.

We indeed had intended to match the scales in panels f and h. But we agree that this hampers visualization of the saturation effect, so we have zoomed in the y-axis scale in panel f.

8. The authors refer to “rates” of reaction when they should be referring to “rate constants”. For example, y-axis labels of right-hand panels of Fig. 2e-h. The accepted nomenclature is that rates of reaction are akin to velocities (with units of molar per second), whereas rate constants multiply concentrations to give those rates (units sec^{-1} , or $M^{-1} sec^{-1}$).

We thank the reviewer for this clarification. We have fixed this in the text, figure captions, and axis labels.

9. It is a little unclear how rate constants for various steps are determined. For example, in Fig. 2f right panel, is k_{switch} simply determined by fitting a hyperbola to the apparent rate constants and taking the limiting value? And for Fig. 2g,h, are the values of k_{switch} and k_{off} taken from the limits of the hyperbolae? And are k_{off} values for the ternary complexes taken from Fig. 2e,f top? If so, they seem a bit off from the dotted lines (e.g., $2e-1$ vs $2.1e-1$, $3e-1$ vs what looks like $2.8e-1$). A better way to get these rate constants would be to write out the differential equations for all the kinetic transformations, and solve numerically in a global fit of all the data. This appears to be described for some of the systems described later in the manuscript, but not for the main system described in Fig. 2. Such an approach would not only lead to more accurate estimates of rate constants, it would avoid misassignment of rate constants, which is a common pitfall in analysis of multistep kinetics, where apparent rate constants have contributions from multiple kinetic steps.

In the Fig. 2 caption, we have clarified that k_{switch} is determined from the saturating value of the hyperbolic fit of the apparent effector on-rates:

“apparent effector on-rate constant plotted against effector concentration (circles) and a linear (e) or hyperbolic (f) fit. k_{switch} is the saturating value of the hyperbolic fit.”

All of the facilitated dissociation data (including in Fig. 2) is globally fit as the reviewer describes, and rather the fit hyperbolae are constrained to the limiting k_{off} and k_{switch} values from these global fits. As the reviewer describes, $k_{\text{off, T:HE}}$ values were taken from the top left fits of Fig. 2e,f; we had been reporting rate constants to 1 significant figure, but we have increased the precision to two to remove this inconsistency.

Response to Referee Comments

We sincerely thank the referees for their work reviewing this manuscript. We have responded to each remaining point raised by the referees below (original referee comments in blue, our responses in black).

Referee #2:

All my comments have been addressed.

Referee #3:

The authors have addressed my previous comments in the revised manuscript and this version is significantly easier to follow. I have only a few minor comments/suggestions:

Lines 240-242: This sentence (" For AS1, target dissociation...") is not clear to me - what is the link between the effector kinetics and limit on target dissociation here?

We have rephrased this and the surrounding sentences to clarify our point that effector association and target dissociation are two steps which could each be rate-limiting, and we sought to increase the rate of the current rate-limiting step without reducing the rate of the other so much that it becomes rate-limiting instead:

"We next explored the factors contributing to the dissociation kinetics, seeking to maximize the dissociation rate enhancement by 1) reducing the base rate of target dissociation and 2) increasing the rate of facilitated dissociation. For (1), the target dissociates 20-fold faster from AS1 than from an unhindered binder fusion (Fig. S7) likely due to minor strain in the target·AS1 complex (Extended Data Fig. 1c). For (2), with AS1, effector association can occur at least 5 times faster than target dissociation from the ternary complex, and hence moderately increasing the energy of the ternary complex could accelerate the target dissociation step without making effector association rate limiting, increasing the overall rate of facilitated dissociation."

Line 262: This section is interesting but a bit difficult to follow. how should the reader understand "deformation magnitude" - is this the extent of the expected X-Y conformational change? How does this related to the extent of clashes with the target in the different designs?

We have rephrased multiple locations in the text to clarify that the deformation here refers to the bending of the ternary complex to resolve the designed clash, rather than of the X-Y conformational change, in particular:

"We next sought to determine how the target·host·effector ternary complex deforms to resolve the designed clash."

"We sought to maximize strain energy in the ternary complex by modulating the magnitude and direction of the deformation required to resolve the designed clash"

Additionally, Fig. S10 cited here has a schematic depicting our meaning of deformation magnitude (as mentioned in the next point, this was mistakenly pointing to Fig. S11 which could have contributed to the confusion).

Lines 273-277: I didn't see these points illustrated in Fig. S11b or S11e but such a figure could be helpful to the reader.

The text was mistakenly referring to Fig. S11 when it should have been referring to Fig. S10. Such illustrations are present in Fig. S10. We apologize for this mistake.

Line 328: "tha" should be "that"

We thank the reviewer for catching this. This has been fixed.

Line 364: Fig. 5d says 1500 fold while the text says 2000-fold.

We thank the reviewer for catching this inconsistency. This arose from moving to reporting two significant figures instead of one during the previous round of revision; this has been fixed in the text.

Referee #4:

The authors have adequately addressed my concerns and suggestions. I am in favor of publishing.

Referee #4 (Remarks on code availability):

I would not be able to provide useful comments on the code.